# On the Identifiability of Poisson Branching Structural Causal Model Using Probability Generating Function

**Yu Xiang**[1][*] **Jie Qiao**[1][*] **Zhefeng Liang**[1] **Zihuai Zeng**[1] **Ruichu Cai**[1,2][†] **Zhifeng Hao**[3]

[1]School of Computer Science, Guangdong University of Technology, Guangzhou, China
[2]Peng Cheng Laboratory, Shenzhen, China
[3]College of Science, Shantou University, Shantou, China
{thexiang2000, qiaojie.chn, lzfeng011021, zzhuaiiii, cairuichu}@gmail.com,
haozhifeng@stu.edu.cn

## Abstract

Causal discovery from observational data, especially for count data, is essential across scientific and industrial contexts, such as biology, economics, and network operation maintenance. For this task, most approaches model count data using Bayesian networks or ordinal relations. However, they overlook the inherent branching structures that are frequently encountered, e.g., a browsing event might trigger an adding cart or purchasing event. This can be modeled by a binomial thinning operator (for branching) and an additive independent Poisson distribution (for noising), known as Poisson Branching Structure Causal Model (PB-SCM). There is a provably sound cumulant-based causal discovery method that allows the identification of the causal structure under a branching structure. However, we show that there still remains a gap in that there exist causal directions that are identifiable while the algorithm fails to identify them. In this work, we address this gap by exploring the identifiability of PB-SCM using the Probability Generating Function (PGF). By developing a compact and exact closed-form solution for the PGF of PB-SCM, we demonstrate that each component in this closed-form solution uniquely encodes a specific local structure, enabling the identification of the local structures by testing their corresponding component appearances in the PGF. Building on this, we propose a practical algorithm for learning causal skeletons and identifying causal directions of PB-SCM using PGF. The effectiveness of our method is demonstrated through experiments on both synthetic and real datasets.

## 1 Introduction

Causal discovery from observational data, particularly for count data is a crucial task that arises in numerous applications, including biology (Wiuf and Stumpf [2006]), economic (Weiß and Kim [2014]), network operation maintenance (Qiao et al. [2023], Cai et al. [2022]), etc. Much effort has been made to model and discover the causal structure from count data. One line of research models the count data as a type of Poisson Bayesian network (Park and Raskutti [2015, 2017]) or ordinal functional model (Ni and Mallick [2022]), based on which various types of discovering methods are investigated. However, the Bayesian network and ordinal modeling ignore the inherent branching structure among the counting relationship, which is frequently encountered in real world (Weiß [2018]). Take Fig. 1 (a) as an example, in online shopping, the purchasing event can be inherited from browsing, cart added, or promotion event, which exhibits a branching structure such that website browsing ($X_1$) may lead to either purchasing ($X_4$) directly without adding to the cart, or adding

---

[*]Equal contribution.
[†]Corresponding author.

38th Conference on Neural Information Processing Systems (NeurIPS 2024).

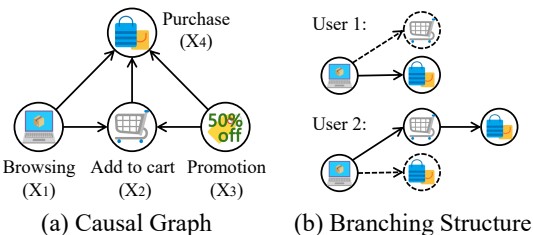

(a) Causal Graph      (b) Branching Structure

Figure 1: (a) An example of the causal graph of count data in an online shopping service. (b) An illustration of the branching structure inherent in count data, where browsing events trigger subsequent events that are summarized as counts.

to the cart ($X_2$) followed by purchasing ($X_4$), as depicted in Fig. 1 (b). Accurately modeling and identifying the underlying causal mechanisms–particularly the branching structure inherent in these count variables–are of interest for providing valuable insights to service providers.

Branching structure modeling is well studied across various domains, notably within the context of the integer-value autoregressive model (Weiß [2018], Al-Osh and Alzaid [1987], McKenzie [1985]). Most of these studies model the relationship among count variables utilizing a thinning operator '∘' (Steutel and van Harn [1979]). For instance, the causal pair $X_1 \to X_4$ in Fig. 1(a) can be modeled as follows: $X_4 = \alpha \circ X_1 + \epsilon$ where $\alpha \circ X_1 := \sum_{n=1}^{X_1} \xi_n^{(\alpha)}$, $\xi_n^{(\alpha)} \overset{\text{i.i.d.}}{\sim}$ Bernoulli$(\alpha)$, $\epsilon \sim$ Pois$(\mu)$. That is, the thinning operator models the branching of an event, e.g., a website browsing event leads to a purchase with probability $\alpha$, while the Poisson noise models the exogenous event. Such modeling can be formalized as the Poisson Branching Structure Causal Model (PB-SCM) (Qiao et al. [2024b]). Recently, Qiao et al. [2024b] further explores the identifiability of PB-SCM and proposes a likelihood-based heuristic searching method for learning causal skeleton and a cumulant-based method for identifying causal direction. Although some identifiability results are developed by Qiao et al. [2024b] using cumulant, we show that there still remains a gap in that there exist causal directions that are identifiable while the algorithm fails to identify them. For example, the current approach cannot fully identify the causal directions among $X_1$, $X_2$, and $X_3$ in Fig. 1 (a). Moreover, the likelihood-based heuristic searching method requires extensive coefficient estimating and structure searching suffering from computational complexity and potential convergence issues.

In this paper, we aim to develop the identifiability of the Poisson Branching Structural Causal Model (PB-SCM) using the Probability Generating Function (PGF) as the characterization of the distribution. Specifically, we first develop the closed-form solution for the PGF of PB-SCM which establishes the connection with the causal structure. With this connection, we find that each component in this closed-form solution is directly associated with a unique local structure, enabling us to learn the skeleton and analyze causal asymmetries without the need for the entire structure. Based on this, we provide an exact pseudo-polynomial time causal structure learning algorithm (i.e., polynomial in the magnitude of the observed variables). We demonstrate the effectiveness of the proposed causal discovery method using synthetic data and real-world data.

Our contributions are threefold. 1) We develop the closed-form solution for PGF of PB-SCM and establish a connection between the closed-form solution of PGF and the graphical patterns. 2) We exploit the closed-form solution of PGF to identify the causal skeleton as well as the causal direction of PB-SCM, which allows us to identify the adjacency and causal direction directly. 3) We propose a practical structure learning algorithm and demonstrate its efficiency and effectiveness through synthetic and real-world data experiments.

## 2  Related work

For brevity, we review causal discovery methods that are applicable and fully identifiable on observational discrete data. Numerous methods and theories have been developed for learning causal structure from observational data (Spirtes et al. [2000], Zhang et al. [2018], Glymour et al. [2019], Cai et al. [2018b], Qiao et al. [2024a]). In particular, for discrete data, one can employ constraint-based methods (Pearl [2009], Spirtes et al. [1995]), score-based methods (Chickering [2002], Tsamardinos et al. [2006]) to identify the causal structure by exploring the conditional independence relation

among variables, but the conditional independence relation can only identify up to the Markov equivalent class (Pearl [2009]). Recently, numerous approaches have been developed for categorical or ordinal data that can identify beyond Markov equivalent class (Cai et al. [2018a], Ni [2022], Ni and Mallick [2022], Peters et al. [2010], Leonelli and Varando [2023], Figueiredo and Oliveira [2023]). However, these methods are rarely suitable for count data. Although some recent works explore the Poisson Bayesian network for the Poisson data (Park and Raskutti [2015, 2017]), and further extend it to the zero-inflated Poisson data (Choi et al. [2020]), there still remains a gap in the identifiability of the Possion branching structure model.

# 3  Poisson Branching Structural Causal Model

In this section, we first introduce the Poisson branching structural causal model (PB-SCM), and then we introduce the preliminary of PGF.

## 3.1  Problem Formulation

Let $\mathbf{X} = \{X_1, \ldots, X_d\}$ denote a $d$-dimensional random count vector, of which the causal relationship consists of a causal directed acyclic graph (DAG) $G(\mathbf{V}, \mathbf{E})$ with the vertex set $\mathbf{V} = \{1, 2, ..., d\}$ and edge set $\mathbf{E}$. We use $Pa(i) = \{j | j \rightarrow i \in \mathbf{E}\}$, $Ch(i) = \{j | i \rightarrow j \in \mathbf{E}\}$ and $Des(i) = \{j | i \rightsquigarrow j \in \mathbf{E}\}$ denote the sets of parents, children and descendants of vertex $i$ in $G(\mathbf{V}, \mathbf{E})$. We assume that any variable in $\mathbf{X}$ satisfies the following Poisson Branching Structural Causal Model (PB-SCM):

**Definition 1** (Poisson Branching Structural Causal Model)**.** *For each random variable $X_i \in \mathbf{X}$, let $\epsilon_i \sim Pois(\mu_i)$ be the noise component of $X_i$, $X_i$ is generated by:*

$$X_i = \sum_{j \in Pa(i)} \alpha_{j,i} \circ X_j + \epsilon_i, \tag{1}$$

*where $\alpha_{j,i} \in (0, 1]$ is the coefficient from vertex $j$ to $i$, and $\alpha \circ X_i$ is a Binomial thinning operator such that $\alpha \circ X_i = \sum_{n=1}^{X_i} \xi_n^{(\alpha)}$, where $\xi_n^{(\alpha)} \overset{i.i.d.}{\sim} \text{Bernoulli}(\alpha)$, independently of $X_i$.*

The formulation is consistent with the existing literature, such as Qiao et al. [2024b], Al-Osh and Alzaid [1987]. We further assume that the *faithfulness assumption*, the *causal Markov assumption*, and the *causal sufficient assumption* hold. These assumptions are commonly used in constraint-based causal discovery methods, e.g., PC algorithm (Spirtes et al. [2000]). With these assumptions, we formalize our goal as follows:

**Goal.**  Given i.i.d. samples $\mathcal{D} = \{x_1^{(j)}, \ldots, x_d^{(j)}\}_{j=1}^m$ from the joint distribution $P(\mathbf{X})$ generated by PB-SCM, our goal is to identify the unknown causal structure $G$ from $\mathcal{D}$.

## 3.2  Preliminary

To develop the identifiability of PB-SCM, we resort to using the probability generating function as the proxy to analyze the distribution and its asymmetry property. Here, we recall the definition of the probability generating function:

**Definition 2** (Probability Generating Function)**.** *Given discrete random vector $\mathbf{X} = [X_1, ..., X_d]^T$ taking values in the non-negative integers $\mathbb{Z}^{\geqslant 0}$, the probability generating function of $\mathbf{X}$ is defined as:*

$$G_{\mathbf{X}}(\mathbf{z}) = \mathbb{E}[z_1^{X_1} \cdots z_d^{X_d}] = \sum_{x_1, ..., x_d = 0}^{\infty} p(x_1, ..., x_d) z_1^{x_1} \cdots z_d^{x_d}, \tag{2}$$

*where $p$ is the probability mass function of $\mathbf{X}$. The power series converges absolutely at least for all complex vectors $\mathbf{z} = [z_1, ..., z_d]^T \in \mathbb{C}^d$ with $\max\{|z_1|, ..., |z_d|\} \leqslant 1$.*

Since the PGF is uniquely defined by the probability mass function, such uniqueness of the power series expansion will in turn define the probabilities, meaning that the analysis of the distribution can be conducted on PGF as a proxy (Johnson et al. [2005]).

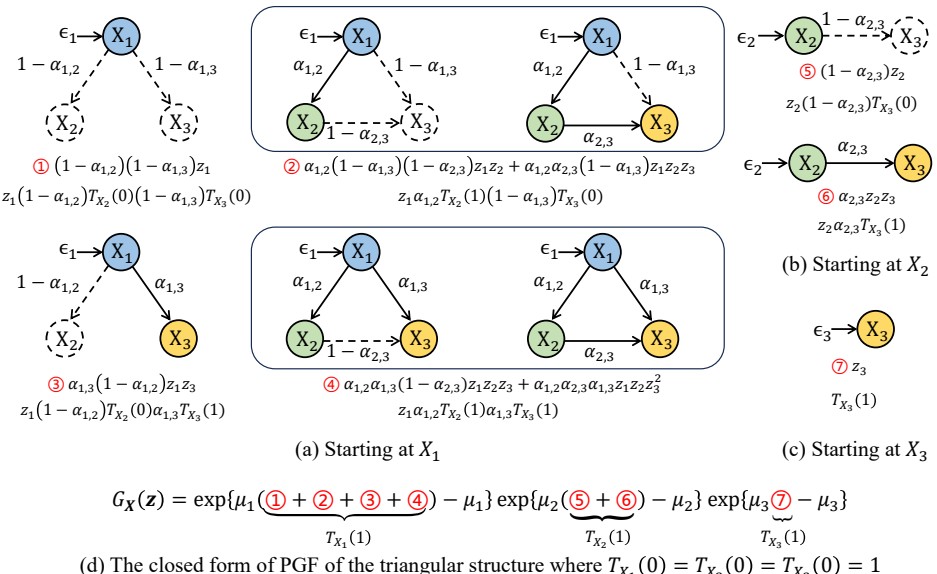

(a) Starting at $X_1$

(b) Starting at $X_2$

(c) Starting at $X_3$

$$G_X(\mathbf{z}) = \exp\{\mu_1(\underbrace{①+②+③+④}_{T_{X_1}(1)}) - \mu_1\}\exp\{\mu_2(\underbrace{⑤+⑥}_{T_{X_2}(1)}) - \mu_2\}\exp\{\mu_3\underbrace{⑦}_{T_{X_3}(1)} - \mu_3\}$$

(d) The closed form of PGF of the triangular structure where $T_{X_1}(0) = T_{X_2}(0) = T_{X_3}(0) = 1$

Figure 3: Illustration of the graphical implication in the closed-form solution for the PGF of PB-SCM. Here, $\alpha_{i,j}T_{X_j}(1)$ indicates that $X_j$ is reached from $X_i$, while $(1 - \alpha_{i,j})T_{X_j}(0)$ indicates the opposite.

# 4 Structure Learning of PB-SCM Using Probability Generating Function

In this section, we demonstrate how to utilize PGF to identify causal structures. We first develop the closed-form solution for the PGF of PB-SCM, which has a compact and exact representation capturing the global structure information. Building on this, we delve into the components of the expansion of the PGF's closed form, demonstrating that each component in PGF is connected to a specific local structure. To effectively capture these components, we develop a local PGF. Finally, based on such a connection, we present theoretical results regarding the graphical implications of the component captured by local PGF, which can be used to discover causal structures.

## 4.1 Motivating Example

Before the formal discussion, we first present a motivating example illustrating how PGF helps analyze the identifiability of PB-SCM.

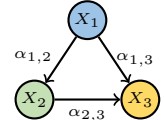

Figure 2: Example triangular structure.

Taking the triangular structure in Fig. 2 as an example, the corresponding closed form of PGF is given in Fig. 3. We can observe that the closed form of the PGF can be separated into three parts, starting at $X_1$, $X_2$, and $X_3$, respectively. Each part of PGF encapsulates the path information in the overall causal structure. For example, considering the term $z_1 z_2 z_3^2$ in ④, the $z_3^2$ represents that there are two paths from $X_1$ to $X_3$ if the path $\alpha_{1,2}, \alpha_{2,3}, \alpha_{1,3}$ are 'open'. By this, one can deduce that for the reserved direction, e.g., $X_3 \to X_2$, such a term $z_3^2$ will not exist, which justifies that the causal direction between $X_2, X_3$ is identifiable.

Another attractive property of the PGF is the ability to analyze the local behavior of the structure. For example, by properly setting $z_1$ approach $0$, we can deduce whether the exist of adjacency between $X_2, X_3$ by testing whether the term $z_2 z_3$ exists as shown in term ⑥. Furthermore, as shown in Fig. 1, by analyzing the local PGF, we can show that the causal direction $X_1 \to X_2 \gets X_3$ is identifiability while the previous cumulant-based method (Qiao et al. [2024b]) can not.

In light of the connection between PGF and the causal structure, we can develop a PGF-based method to effectively identify the causal structure. In the following section, we first demonstrate how the PGF encapsulates the causal structure by developing the closed-form solution of the PGF. Then, benefiting from the local analysis of PGF, we address the identifiability gap of PB-SCM and present a more general identifiability result.

## 4.2 Closed Form Solution of Probability Generating Function

In order to analyze the identifiability of PB-SCM, we first establish a fundamental theorem that provides the closed-form solution for the PGF of PB-SCM:

**Theorem 1** (Closed-form solution for PGF of PB-SCM). *Given a random vector* $\mathbf{X} = [X_1, ..., X_n]^T$ *following PB-SCM, let* $\mathbf{z}_{(j)} = \{z_l | l \in Des(j) \cup \{j\}\}$, *the PGF of* $P(\mathbf{X})$ *is given by* $G_{\mathbf{X}}(\mathbf{z}) = \prod_{i \in [d]} G_{\epsilon_i}\Big(z_i \times \prod_{j \in Ch(i)} G_{i,j}(\mathbf{z}_{(j)})\Big)$, *where*

$$G_{i,j}(\mathbf{z}_{(j)}) = \begin{cases} G_{B(\alpha_{i,j})}\left(z_j \times \prod_{k \in Ch(j)} G_{j,k}(\mathbf{z}_{(k)})\right) &, Ch(j) \neq \emptyset \\ G_{B(\alpha_{i,j})}(z_j) &, Otherwise \end{cases}, \tag{3}$$

*in which* $G_{\epsilon_i}(\cdot)$ *is the PGF of Poisson noise* $\epsilon_i$ *and* $G_{B(\alpha_{i,j})}(\cdot)$ *is the PGF of Bernoulli distribution with parameter* $\alpha_{i,j}$.

The main idea of the proof is to decompose the expectation of PGF according to the causal structure, e.g., a conditional expectation in PGF can be derived as $\mathbb{E}\left[z_3^{\alpha_{1,3} \circ \epsilon_1} | \epsilon_1\right] = G_{B(\alpha_{1,3})}(z_3)^{\epsilon_1}$ since $\alpha_{1,3} \circ \epsilon_1 | \epsilon_1 \sim \text{Binomial}(n = \epsilon_1, p = \alpha_{1,3})$. Applying the above rule recursively with the law of expectation, we can obtain Theorem 1. The overall proof is given in Appendix A.1.

Theorem 1 reveals the inherent relationship between the causal structure and the close-formed solution of PGF. To see this connection more concretely, we further develop the following theorem:

**Theorem 2.** *Given a random vector* $\mathbf{X} = [X_1, ..., X_d]^T$ *following PB-SCM, the PGF of* $P(\mathbf{X})$ *can be expressed by:*

$$G_{\mathbf{X}}(\mathbf{z}) = \prod_{i \in [d]} \exp\{\mu_i \times (T_{X_i}(1) - 1)\}, \tag{4}$$

*where* $T_{X_i}(1) = z_i \sum_{\mathbf{s} \in \{0,1\}^{|Ch(i)|}} \prod_{j \in Ch(i)} \alpha_{i,j}^{s_j} T_{X_j}(s_j), T_{X_i}(0) = 1$ *and* $\alpha_{i,j}^{s_j} = \alpha_{i,j}$ *if* $s_j = 1$ *and* $\alpha_{i,j}^{s_j} = 1 - \alpha_{i,j}$ *if* $s_j = 0$.

Theorem 2 provides a more compact closed-form solution for the PGF of PB-SCM. Intuitively, it consists of different path information starting at different vertex. Taking the triangular structure in Fig. 3 as an example, the closed-form solution of PGF can be expressed as follows:

$$G_{\mathbf{X}}(\mathbf{z}) = \exp\Big\{\mu_1 \times [\underbrace{(1 - \alpha_{1,2})(1 - \alpha_{1,3})z_1}_{①} + \underbrace{\alpha_{1,2}(1 - \alpha_{1,3})z_1z_2 + \alpha_{1,2}\alpha_{2,3}(1 - \alpha_{1,3})z_1z_2z_3}_{②}$$

$$+ \underbrace{\alpha_{1,3}(1 - \alpha_{1,2})z_1z_3}_{③} + \underbrace{\alpha_{1,2}\alpha_{1,3}(1 - \alpha_{2,3})z_1z_2z_3 + \alpha_{1,2}\alpha_{2,3}\alpha_{1,3}z_1z_2z_3^2}_{④}] - \mu_1\Big\}$$

$$\times \exp\Big\{\mu_2 \times [\underbrace{(1 - \alpha_{2,3})z_2}_{⑤} + \underbrace{\alpha_{2,3}z_2z_3}_{⑥}] - \mu_2\Big\} \times \exp\{\mu_3 \times (\underbrace{z_3}_{⑦} - 1)\},$$

$$\tag{5}$$

which is the combination of three main terms $T_{X_1}(1)$, $T_{X_2}(1)$, and $T_{X_3}(1)$ given in Theorem 2. Here $T_{X_i}(1)$ contains all possible path information from $X_i$ to all its descendants while $T_{X_i}(0)$ denotes the vertex is 'close' which will not be reached at the current path. Theorem 2 shows that for each $T_{X_i}(1)$, it enumerates all possible combinations of 'open' and 'close' of the children of $X_i$ recursively. For example, in terms ③, we have $X_2$ 'close' and $X_3$ 'open' such that only $z_3$ appears in this term and we obtain $\alpha_{1,3}(1 - \alpha_{1,2})z_1z_3$. Formally, we provide the graphical implication of the closed-form solution of PGF as follows:

**Proposition 1** (Graphical implication of closed-form solution of PGF). *Given a random vector* $\mathbf{X} = [X_1, ..., X_d]^T$ *following PB-SCM, for any subgraph* $G(\mathbf{L}, \mathbf{E})$ *with the subset of vertices* $\mathbf{L} \subseteq \mathbf{V}$ *such that the* $i$-*th vertex is the root vertex in* $G(\mathbf{L}, \mathbf{E})$, *a component* $Cz_i \prod_{j \in L \setminus \{i\}} z_j^{p_j}$ *with constant* $C \neq 0$, *exponent* $p_j \in \mathbb{Z}^+$ *exits in* $T_{X_i}(1)$, *if and only if there exists a subgraph* $G(\mathbf{L}, \mathbf{E}')$ *with subset of the edges* $\mathbf{E}' \subseteq \mathbf{E}$ *such that for each* $j \in \mathbf{L} \setminus \{i\}$, *there are at least* $p_j$ *directed paths from* $X_i$ *to* $X_j$ *in the subgraph* $G(\mathbf{L}, \mathbf{E}')$.

Proposition 1 reveals the graphical implication of each term in the closed-form solution of PGF that each term is related to a number of directed paths in a certain subset graph. In particular, for the

highest order, in terms ④, $z_1 z_2 z_3^2$ indicates that there are two paths from $X_1$ to $X_3$ and one path from $X_1$ to $X_2$ in this triangular structure, which implies that $X_1, X_2$ is the cause of $X_3$ offering a way to identify the causal direction by detecting whether $z_3^2$ exists. In fact, the cumulant-based method is exactly the method that identifies the causal direction by detecting the highest order of $z_i$ using cumulant (Qiao et al. [2024b]).

However, one drawback of detecting the highest order is that the highest order of $z_i$ does not always contain full identification for PB-SCM. For example, in Fig. 1 (a), since $z_1 z_2 z_4^2, z_2 z_3 z_4^2$ has the order of 2 while $z_1 z_2, z_2 z_3$ only has an order of 1, we can only conclude the causal direction from $X_1, X_2, X_3$ to $X_4$ but not causal relation among $X_1, X_2, X_3$ using merely the highest order. Fortunately, the causal structure $X_1 \rightarrow X_2 \leftarrow X_3$ remains identifiable since the term $z_1 z_2 z_3$ does not exist in the PGF. This indicates that there is no directed path starting from any vertex that passes through all three vertices $X_1, X_2, X_3$. In this case, we can conclude that it must be the collider structure and allows us to identify the causal structure by detecting whether such a term exists. Another drawback of detecting the highest order term is that it requires multiple derivatives of PGF or complex construction with higher order cumulant, hindering the analysis of the identifiability. In this work, benefiting the local property of the PGF, we explore a consistent way to identify the causal structure while only lower-order information is required.

### 4.3 Local Probability Generating Function

One of the attractive properties of PGF is the ability to isolate and examine some specific components by strategically setting the other variables $z$ approach zero within the probability generating function, which allows us to investigate the local structure and devise a practical structure learning algorithm. Formally, we define the following local PGF:

**Definition 3** (Local Probability Generating Function). *Given random vector $\mathbf{X} = [X_1, ..., X_d]^T$ following PB-SCM and its PGF $G_\mathbf{X}(\mathbf{z})$. The local PGF of vertices $\mathbf{L} \subset [d]$ is the pointwise limit of $G_\mathbf{X}(\mathbf{z})$ as the $\mathbf{z}_{\backslash \mathbf{L}}$ approach $\mathbf{0}$, $\mathbf{z}_{\backslash \mathbf{L}} = \{z_i | i \in [d] \setminus \mathbf{L}\}$, denoted as: $G_\mathbf{X}^\mathbf{L}(\mathbf{z}) = \lim_{\mathbf{z}_{\backslash \mathbf{L}} \rightarrow \mathbf{0}} G_\mathbf{X}(\mathbf{z})$.*

The local PGF can be constructed using the original PGF by setting the limit of $z$. By this, similarly to Theorem 2, we have the following closed-form solution for the local PGF:

**Theorem 3.** *Given a random vector $\mathbf{X} = [X_1, ..., X_n]^T$ following PB-SCM and its PGF $G_\mathbf{X}(\mathbf{z})$. For a subset $\mathbf{L} \subset [d]$ of the set of vertices, and the set of the children of vertex $i$ within $\mathbf{L}$, denoted by $Ch_\mathbf{L}(i) = Ch(i) \cap \mathbf{L}$, the local PGF of $\mathbf{L}$ can be expressed by:*

$$G_\mathbf{X}^\mathbf{L}(\mathbf{z}) = \prod_{i \in \mathbf{L}} \exp\left\{\mu_i \times \left(T_{X_i}^\mathbf{L}(1) - 1\right)\right\} \prod_{i \in [d] \setminus \mathbf{L}} \exp\{-\mu_i\}, \qquad (6)$$

*where $T_{X_i}^\mathbf{L}(1) = z_i \sum_{\mathbf{s} \in \{0,1\}^{|Ch_\mathbf{L}(i)|}} \prod_{j \in Ch_\mathbf{L}(i)} \alpha_{i,j}^{s_j} T_{X_j}^\mathbf{L}(s_j) \prod_{j \in Ch(i) \setminus Ch_\mathbf{L}(i)} (1 - \alpha_{i,j})$ and $T_{X_i}^\mathbf{L}(0) = 1$.*

Taking Fig. 3 as an example, the local PGF of vertices $\mathbf{L} = \{X_2, X_3\}$ is $G_\mathbf{X}^{\{X_2, X_3\}} = \exp\{-\mu_1\} \exp\{\mu_2[(1 - \alpha_{2,3})z_2 + \alpha_{2,3}z_2 z_3] - \mu_2\} \exp\{\mu_3 z_3 - \mu_3\}$. Such local probability generating function allow us to analysis and to identify the local structure, e.g., the term $z_2 z_3$ in the local PGF $G_\mathbf{X}^{\{X_2, X_3\}}$ represent the adjacent relation between $X_2$ and $X_3$.

In the following sections, we will further explore the identifiability of several specific structures that serve as fundamental local structures within a graph.

### 4.4 Identifiability

In this section, we address the identifiability of the PB-SCM using the PGF. Our focus is on identifying three fundamental local structures to reconstruct the causal graph from causal skeleton to causal direction: (i) the adjacent relation, (ii) the local triangular structure, and (iii) the local collider structure.

**Adjacent relation.** We first address the identifiability of the adjacent relation, which is involved in identifying the causal skeleton. For each pair of vertices $X_i, X_j \in \mathbf{X}$, the component $z_i z_j$ appears in the PGF if and only if $X_i$ and $X_j$ are adjacent, indicating that there exists a path either from $X_i$ to $X_j$ or from $X_j$ to $X_i$. Therefore, we can detect the adjacency relation by testing whether the second

partial derivative $\frac{\partial^2 \log G_{\mathbf{X}}^{\{i,j\}}(\mathbf{z})}{\partial z_i \partial z_j}$ equal to zero in a local structure. In order to construct a hypothesis test for such a condition, we formulate the condition as a rank condition. Formally, the adjacent relation can be identified using the following theorem:

**Theorem 4** (Identifiability of adjacent vertices). *Let $X_i, X_j \in \mathbf{X}$ be two arbitrary vertices with the corresponding local PGF $G_{\mathbf{X}}^{\{i,j\}}(\mathbf{z})$. Define the matrix $\mathbf{A}^{\{i,j\}} = \begin{pmatrix} G_{\mathbf{X}}^{\{i,j\}}(\mathbf{z}) & \frac{\partial G_{\mathbf{X}}^{\{i,j\}}(\mathbf{z})}{\partial z_i} \\ \frac{\partial G_{\mathbf{X}}^{\{i,j\}}(\mathbf{z})}{\partial z_j} & \frac{\partial^2 G_{\mathbf{X}}^{\{i,j\}}(\mathbf{z})}{\partial z_i \partial z_j} \end{pmatrix}$ with $z_i, z_j$ approach 1, the condition $\mathrm{Rank}(\mathbf{A}^{\{i,j\}}) = 1$ if and only if $X_i$ is non-adjacent to $X_j$.*

Intuitively, Theorem 4 identify the adjacent relation by detecting the existence of $z_i z_j$ in the logarithm of the PGF, i.e., the $T_{X_i}(i)$ inside the exponential function. By this, the identification of the causal skeleton is given. Next, we show the identifiability of a local triangular structure.

**Triangular structure.** For any three vertices $X_i, X_j, X_k \in \mathbf{X}$ forming a triangular structure. Such a structure must have one and only one vertex with an in-degree of 2. This asymmetry is captured by the second-order derivative of local PGF, e.g., $z_i z_j z_k^2$ where $X_k$ has an in-degree of 2. By this, the causal direction $X_i \rightarrow X_k$ and $X_j \rightarrow X_k$ in the local triangular structure is identifiable as follows:

**Theorem 5** (Identifiability of local triangular structure). *Let $X_i, X_j, X_k \in \mathbf{X}$ form a triangular structure with the corresponding local PGF $G_{\mathbf{X}}^{\{i,j,k\}}(\mathbf{z})$. Define the matrix $\mathbf{B}^{\{i,j,k\}} = \begin{pmatrix} G_{\mathbf{X}}^{\{i,j,k\}}(\mathbf{z}) & \frac{\partial G_{\mathbf{X}}^{\{i,j,k\}}(\mathbf{z})}{\partial z_k} \\ \frac{\partial G_{\mathbf{X}}^{\{i,j,k\}}(\mathbf{z})}{\partial z_k} & \frac{\partial^2 G_{\mathbf{X}}^{\{i,j,k\}}(\mathbf{z})}{\partial z_k^2} \end{pmatrix}$ with $z_i, z_j, z_k$ approach 1, the condition $\mathrm{Rank}(\mathbf{B}^{\{i,j,k\}}) = 2$ if and only if $X_k$ is the vertex with an in-degree of 2 in this triangular structure, i.e., $X_i \overset{\frown}{\rightarrow} X_k \leftarrow X_j$.*

Note that the causal direction in the triangular structure is not fully identifiable because it does not exhibit asymmetry and can always construct an equivalent PGF in the reversed direction which is also discussed in Qiao et al. [2024b]. Next, we consider the identifiability of the local collider structure, which is also referred to as the unshielded collider structure.

**Local collider structure.** Given three adjacent vertices $X_i - X_j - X_k$. If they form a collider structure $X_i \rightarrow X_j \leftarrow X_k$, the corresponding pattern in the closed-form solution of PGF is $z_i z_j + z_j z_k$ but no $z_i z_j z_k$, reflecting the absence of non-block path among these three vertices. Thus we have $\frac{\partial^2 \log G_{\mathbf{X}}^{\{i,j,k\}}(\mathbf{z})}{\partial z_i \partial z_k} = 0$. Similarly, we can construct a rank condition for identifying such local collider structure using the following theorem:

**Theorem 6** (Identifiability of local collider structure). *Let $X_i, X_j, X_k \in \mathbf{X}$ be three adjacent vertices with the corresponding local PGF $G_{\mathbf{X}}^{\{i,j,k\}}(\mathbf{z})$. Define the matrix $\mathbf{C}^{\{i,j,k\}} = \begin{pmatrix} G_{\mathbf{X}}^{\{i,j,k\}}(\mathbf{z}) & \frac{\partial G_{\mathbf{X}}^{\{i,j,k\}}(\mathbf{z})}{\partial z_i} \\ \frac{\partial G_{\mathbf{X}}^{\{i,j,k\}}(\mathbf{z})}{\partial z_k} & \frac{\partial^2 G_{\mathbf{X}}^{\{i,j,k\}}(\mathbf{z})}{\partial z_i \partial z_k} \end{pmatrix}$ with $z_i, z_j, z_k$ approach 1, the condition $\mathrm{Rank}(\mathbf{C}^{\{i,j,k\}}) = 1$ if and only if the vertices $X_i, X_j, X_k$ form the structure $X_i \rightarrow X_j \leftarrow X_k$.*

Combining the identifiability in Theorem 5 and Theorem 6 of the local structure, we conclude the following graphical implication of the identification of PB-SCM.

**Theorem 7** (Graphical implication of identifiability). *Given a pair of adjacent vertices $X_i, X_j \in \mathbf{X}$ following PB-SCM, the causal direction of $X_i \rightarrow X_j$ is identifiable if there exists a vertex $X_k \in \mathbf{X} \setminus \{X_i, X_j\}$ such that $X_k \rightarrow X_j$.*

### 4.5 Learning Causal Structure of PB-SCM Using Probability Generating Function

In this section, we propose a practical algorithm for learning the causal structure of PB-SCM using PGF. Note that the probability generating function can be estimated by employing the empirical probability generating function (Nakamura and Pérez-Abreu [1993]). The algorithm is given in Alg. 1. Our algorithm involves two steps: learning the skeleton of DAG $G$ using the result developed in Theorem 4 and inferring the causal direction using the results developed in Theorem 5 and 6.

**Learning Causal Skeleton.** To learn the causal skeleton, following the Theorem 4, we construct the matrix $\mathbf{A}^{\{i,j\}}$ for each pair of vertices $X_i, X_j \in \mathbf{X}$, and assess its rank to determine adjacency

between the vertices (Line 2-4). Notably, learning the causal skeleton is efficient because each pair of vertices requires only one examination.

**Learning Causal Direction.** Given the skeleton, we orient the causal direction following the Theorem 5 and Theorem 6. Our initial focus is on orienting within triangular structures. For each triangular structure, we enumerate the matrix $\mathbf{B}^{\{i,j,k\}}$ to test whether $X_k$ has indegree of two based on Theorem 5 and orient $X_i \to X_k$ and $X_j \to X_k$ if detected (Line 5-7). After the orientation in the triangular structure, we focus on orienting the remaining undirected edges following Theorem 6. Specifically, for each undirected edge $X_i - X_j$, we first consider testing the causal direction in a pattern like $X_i - X_j \leftarrow X_k$. Then, we consider testing the causal direction in pattern $X_i - X_j - X_k$. Such a test is conducted by constructing the matrix $\mathbf{C}^{\{i,j,k\}}$ and to test whether the rank is 1, and then orient $X_i \to X_j \leftarrow X_k$ if detected (Line 8-10).

**Rank Test.** To test the rank of $\mathbf{A}^{\{i,j\}}$, $\mathbf{B}^{\{i,j,k\}}$ and $\mathbf{C}^{\{i,j,k\}}$, we employ a rank test by testing whether the second (minimum) eigenvalue $\lambda_2$ is zero, i.e., $H_0 : \lambda_2 = 0$ $v.s.$ $H_1 : \lambda_2 \neq 0$. Since the trace of the matrix converges to a normal distribution based on the central limit theorem, and thus the sum of eigenvalues is also Gaussian. Thus, if $H_0$ is true, we approximate the eigenvalue as a normal distribution with zero mean. We further employ bootstrap method (Efron and Tibshirani [1994]) to estimate the variance of such distribution by calculating the bootstrapping statistic $\lambda_2^+$ from $N$ resampling dataset $\mathcal{D}^+ \in \{\mathcal{D}_i^+ | \mathcal{D}_{i=1,..,N}^+ \subset \mathcal{D}, \}$ and estimate the variance of $\lambda_2^+$. Building on this, the p-value of $\lambda_2$ from the original dataset can be obtained.

---

**Algorithm 1:** Causal Discovery for PB-SCM Using Probability Generating Function

---

**Input:** Data set $\mathcal{D}$
**Output:** Causal Graph $G$
1   $G \leftarrow empty\ graph$;
    // Learning Causal Skeleton
2   **for** *each pair of vertices $i, j$ in $G$* **do**
3     **if** *Rank condition* $\mathrm{Rank}(\mathbf{A}^{\{i,j\}}) \neq 1$ **then**
4       Add undirected edge "$X_i - X_j$" in $G$ based on Theorem 4.

    // Learning Causal Direction
5   **for** *each triangular $X_i, X_j, X_k \in G$* **do**
6     **if** *Rank condition* $\mathrm{Rank}(\mathbf{B}^{\{i,j,k\}}) \neq 1$ **then**
7       Orient "$X_i \to X_k \leftarrow X_j$" in $G$ based on Theorem 5.

8   **for** *each structure $X_i - X_j \leftarrow X_k \in G$ or $X_i - X_j - X_k \in G$* **do**
9     **if** $\mathrm{Rank}(\mathbf{C}^{\{i,j,k\}}) = 1$ **then**
10      Orient "$X_i \to X_j \leftarrow X_k$" in $G$ based on Theorem 6.

11 **Return** $G$

---

**Complexity Analysis.** In the step of learning skeleton, we determine whether each pair of vertices is adjacent by testing the rank of the matrix following Theorem 4, and hence the complexity of skeleton learning is $\mathcal{O}(\frac{1}{2}d(d-1))$. In the step of learning causal direction, we consider the complete graph, where there are $\binom{d}{3} = \frac{d(d-1)(d-2)}{6}$ triangular structures, and for each triangular structures, we test the rank of the matrix for three vertices following Theorem 5, and hence the complexity is $\mathcal{O}(\frac{d(d-1)(d-2)}{2})$. Similarly, in orienting collider structures from a complete graph, we choose three adjacent vertices and test the rank once following Theorem 6, hence the complexity is $\mathcal{O}(\frac{d(d-1)(d-2)}{6})$. Therefore, the total complexity is $\mathcal{O}(\frac{1}{2}d(d-1) + \frac{2d(d-1)(d-2)}{3})$.

## 5   Experiment

### 5.1   Synthetic Experiments

In this section, we test our proposed method on synthetic data. We first conduct control experiments on synthetic data to evaluate the sensitivity of our method to sample size and different indegree rates. Following this, we present case studies involving 3, 4, and 5 vertices to further illustrate the identifiability of our approach. The baseline methods include the cumulant-based method (Cumulant) (Qiao et al. [2024b]), OCD (Ni and Mallick [2022]), PC (Spirtes et al. [2000]), GES (Chickering [2002]).

**Sensitivity Experiment**   In the sensitivity experiment, we synthesize data with fixed parameters while traversing the target parameter. The default settings are as follows, sample size=30000, number of vertices=10, indegree rate=3.0, range of causal coefficient $\alpha_{i,j} \in [0.1, 0.3]$, range of the mean of Poisson noise $\mu_i \in [0.05, 0.15]$. Each simulation is repeated 15 times.

Table 1: Sensitivity to Avg. In-degree Rate.

|  | F1↑ | | | | SHD↓ | | | |
|---|---|---|---|---|---|---|---|---|
| Avg. In-degree | 2.0 | 2.5 | 3.0 | 3.5 | 2.0 | 2.5 | 3.0 | 3.5 |
| Ours | **0.74 ± 0.05** | **0.81 ± 0.07** | **0.86 ± 0.03** | **0.89 ± 0.04** | **9.67 ± 1.99** | **8.33 ± 2.28** | **6.40 ± 1.68** | **5.27 ± 1.39** |
| Cumulant | 0.73 ± 0.03 | 0.77 ± 0.02 | 0.80 ± 0.04 | 0.83 ± 0.03 | 13.40 ± 1.28 | 14.10 ± 1.51 | 13.00 ± 2.37 | 13.20 ± 2.23 |
| PC | 0.60 ± 0.17 | 0.62 ± 0.11 | 0.54 ± 0.12 | 0.60 ± 0.12 | 9.90 ± 3.45 | 11.80 ± 2.48 | 15.90 ± 3.91 | 16.10 ± 3.21 |
| GES | 0.48 ± 0.14 | 0.48 ± 0.11 | 0.41 ± 0.11 | 0.37 ± 0.10 | 14.90 ± 4.48 | 19.50 ± 4.61 | 25.90 ± 4.18 | 30.5 ± 4.06 |
| OCD | 0.23 ± 0.22 | 0.27 ± 0.23 | 0.28 ± 0.16 | 0.37 ± 0.14 | 16.10 ± 3.70 | 19.40 ± 5.50 | 23.60 ± 4.62 | 24.30 ± 4.67 |

Table 2: Sensitivity to Sample Size.

|  | F1↑ | | | | SHD↓ | | | |
|---|---|---|---|---|---|---|---|---|
| Sample Size | 5000 | 15000 | 30000 | 50000 | 5000 | 15000 | 30000 | 50000 |
| Ours | **0.75 ± 0.09** | **0.82 ± 0.04** | **0.86 ± 0.03** | **0.87 ± 0.04** | **11.50 ± 3.34** | **9.27 ± 2.37** | **6.40 ± 1.68** | **5.87 ± 1.25** |
| Cumulant | 0.72 ± 0.04 | 0.78 ± 0.02 | 0.80 ± 0.04 | 0.80 ± 0.03 | 19.90 ± 3.35 | 15.00 ± 1.41 | 13.00 ± 2.49 | 13.60 ± 2.63 |
| PC | 0.45 ± 0.11 | 0.54 ± 0.11 | 0.54 ± 0.13 | 0.66 ± 0.09 | 19.50 ± 4.30 | 15.70 ± 3.77 | 15.90 ± 4.12 | 13.00 ± 2.79 |
| GES | 0.39 ± 0.10 | 0.44 ± 0.20 | 0.41 ± 0.11 | 0.43 ± 0.22 | 23.70 ± 4.14 | 22.70 ± 7.85 | 25.90 ± 4.41 | 24.10 ± 8.84 |
| OCD | 0.30 ± 0.12 | 0.35 ± 0.18 | 0.28 ± 0.16 | 0.38 ± 0.20 | 21.90 ± 3.35 | 20.80 ± 4.57 | 23.60 ± 4.62 | 20.90 ± 5.74 |

In the control experiments on the average in-degree given in Table 1, as the average in-degree controls the sparse of causal structure, the higher the in-degree rate, the less sparse in causal structure leading to a decrease of performance of the baseline methods, PC, GES and OCD. In contrast, our method and Cumulant show improved performance as they benefit from the denser structure which provides more identifiable structures. Additionally, our method outperforms Cumulant by overcoming identifiability limitations through leveraging PGF, demonstrating better performance.

In the control experiments on sample size presented in Table 2, our method's performance improves with increasing sample sizes, consistently outperforming all baseline methods. Furthermore, it surpasses Cumulant under the same conditions due to its efficient identification of directions in local structures without relying on high-order statistics, thus enhancing accuracy.

**Case Study**    To demonstrate the identifiability of the proposed PGF-based method, we present case studies using causal graphs with 3, 4, and 5 vertices. The results of our method and the baseline methods are summarized in Table 3.

Table 3: Case studies of causal graphs with in total 3, 4, and 5 vertices, respectively. Red undirected edges indicate that adjacency has been learned but the direction cannot be determined, while red directed edges indicate incorrectly learned directions.

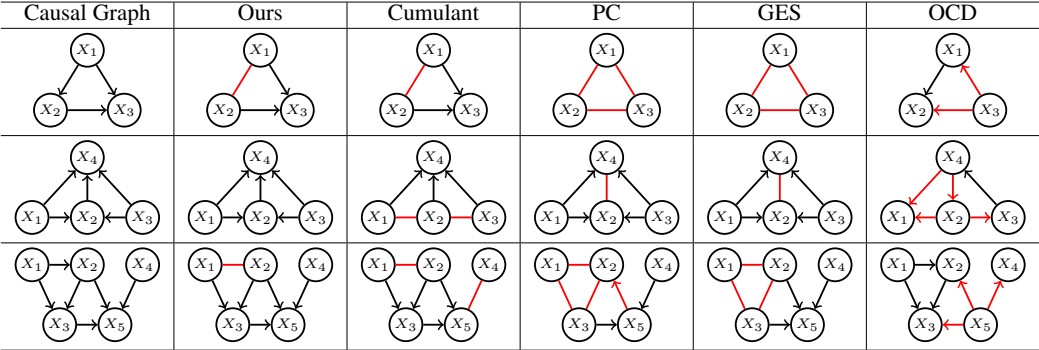

Generally, our PGF-based method successfully identifies almost all the edges, except for $X_1 \rightarrow X_2$ in the graphs with 3 and 5 vertices. This is because there is neither a vertex with an indegree of 2 to provide causal asymmetry, nor an additional vertex to form the local collider structure, which aligns with our theoretical results. Notably, the cumulant-based method fails to identify the structure $X_1 \rightarrow X_2 \leftarrow X_3$ in the 4-vertex graph and the edge $X_4 \rightarrow X_5$ in the 5-vertex graph, as it cannot leverage low-order information. This limitation, however, is successfully addressed by the PGF-based method, which can utilize such information effectively. Regarding other baseline methods, PC and GES encounter difficulties in identifying the Markov equivalent class, while OCD illustrates identifiability issues in PB-SCM.

## 5.2 Real World Experiments

In this section, we evaluate the performance of our proposed method on two real-world datasets to assess its effectiveness in realistic scenarios.

**Football Events Dataset** We test the proposed method on a real-world football events dataset[1], which includes 941,009 events from 9,074 games across Europe. The experiment focused on analyzing the causal relationships between specific events such as Foul, Yellow card, Second yellow card, Red card, and Substitution, as depicted in Fig. 4. The goal is to identify causal relationships from the counts of these events.

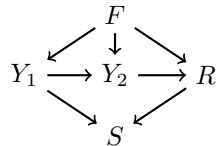

Figure 4: Football Event Graph ($F$: Foul, $Y_1$: Yellow card, $Y_2$: Second yellow card, $R$: Red card, $S$: Substitution)

In detail, we orientate in the local triangular structures $Y_1 - Y_2 - F$, $Y_2 - R - F$, and the local collider structure $Y_1 - S - R$. As a result, Our method successfully identifies adjacent vertex $Foul -$ First Yellow Card and other six causal directions, which is consistent with our theoretical result.

**Shopping Mall Paid Search Campaign Count Data** We also evaluate our method using a shopping mall paid search campaign dataset[2]. This dataset contains information from a five-month paid search campaign for a U.S. shopping mall, spanning from July to November 2021. In this analysis, we focus on the count variables *Impressions*, *Clicks*, and *Conversions*, which represent fundamental count data in e-commerce scenarios. These variables exhibit the following causal relationships: *Impressions* → *Clicks*, *Clicks* → *Conversions*, and *Impressions* → *Conversions*, forming a triangular structure.

Our experimental results demonstrate that the proposed method successfully identifies the adjacent relation between *Impressions* and *Clicks*, and the other two causal directions. These results align with our theoretical findings, suggesting that the method has the potential to be applied to real-world scenarios involving count data.

## 6 Conclusion

In this work, we investigate the identifiability of the Poisson branching structural causal model using the probability generating function. We derive a nontrivial closed-form solution for the PGF of PB-SCM and further establish the connection between the closed-form solution of PGF to the causal structure, showing that the closed-form solution of PGF encompasses the path information with various subgraphs. With this connection, we employ the local property of PGF and propose a simple yet efficient way to identify the local causal structure of PB-SCM by constructing a matrix with rank test. By this, we provide a practical algorithm and a hypothesis test approach for testing the causal structure and verifying the effectiveness of the algorithm via synthesis and real-world data. The proposed theoretical results take a meaningful step in understanding the causal mechanism and completing the identifiability result of PB-SCM.

The main limitation of this work is that the explicit estimation of the probability generating function does not scale well to the number of nodes. Developing a sample-efficient estimating method could be a promising direction. In addition, causal faithfulness and sufficient assumption may restrict the usage of this work in a border scenario, which needs to be processed with extra detection steps to eliminate the effect.

## 7 Acknowledgments

This research was supported in part by National Key R&D Program of China (2021ZD0111501), National Science Fund for Excellent Young Scholars (62122022), Natural Science Foundation of China (61876043, 61976052, 62406080), the major key project of PCL (PCL2021A12). We sincerely appreciate the comments from anonymous reviewers, which greatly helped to improve the paper.

---

[1]https://www.kaggle.com/datasets/secareanualin/football-events
[2]https://www.kaggle.com/datasets/marceaxl82/shopping-mall-paid-search-campaign-dataset

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

# Appendix

## Table of Contents

## A Proof of Closed-Form Solution for PGF of PB-SCM

### A.1 Proof of Theorem 1

**Theorem 1** (Closed form of PGF within PB-SCM). *Given a random vector* $\mathbf{X} = [X_1, ..., X_n]^T$ *following PB-SCM, let* $\mathbf{z}_{(j)} = \{z_l | l \in Des(j) \cup \{j\}\}$, *the PGF of* $P(\mathbf{X})$ *is given by* $G_{\mathbf{X}}(\mathbf{z}) = \prod_{i \in [d]} G_{\epsilon_i}\left(z_i \times \prod_{j \in Ch(i)} G_{i,j}(\mathbf{z}_{(j)})\right)$, *where*

$$G_{i,j}(\mathbf{z}_{(j)}) = \begin{cases} G_{B(\alpha_{i,j})}\left(z_j \times \prod_{k \in Ch(j)} G_{j,k}(\mathbf{z}_{(k)})\right) & , Ch(j) \neq \emptyset \\ G_{B(\alpha_{i,j})}(z_j) & , Otherwise \end{cases}, \tag{A.1}$$

*in which* $G_{\epsilon_i}(\cdot)$ *is the PGF of Poisson noise* $\epsilon_i$ *and* $G_{B(\alpha_{i,j})}(\cdot)$ *is the PGF of Bernoulli distribution with parameter* $\alpha_{i,j}$.

#### A.1.1 An Illustrative Example for Deriving the Closed Form

Before formal proof, we first provide an intuition of proof through a detailed example. Deriving the closed form of a multivariate PGF involves decomposing the expectation using the law of total expectation. This ensures that each decomposed expectation involves only one random variable, each of which has a closed-form PGF, as most univariate PGFs admit a closed form. We first introduce some closed forms of univariate PGFs used in the proof. Specifically, the univariate PGFs of the Poisson and Bernoulli distributions are given by: $G_{\epsilon}(z) = \mathbb{E}[z^{\epsilon}] = \exp\{\mu(z-1)\}$ when $\epsilon \sim \text{Poisson}(\mu)$ and $G_{B(\alpha)}(z) = \mathbb{E}[z^x] = 1 - \alpha + \alpha z$ where $x \sim B(\alpha)$, the Bernoulli distribution with parameter $\alpha$.

Consider the triangular structure in Fig. 5, the corresponding PGF is expressed as follows:

$$G_{\mathbf{X}}(\mathbf{z}) = \mathbb{E}\left[z_1^{X_1} z_2^{X_2} z_3^{X_3}\right] = \mathbb{E}\left[z_1^{\epsilon_1} z_2^{\alpha_{1,2} \circ \epsilon_1 + \epsilon_2} z_3^{\alpha_{2,3} \circ \alpha_{1,2} \circ \epsilon_1 + \alpha_{2,3} \circ \epsilon_2 + \epsilon_3}\right]. \tag{A.2}$$

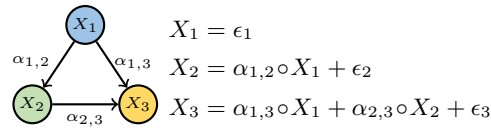

$$X_1 = \epsilon_1$$
$$X_2 = \alpha_{1,2} \circ X_1 + \epsilon_2$$
$$X_3 = \alpha_{1,3} \circ X_1 + \alpha_{2,3} \circ X_2 + \epsilon_3$$

Figure 5: Example triangular structure.

Since PB-SCM is a kind of additive noise model, we can represent each variable by the noise of ancestors with the corresponding coefficient along with the causal path. Next, since noises are independent of each other, by rearranging the components that share the same noise, we can further decompose the expectation:

$$G_{\mathbf{X}}(\mathbf{z}) = \mathbb{E}\left[z_1^{\epsilon_1} z_2^{\alpha_{1,2} \circ \epsilon_1} z_3^{\alpha_{1,3} \circ \epsilon_1} z_3^{\alpha_{2,3} \circ \alpha_{1,2} \circ \epsilon_1}\right] \mathbb{E}\left[z_2^{\epsilon_2} z_3^{\alpha_{2,3} \circ \epsilon_2}\right] \mathbb{E}\left[z_3^{\epsilon_3}\right]. \tag{A.3}$$

This form reveals the underlying causal mechanism, where the power of $z_j$ in the expectation involving $z_i^{\epsilon_i}$ is the noise component from $X_i$ to $X_j$ with the corresponding path coefficients. That is, each expectation captures the noise's potential influences, along with their respective causal paths.

To analyze this expression more closely, we focus on the first expectation involving $z_1^{\epsilon_1}$ in Eq. A.3. By applying the law of total expectation, we decompose this expectation, ensuring that each decomposed inner expectation depends on only one variable:

$$\begin{aligned}
\mathbb{E}\left[z_1^{\epsilon_1} z_2^{\alpha_{1,2} \circ \epsilon_1} z_3^{\alpha_{1,3} \circ \epsilon_1} z_3^{\alpha_{2,3} \circ \alpha_{1,2} \circ \epsilon_1}\right] &= \mathbb{E}\left[z_1^{\epsilon_1} \mathbb{E}\left[z_3^{\alpha_{1,3} \circ \epsilon_1} | \epsilon_1\right] \mathbb{E}\left[z_2^{\alpha_{1,2} \circ \epsilon_1} z_3^{\alpha_{2,3} \circ \alpha_{1,2} \circ \epsilon_1} | \epsilon_1\right]\right] \\
&= \mathbb{E}\left[z_1^{\epsilon_1} \mathbb{E}\left[z_3^{\alpha_{1,3} \circ \epsilon_1} | \epsilon_1\right] \mathbb{E}\left[z_2^{\alpha_{1,2} \circ \epsilon_1} \mathbb{E}\left[z_3^{\alpha_{2,3} \circ \alpha_{1,2} \circ \epsilon_1} | \alpha_{1,2} \circ \epsilon_1\right] | \epsilon_1\right]\right].
\end{aligned} \tag{A.4}$$

An immediate observation from Eq. A.4 is that, to establish conditional independence and decompose the expectation, we condition the variables following the causal order ($\epsilon_1$ corresponds to $X_1$ and $\alpha_{1,2} \circ \epsilon_1$ corresponds to $X_2$), thereby encapsulating the graph structure within the PGF.

Next, each decomposed expectation involves only one variable, e.g., $\mathbb{E}\left[z_3^{\alpha_{1,3} \circ \epsilon_1} | \epsilon_1\right]$ is the PGF of $\alpha_{1,3} \circ \epsilon_1 | \epsilon_1 \sim \text{Binomial}(n = X, p = \alpha)$, has a closed form PGF $G_{B(\alpha_{1,3})}(z_3)^{\epsilon_1}$. Similar to other expectations, we can finally obtain the closed form of $G_{\mathbf{X}}(\mathbf{z})$:

$$\begin{aligned}
G_{\mathbf{X}}(\mathbf{z}) &= \mathbb{E}\left[[z_1 \times G_{B(\alpha_{1,3})}(z_3)]^{\epsilon_1} \times \mathbb{E}\left[[z_2 \times G_{B(\alpha_{2,3})}(z_3)]^{\alpha_{1,2} \circ \epsilon_1} | \epsilon_1\right]\right] \\
&= \mathbb{E}\left[[z_1 \times G_{B(\alpha_{1,3})}(z_3) \times G_{B(\alpha_{1,2})}(z_2 \times G_{B(\alpha_{2,3})}(z_3))]^{\epsilon_1}\right] \\
&= G_{\epsilon_1}(z_1 \times G_{B(\alpha_{1,3})}(z_3) \times G_{B(\alpha_{1,2})}(z_2 \times G_{B(\alpha_{2,3})}(z_3))).
\end{aligned} \tag{A.5}$$

Note that, $\mathbb{E}\left[[z_2 \times G_{B(\alpha_{2,3})}(z_3)]^{\alpha_{1,2} \circ \epsilon_1} | \epsilon_1\right]$ is a PGF of $\alpha_{1,2} \circ \epsilon_1 | \epsilon_1$ where the input is $z_2 \times G_{B(\alpha_{2,3})}(z_3)$. Similar to this, we have:

$$\mathbb{E}\left[z_2^{\epsilon_2} z_3^{\alpha_{2,3} \circ \epsilon_2}\right] = \mathbb{E}\left[z_2^{\epsilon_2} \mathbb{E}\left[z_3^{\alpha_{2,3} \circ \epsilon_2} | \epsilon_2\right]\right] = \mathbb{E}\left[z_2^{\epsilon_2} G_{B(\alpha_{2,3})}(z_3)^{\epsilon_2}\right] = G_{\epsilon_2}(z_2 G_{B(\alpha_{2,3})}(z_3)), \tag{A.6}$$

and $\mathbb{E}\left[z_3^{\epsilon_3}\right] = G_{\epsilon_3}(z_3)$. Then the close form of PGF of Fig. 5 is as follows:

$$\begin{aligned}
G_{\mathbf{X}}(\mathbf{z}) =& G_{\epsilon_1}(z_1 \times G_{B(\alpha_{1,3})}(z_3) \times G_{B(\alpha_{1,2})}(z_2 \times G_{B(\alpha_{2,3})}(z_3))) \\
&\times G_{\epsilon_2}(z_2 \times G_{B(\alpha_{2,3})}(z_3)) \times G_{\epsilon_3}(z_3).
\end{aligned} \tag{A.7}$$

Therefore, to derive the closed form of PGF, we need to apply the law of total expectation following the causal order, which constructs a recursive formula. To formalize this in a recursive formula, we further define some graphical concepts.

**Notation.** We use $\mathbf{P}^{i \rightsquigarrow j} = \left\{P_k^{i \rightsquigarrow j}\right\}_{k=1}^{|\mathbf{P}^{i \rightsquigarrow j}|}$ denotes the set of all directed paths from vertex $i$ to $j$, where $P_k^{i \rightsquigarrow j} = (i, k_1, k_2, ..., k_p, j)$, $p = |P_k^{i \rightsquigarrow j}| - 2$, denote the $k$-th directed path from vertex $i$ to $j$. For each directed path $P_k^{i \rightsquigarrow j}$, we use $A_k^{i \rightsquigarrow j} = (\alpha_{i,k_1}, \alpha_{k_1,k_2}, \ldots, \alpha_{k_p,j})$ denote the corresponding *coefficients sequence* of path $P_k^{i \rightsquigarrow j}$. We let $\mathbf{P}^{i \rightsquigarrow i} = \{P^{i \rightsquigarrow i}\}$ also be a valid directed path for simplicity. Besides, we use $A_k^{i \rightsquigarrow j} \circ X_i := \alpha_{k_p,j} \circ \cdots \circ \alpha_{k_1,k_2} \circ \alpha_{i,k_1} \circ X_i$ denote to perform a consecutive thinning operation on $X_i$ based on the path sequence.

### A.1.2  Formal Proof

Following the definitions, the PGF of joint distribution $P(X_1, ..., X_n)$ under PB-SCM is given by $G_{\mathbf{X}}(\mathbf{z}) = E[z_1^{X_1} \cdots z_n^{X_n}]$. Since PB-SCM is a kind of additive noise model, we can represent each variable by the noise of ancestors with the corresponding coefficient sequence along with the directed path:

$$X_i = \sum_{h \in An(i)} \sum_{p=1}^{|P^{h \rightsquigarrow i}|} A_p^{h \rightsquigarrow i} \circ \epsilon_h + \epsilon_i, \tag{A.8}$$

where $An(i)$ denotes the set of ancestors of vertex $i$.

Next, by rearranging the components that share the same noise, we can further decompose the expectation as noises are independent of each other:

$$\begin{aligned}
G_{\mathbf{X}}(\mathbf{z}) &= \mathbb{E}\left[ z_1^{\sum_{h \in An(1)} \sum_{p=1}^{|P^{h \rightsquigarrow 1}|} A_p^{h \rightsquigarrow 1} \circ \epsilon_h + \epsilon_1} \times \cdots \times z_n^{\sum_{h \in An(n)} \sum_{p=1}^{|P^{h \rightsquigarrow n}|} A_p^{h \rightsquigarrow n} \circ \epsilon_h + \epsilon_n} \right] \\
&= \mathbb{E}\left[ \prod_{i \in [n]} \left[ z_i^{\epsilon_i} \prod_{j \in \mathrm{Des}(i)} z_j^{\sum_{p=1}^{|P^{i \rightsquigarrow j}|} A_p^{i \rightsquigarrow j} \circ \epsilon_i} \right] \right] = \prod_{i \in [n]} \mathbb{E}\left[ z_i^{\epsilon_i} \prod_{j \in \mathrm{Des}(i)} \prod_{p=1}^{|P^{i \rightsquigarrow j}|} z_j^{A_p^{i \rightsquigarrow j} \circ \epsilon_i} \right],
\end{aligned} \tag{A.9}$$

where $\mathrm{Des}(i)$ is the descendent set of vertex $i$.

We start by introducing the following Lemmas.

**Lemma 1.** *For each* $\mathbb{E}\left[ z_i^{\epsilon_i} \prod_{j \in Des(i)} \prod_{p=1}^{|P^{i \rightsquigarrow j}|} z_j^{A_p^{i \rightsquigarrow j} \circ \epsilon_i} \right]$, *we have:*

$$\mathbb{E}\left[ z_i^{\epsilon_i} \prod_{j \in Des(i)} \prod_{p=1}^{|P^{i \rightsquigarrow j}|} z_j^{A_p^{i \rightsquigarrow j} \circ \epsilon_i} \right] = G_{\epsilon_i}\left( z_i \prod_{j \in Ch(i)} \mathbb{E}\left[ z_j^{\xi_n^{(\alpha_{i,j})}} \prod_{k \in Des(j)} \prod_{p=1}^{|P^{j \rightsquigarrow k}|} z_k^{A_p^{j \rightsquigarrow k} \circ \xi_n^{(\alpha_{i,j})}} \right] \right). \tag{A.10}$$

*Proof.* We first condition $\epsilon_i$ using the law of total expectation and obtain:

$$\begin{aligned}
\mathbb{E}\left[ z_i^{\epsilon_i} \prod_{j \in \mathrm{Des}(i)} \prod_{p=1}^{|P^{i \rightsquigarrow j}|} z_j^{A_p^{i \rightsquigarrow j} \circ \epsilon_i} \right] &= \mathbb{E}\left[ z_i^{\epsilon_i} \mathbb{E}\left[ \prod_{j \in \mathrm{Des}(i)} \prod_{p=1}^{|P^{i \rightsquigarrow j}|} z_j^{A_p^{i \rightsquigarrow j} \circ \epsilon_i} \Big| \epsilon_i \right] \right] \\
&= \mathbb{E}\left[ z_i^{\epsilon_i} \mathbb{E}\left[ \prod_{j \in Ch(i)} \left( z_j^{\alpha_{i,j} \circ \epsilon_i} \prod_{k \in Des(j)} \prod_{p=1}^{|P^{j \rightsquigarrow k}|} z_k^{A_p^{j \rightsquigarrow k} \circ \alpha_{i,j} \circ \epsilon_i} \right) \Big| \epsilon_i \right] \right]
\end{aligned} \tag{A.11}$$

Now, regarding the definition of thinning operation $\alpha_{i,j} \circ X = \sum_{n=1}^{X} \xi_n^{(\alpha_{i,j})}$, where $\xi_n^{(\alpha_{i,j})} \sim$ Bernoulli($\alpha$) are i.i.d. Bernoulli random variables, Eq. A.11 can be rewritten as follows:

$$\begin{aligned}
\mathbb{E}\left[ z_i^{\epsilon_i} \prod_{j \in \mathrm{Des}(i)} \prod_{p=1}^{|P^{i \rightsquigarrow j}|} z_j^{A_p^{i \rightsquigarrow j} \circ \epsilon_i} \right] &= \mathbb{E}\left[ z_i^{\epsilon_i} \prod_{j \in Ch(i)} \mathbb{E}\left[ z_j^{\sum_{n=1}^{\epsilon_i} \xi_n^{(\alpha_{i,j})}} \prod_{k = Des(j)} \prod_{p=1}^{|P^{j \rightsquigarrow k}|} z_k^{\sum_{n=1}^{\epsilon_i} A_p^{j \rightsquigarrow k} \circ \xi_n^{(\alpha_{i,j})}} \Big| \epsilon_i \right] \right] \\
&= \mathbb{E}\left[ z_i^{\epsilon_i} \prod_{j \in Ch(i)} \mathbb{E}\left[ \prod_{n=1}^{\epsilon_i} z_j^{\xi_n^{(\alpha_{i,j})}} \prod_{k = Des(j)} \prod_{p=1}^{|P^{j \rightsquigarrow k}|} \prod_{n=1}^{\epsilon_i} z_k^{A_p^{j \rightsquigarrow k} \circ \xi_n^{(\alpha_{i,j})}} \Big| \epsilon_i \right] \right] \\
&= \mathbb{E}\left[ z_i^{\epsilon_i} \prod_{j \in Ch(i)} \mathbb{E}\left[ \prod_{n=1}^{\epsilon_i} \left( z_j^{\xi_n^{(\alpha_{i,j})}} \prod_{k = Des(j)} \prod_{p=1}^{|P^{j \rightsquigarrow k}|} z_k^{A_p^{j \rightsquigarrow k} \circ \xi_n^{(\alpha_{i,j})}} \right) \Big| \epsilon_i \right] \right].
\end{aligned} \tag{A.12}$$

Since we condition the $\epsilon_i$, which can be regarded as a constant in the expectation, we have:

$$
\mathbb{E}\left[z_i^{\epsilon_i}\prod_{j=\mathrm{Des}(i)}\prod_{p=1}^{|P^{i\rightsquigarrow j}|}z_j^{A_p^{i\rightsquigarrow j}\circ\epsilon_i}\right]=\mathbb{E}\left[z_i^{\epsilon_i}\prod_{j\in Ch(i)}\mathbb{E}\left[\left(z_j^{\xi_n^{(\alpha_{i,j})}}\prod_{k=\mathrm{Des}(j)}\prod_{p=1}^{|P^{j\rightsquigarrow k}|}z_k^{A_p^{j\rightsquigarrow k}\circ\xi_n^{(\alpha_{i,j})}}\right)^{\epsilon_i}\middle|\epsilon_i\right]\right]
$$

$$
=\mathbb{E}\left[z_i^{\epsilon_i}\prod_{j\in Ch(i)}\mathbb{E}\left[z_j^{\xi_n^{(\alpha_{i,j})}}\prod_{k=\mathrm{Des}(j)}\prod_{p=1}^{|P^{j\rightsquigarrow k}|}z_k^{A_p^{j\rightsquigarrow k}\circ\xi_n^{(\alpha_{i,j})}}\right]^{\epsilon_i}\right]
$$

$$
=\mathbb{E}\left[\left(z_i\prod_{j\in Ch(i)}\mathbb{E}\left[z_j^{\xi_n^{(\alpha_{i,j})}}\prod_{k=\mathrm{Des}(j)}\prod_{p=1}^{|P^{j\rightsquigarrow k}|}z_k^{A_p^{j\rightsquigarrow k}\circ\xi_n^{(\alpha_{i,j})}}\right]\right)^{\epsilon_i}\right],
$$
$$(A.13)$$

which is the PGF of $\epsilon_i$, with the input being $z_i\prod_{j\in Ch(i)}\mathbb{E}\left[z_j^{\xi_n^{(\alpha_{i,j})}}\prod_{k=\mathrm{Des}(j)}\prod_{p=1}^{|P^{j\rightsquigarrow k}|}z_k^{A_p^{j\rightsquigarrow k}\circ\xi_n^{(\alpha_{i,j})}}\right]$.
Therefore, we finally have:

$$
\mathbb{E}\left[z_i^{\epsilon_i}\prod_{j=\mathrm{Des}(i)}\prod_{p=1}^{|P^{i\rightsquigarrow j}|}z_j^{A_p^{i\rightsquigarrow j}\circ\epsilon_i}\right]=G_{\epsilon_i}\left(z_i\prod_{j\in Ch(i)}\mathbb{E}\left[z_j^{\xi_n^{(\alpha_{i,j})}}\prod_{k=\mathrm{Des}(j)}\prod_{p=1}^{|P^{j\rightsquigarrow k}|}z_k^{A_p^{j\rightsquigarrow k}\circ\xi_n^{(\alpha_{i,j})}}\right]\right).
$$
$$(A.14)$$

This completes the proof of Lemma 1. $\qquad\square$

**Lemma 2.** *Let* $G_{i,j}(\mathbf{z}_{(j)})=\mathbb{E}\left[z_j^{\xi_n^{(\alpha_{i,j})}}\prod_{k=Des(j)}\prod_{p=1}^{|P^{j\rightsquigarrow k}|}z_k^{A_p^{j\rightsquigarrow k}\circ\xi_n^{(\alpha_{i,j})}}\right]$, *we have:*

$$
G_{i,j}(\mathbf{z}_{(j)})=\begin{cases}G_{B(\alpha_{i,j})}\left(z_j\times\prod_{k\in Ch(j)}G_{j,k}(\mathbf{z}_{(k)})\right) & ,Ch(j)\neq\emptyset\\ G_{B(\alpha_{i,j})}(z_j) & ,Otherwise\end{cases}.
$$
$$(A.15)$$

*Proof.* (i) If $\mathrm{Ch}(j)=\emptyset$, which means vertex $j$ has no descendant, we have:

$$
G_{i,j}(\mathbf{z}_{(j)})=\mathbb{E}\left[z_j^{\xi_n^{(\alpha_{i,j})}}\prod_{k=\mathrm{Des}(j)}\prod_{p=1}^{|P^{j\rightsquigarrow k}|}z_k^{A_p^{j\rightsquigarrow k}\circ\xi_n^{(\alpha_{i,j})}}\right]=\mathbb{E}\left[z_j^{\xi_n^{(\alpha_{i,j})}}\right],
$$
$$(A.16)$$

where $\xi_n^{(\alpha_{i,j})}$ is the Bernoulli distribution with parameter $\alpha_{i,j}$. Therefore, $\mathbb{E}\left[z_j^{\xi_n^{(\alpha_{i,j})}}\right]$ is the PGF of Bernoulli$(\alpha_{i,j})$ with input $z_j$, we have $G_{i,j}(\mathbf{z}_{(j)})=\mathbb{E}\left[z_j^{\xi_n^{(\alpha_{i,j})}}\right]=G_{B(\alpha_{i,j})}(z_j)$.

(ii) If $\mathrm{Ch}(j)\neq\emptyset$, we condition the Bernoulli variable $\xi_n^{(\alpha_{i,j})}$, and obtain:

$$
G_{i,j}(\mathbf{z}_{(j)})=\mathbb{E}\left[z_j^{\xi_n^{(\alpha_{i,j})}}\mathbb{E}\left[\prod_{k\in\mathrm{Des}(j)}\prod_{p=1}^{|P^{j\rightsquigarrow k}|}z_k^{A_p^{j\rightsquigarrow k}\circ\xi_n^{(\alpha_{i,j})}}\middle|\xi_n^{(\alpha_{i,j})}\right]\right]
$$

$$
=\mathbb{E}\left[z_j^{\xi_n^{(\alpha_{i,j})}}\prod_{k\in Ch(j)}\mathbb{E}\left[z_k^{\alpha_{j,k}\circ\xi_n^{(\alpha_{i,j})}}\prod_{l\in\mathrm{Des}(k)}\prod_{p=1}^{|P^{k\rightsquigarrow l}|}z_l^{A_p^{k\rightsquigarrow l}\circ\alpha_{j,k}\circ\xi_n^{(\alpha_{i,j})}}\middle|\xi_n^{(\alpha_{i,j})}\right]\right]
$$
$$(A.17)$$

Similar to Lemma 1, we have $\alpha_{j,k} \circ \xi_n^{(\alpha_{i,j})} = \sum_{n=1}^{\xi_n^{(\alpha_{i,j})}} \xi_n^{(\alpha_{j,k})}$. Therefore, Eq. A.17 can be rewritten as follows:

$$
G_{i,j}(\mathbf{z}_{(j)}) = \mathbb{E}\left[ z_j^{\xi_n^{(\alpha_{i,j})}} \prod_{k \in \mathrm{Ch}(j)} \mathbb{E}\left[ \left( z_k^{\xi_n^{(\alpha_{j,k})}} \prod_{l \in \mathrm{Des}(k)} \prod_{p=1}^{|P^{k \rightsquigarrow l}|} z_l^{A_p^{k \rightsquigarrow l} \circ \xi_n^{(\alpha_{j,k})}} \right)^{\xi_n^{(\alpha_{i,j})}} \Bigg| \xi_n^{(\alpha_{i,j})} \right] \right]
$$

$$
= \mathbb{E}\left[ z_j^{\xi_n^{(\alpha_{i,j})}} \prod_{k \in \mathrm{Ch}(j)} \mathbb{E}\left[ z_k^{\xi_n^{(\alpha_{j,k})}} \prod_{l \in \mathrm{Des}(k)} \prod_{p=1}^{|P^{k \rightsquigarrow l}|} z_l^{A_p^{k \rightsquigarrow l} \circ \xi_n^{(\alpha_{j,k})}} \right]^{\xi_n^{(\alpha_{i,j})}} \right]
$$

$$
= \mathbb{E}\left[ \left( z_j \prod_{k \in \mathrm{Ch}(j)} \mathbb{E}\left[ z_k^{\xi_n^{(\alpha_{j,k})}} \prod_{l \in \mathrm{Des}(k)} \prod_{p=1}^{|P^{k \rightsquigarrow l}|} z_l^{A_p^{k \rightsquigarrow l} \circ \xi_n^{(\alpha_{j,k})}} \right] \right)^{\xi_n^{(\alpha_{i,j})}} \right],
$$

(A.18)

which is the PGF of Bernoulli variable $\xi_n^{(\alpha_{i,j})}$. Therefore, we have

$$
G_{i,j}(\mathbf{z}_{(j)}) = G_{B(\alpha_{i,j})}\left( z_j \prod_{k \in \mathrm{Ch}(j)} \mathbb{E}\left[ z_k^{\xi_n^{(\alpha_{j,k})}} \prod_{l \in \mathrm{Des}(k)} \prod_{p=1}^{|P^{k \rightsquigarrow l}|} z_l^{A_p^{k \rightsquigarrow l} \circ \xi_n^{(\alpha_{j,k})}} \right] \right),
$$

(A.19)

where $\mathbb{E}\left[ z_k^{\xi_n^{(\alpha_{j,k})}} \prod_{l \in \mathrm{Des}(k)} \prod_{p=1}^{|P^{k \rightsquigarrow l}|} z_l^{A_p^{k \rightsquigarrow l} \circ \xi_n^{(\alpha_{j,k})}} \right]$ is $G_{j,k}(\mathbf{z}_{(k)})$ according to the definition, that is

$$
G_{i,j}(\mathbf{z}_{(j)}) = G_{B(\alpha_{i,j})}\left( z_j \times \prod_{k \in \mathrm{Ch}(j)} G_{j,k}(\mathbf{z}_{(k)}) \right),
$$

(A.20)

which completes the proof of Lemma 2. $\qquad\square$

Regarding to the Eq. A.9, by leveraging the Lemma 1, we have:

$$
G_{\mathbf{X}}(\mathbf{z}) = \prod_{i \in [n]} \mathbb{E}\left[ z_i^{\epsilon_i} \prod_{j \in \mathrm{Des}(i)} \prod_{p=1}^{|P^{i \rightsquigarrow j}|} z_j^{A_p^{i \rightsquigarrow j} \circ \epsilon_i} \right]
$$

$$
= \prod_{i \in [n]} G_{\epsilon_i}\left( z_i \prod_{j \in \mathrm{Ch}(i)} \mathbb{E}\left[ z_j^{\xi_n^{(\alpha_{i,j})}} \prod_{k \in \mathrm{Des}(j)} \prod_{p=1}^{|P^{j \rightsquigarrow k}|} z_j^{A_p^{j \rightsquigarrow k} \circ \xi_n^{(\alpha_{i,j})}} \right] \right)
$$

(A.21)

Next, according to the Lemma 2, we have:

$$
G_{\mathbf{X}}(\mathbf{z}) = \prod_{i \in [n]} G_{\epsilon_i}\left( z_i \prod_{j \in \mathrm{Ch}(i)} \mathbb{E}\left[ z_j^{\xi_n^{(\alpha_{i,j})}} \prod_{k \in \mathrm{Des}(j)} \prod_{p=1}^{|P^{j \rightsquigarrow k}|} z_k^{A_p^{j \rightsquigarrow k} \circ \xi_n^{(\alpha_{i,j})}} \right] \right) = \prod_{i \in [n]} G_{\epsilon_i}\left( z_i \prod_{j \in \mathrm{Ch}(i)} G_{i,j}(\mathbf{z}_{(j)}) \right)
$$

(A.22)

where $G_{i,j}(\mathbf{z}_{(j)}) = \begin{cases} G_{B(\alpha_{i,j})}\left( z_j \times \prod_{k \in \mathrm{Ch}(j)} G_{j,k}(\mathbf{z}_{(k)}) \right) & , \mathrm{Ch}(j) \neq \emptyset \\ G_{B(\alpha_{i,j})}(z_j) & , \mathrm{Otherwise} \end{cases}$ . This completes the proof.

## A.2 Proof of Theorem 2

**Theorem 2.** *Given a random vector* $\mathbf{X} = [X_1, ..., X_d]^T$ *following PB-SCM, the PGF of* $P(\mathbf{X})$ *can be expressed by:*

$$
G_{\mathbf{X}}(\mathbf{z}) = \prod_{i \in [d]} \exp\{\mu_i \times (T_{X_i}(1) - 1)\},
$$

(A.23)

*where* $T_{X_i}(1) = z_i \sum_{\mathbf{s} \in \{0,1\}^{|Ch(i)|}} \prod_{j \in Ch(i)} \alpha_{i,j}^{s_j} T_{X_j}(s_j), T_{X_i}(0) = 1$ *and* $\alpha_{i,j}^{s_j} = \alpha_{i,j}$ *if* $s_j = 1$ *and* $\alpha_{i,j}^{s_j} = 1 - \alpha_{i,j}$ *if* $s_j = 0$.

*Proof.* According to Theorem 1, we have

$$G_{\mathbf{X}}(\mathbf{z}) = \prod_{i\in[n]} G_{\epsilon_i}\left(z_i \times \prod_{j\in Ch(i)} G_{i,j}(\mathbf{z}_{(j)})\right). \tag{A.24}$$

Since $G_{\epsilon_i}(\cdot)$ is the PGF of Poisson noise $\epsilon_i$, we have:

$$G_{\mathbf{X}}(\mathbf{z}) = \prod_{i\in[n]} \exp\left\{\mu\left(z_i \times \prod_{j\in Ch(i)} G_{i,j}(\mathbf{z}_{(j)}) - 1\right)\right\}. \tag{A.25}$$

Therefore, to complete the proof, it suffices to show that:

$$T_{X_i}(1) = z_i \times \prod_{j\in Ch(i)} G_{i,j}(\mathbf{z}_{(j)}) \tag{A.26}$$

holds for each vertex in the recursive process.

We proceed by structural induction. Specifically, we first show that Eq. A.26 holds for the leaf vertices. Then, assuming that the expression holds for each child vertex $j$ of a parent vertex $i$ (i.e., $j \in \mathrm{Ch}(i)$), we demonstrate that the expression also holds for the parent vertex $i$.

**Base case** Given an arbitrary leaf vertex $l$, since $\mathrm{Ch}(l) = \emptyset$ and by the definition of $T_{X_l}(1)$, we have $T_{X_l}(1) = z_l \times 1$. Therefore, Eq. A.26 holds for the leaf vertex.

**Inductive Step** Next, Assuming that for each child vertex $j$ of $i$, the following holds: $T_{X_j}(1) = z_j \times \prod_{k\in Ch(j)} G_{j,k}(\mathbf{z}_{(k)})$. Our goal is to prove the $T_{X_i}(1) = z_i \times \prod_{j\in Ch(i)} G_{i,j}(\mathbf{z}_{(j)})$.

When $Ch(i) \neq \emptyset$, according to the definition of $G_{i,j}(\mathbf{z}_{(j)})$, we have:

$$\begin{aligned}
\prod_{j\in Ch(i)} G_{i,j}(\mathbf{z}_{(j)}) &= \prod_{j\in Ch(i)} G_{B(\alpha_{i,j})}\left(z_j \times \prod_{k\in Ch(j)} G_{j,k}(\mathbf{z}_{(k)})\right) \\
&= \prod_{j\in Ch(i)} \left(1 - \alpha_{i,j} + \alpha_{i,j} z_j \times \prod_{k\in Ch(j)} G_{j,k}(\mathbf{z}_{(k)})\right).
\end{aligned} \tag{A.27}$$

By expanding Eq. A.27, we have:

$$\prod_{j\in Ch(i)} G_{i,j}(\mathbf{z}_{(j)}) = \sum_{S\subseteq Ch(i)} \left[\prod_{j\in Ch(i)\setminus S}(1-\alpha_{i,j})\prod_{j\in S}\alpha_{i,j}\left(z_j\prod_{k\in Ch(j)} G_{j,k}(\mathbf{z}_{(k)})\right)\right], \tag{A.28}$$

where $S$ represents all subsets of $\mathrm{Ch}(i)$, corresponding to all possible combinations of child vertices of vertex $i$.

Next, regarding the definition of $T_{X_i}(1)$, since $\alpha_{i,j}^{s_j} = \begin{cases} \alpha_{i,j} & s_j = 1 \\ 1-\alpha_{i,j} & s_j = 0 \end{cases}$, we have:

$$\begin{aligned}
T_{X_i}(1) &= z_i \sum_{\mathbf{s}\in\{0,1\}^{|Ch(i)|}} \prod_{j\in Ch(i)} \alpha_{i,j}^{s_j} T_{X_j}(s_j) \\
&= z_i \sum_{S\subseteq Ch(i)} \left[\prod_{j\in Ch(i)\setminus S} \alpha_{i,j}^0 T_{X_j}(0) \prod_{j\in S} \alpha_{i,j}^1 T_{X_j}(1)\right] \\
&= z_i \sum_{S\subseteq Ch(i)} \left[\prod_{j\in Ch(i)\setminus S}(1-\alpha_{i,j})\prod_{j\in S}\alpha_{i,j}\left(z_j\prod_{k\in Ch(j)} G_{j,k}(\mathbf{z}_{(k)})\right)\right],
\end{aligned} \tag{A.29}$$

given $T_{X_j}(1) = z_j \times \prod_{k\in Ch(j)} G_{j,k}(\mathbf{z}_{(k)})$. Then, comparing Eq. A.28 and Eq. A.29, we find that:

$$T_{X_i}(1) = z_i \prod_{j\in Ch(i)} G_{i,j}(\mathbf{z}_{(j)}). \tag{A.30}$$

Substituting back into A.25, we have:

$$\begin{aligned}
G_{\mathbf{X}}(\mathbf{z}) &= \prod_{i\in[n]} \exp\left\{\mu_i\left(z_i \times \prod_{j\in Ch(i)} G_{i,j}(\mathbf{z}_{(j)}) - 1\right)\right\} \\
&= \prod_{i\in[n]} \exp\{\mu_i(T_{X_i}(1) - 1)\},
\end{aligned} \tag{A.31}$$

which completes the proof. □

### A.3  Proof of Theorem 3

**Theorem 3.** *Given a random vector $\mathbf{X} = [X_1, ..., X_n]^T$ following PB-SCM and its PGF $G_{\mathbf{X}}(\mathbf{z})$. For a subset $\mathbf{L} \subset [d]$ of the set of vertices, and the set of the children of vertex $i$ within $\mathbf{L}$, denoted by $Ch_{\mathbf{L}}(i) = Ch(i) \cap \mathbf{L}$, the local PGF of $\mathbf{L}$ can be expressed by:*

$$G_{\mathbf{X}}^{\mathbf{L}}(\mathbf{z}) = \prod_{i \in \mathbf{L}} \exp\left\{\mu_i \times \left(T_{X_i}^{\mathbf{L}}(1) - 1\right)\right\} \prod_{i \in [d]\setminus\mathbf{L}} \exp\{-\mu_i\}, \tag{A.32}$$

*where $T_{X_i}^{\mathbf{L}}(1) = z_i \displaystyle\sum_{\mathbf{s} \in \{0,1\}^{|Ch_{\mathbf{L}}(i)|}} \prod_{j \in Ch_{\mathbf{L}}(i)} \alpha_{i,j}^{s_j} T_{X_j}^{\mathbf{L}}(s_j) \prod_{j \in Ch(i)\setminus Ch_{\mathbf{L}}(i)} (1 - \alpha_{i,j})$ and $T_{X_i}^{\mathbf{L}}(0) = 1$.*

*Proof.* According to the definition of Local PGF, we have:

$$
\begin{aligned}
G_{\mathbf{X}}^{\mathbf{L}}(\mathbf{z}) &= \lim_{\mathbf{z}\setminus\mathbf{L}\to 0} \prod_{i\in[d]} G_{\epsilon_i}\left(z_i \times \prod_{j\in Ch(i)} G_{i,j}(\mathbf{z}_{(j)})\right) \\
&= \lim_{\mathbf{z}\setminus\mathbf{L}\to 0} \prod_{i\in\mathbf{L}} G_{\epsilon_i}\left(z_i \times \prod_{j\in Ch(i)} G_{i,j}(\mathbf{z}_{(j)})\right) \prod_{j\in[d]\setminus\mathbf{L}} G_{\epsilon_i}\left(z_i \times \prod_{j\in Ch(i)} G_{i,j}(\mathbf{z}_{(j)})\right) \\
&= \lim_{\mathbf{z}\setminus\mathbf{L}\to 0} \prod_{i\in\mathbf{L}} G_{\epsilon_i}\left(z_i \times \prod_{j\in Ch(i)} G_{i,j}(\mathbf{z}_{(j)})\right) \prod_{j\in[d]\setminus\mathbf{L}} \lim_{\mathbf{z}\setminus\mathbf{L}\to 0} G_{\epsilon_i}\left(z_i \times \prod_{j\in Ch(i)} G_{i,j}(\mathbf{z}_{(j)})\right).
\end{aligned}
\tag{A.33}
$$

Since $z_j$ approaches zero for each $j \in [d] \setminus \mathbf{L}$, we have:

$$\prod_{j\in[d]\setminus\mathbf{L}} \lim_{\mathbf{z}\setminus\mathbf{L}\to 0} G_{\epsilon_i}\left(z_i \times \prod_{j\in Ch(i)} G_{i,j}(\mathbf{z}_{(j)})\right) \tag{A.34}$$

$$= \prod_{j\in[d]\setminus\mathbf{L}} \lim_{\mathbf{z}\setminus\mathbf{L}\to 0} \exp\left\{\mu_j\left(z_j \times \prod_{k\in Ch(j)} G_{j,k}(\mathbf{z}_{(k)}) - 1\right)\right\} = \prod_{j\in[d]\setminus\mathbf{L}} \exp\{-\mu_j\}. \tag{A.35}$$

Therefore, we have:

$$\log G_{\mathbf{X}}^{\mathbf{L}}(\mathbf{z}) = \lim_{\mathbf{z}\setminus\mathbf{L}\to 0} \prod_{i\in\mathbf{L}} G_{\epsilon_i}\left(z_i \times \prod_{j\in Ch(i)} G_{i,j}(\mathbf{z}_{(j)})\right) \prod_{j\in[d]\setminus\mathbf{L}} \exp\{-\mu_i\}. \tag{A.36}$$

Next, we consider the $\lim_{\mathbf{z}\setminus\mathbf{L}\to 0} G_{\epsilon_i}\left(z_i \times \prod_{j\in Ch(i)} G_{i,j}(\mathbf{z}_{(j)})\right)$. According to Theorem 2, it can be expressed as follows :

$$
\begin{aligned}
&\lim_{\mathbf{z}\setminus\mathbf{L}\to 0} G_{\epsilon_i}\left(z_i \times \prod_{j\in Ch(i)} G_{i,j}(\mathbf{z}_{(j)})\right) \\
&= \lim_{\mathbf{z}\setminus\mathbf{L}\to 0} \exp\{\mu_i(T_{X_i}(1) - 1)\} \\
&= \exp\left\{\mu_i\left(\lim_{\mathbf{z}\setminus\mathbf{L}\to 0} T_{X_i}(1) - 1\right)\right\} \\
&= \exp\left\{\mu_i \times \left(z_i \lim_{\mathbf{z}\setminus\mathbf{L}\to 0} \sum_{\mathbf{s}\in\{0,1\}^{|Ch(i)|}} \prod_{j\in Ch(i)} \alpha_{i,j}^{s_j} T_{X_j}(s_j) - 1\right)\right\}.
\end{aligned}
\tag{A.37}
$$

Next, we are going to show that, when $\lim_{\mathbf{z}\setminus\mathbf{L}\to 0} T_{X_j}(s_j) = T_{X_j}^{\mathbf{L}}(s_j)$, we have

$$\lim_{\mathbf{z}\setminus\mathbf{L}\to 0} T_{X_i}(1) = z_i \sum_{\mathbf{s}\in\{0,1\}^{|Ch_{\mathbf{L}}(i)|}} \prod_{j\in Ch_{\mathbf{L}}(i)} \alpha_{i,j}^{s_j} T_{X_j}^{\mathbf{L}}(s_j) \prod_{j\in Ch(i)\setminus Ch_{\mathbf{L}}(i)} (1 - \alpha_{i,j}) = T_{X_i}^{\mathbf{L}}(1). \tag{A.38}$$

Let $Ch_{\mathbf{L}}(i) = Ch(i) \cap \mathbf{L}$, we have:

$$
\begin{aligned}
&\lim_{\mathbf{z}\setminus\mathbf{L}\to 0} G_{\epsilon_i}\left(z_i \times \prod_{j\in Ch(i)} G_{i,j}(\mathbf{z}_{(j)})\right) \\
&= \lim_{\mathbf{z}\setminus\mathbf{L}\to 0} \exp\left\{\mu_i \times \left(z_i \sum_{\mathbf{s}\in\{0,1\}^{|Ch(i)|}} \prod_{j\in Ch_{\mathbf{L}}(i)} \alpha_{i,j}^{s_j} T_{X_j}(s_j) \prod_{j\in Ch(i)\setminus Ch_{\mathbf{L}}(i)} \alpha_{i,j}^{s_j} T_{X_j}(s_j) - 1\right)\right\}.
\end{aligned}
\tag{A.39}
$$

In scenarios where $j \in Ch(i) \setminus Ch_\mathbf{L}(i)$ and $s_j = 1$, considering that $z_j \in \mathbf{z}_{\setminus \mathbf{L}}$ and $z_j \to 0$, (that is, $z_j$ approach to zero for those vertices $j \notin \mathbf{L}$) ,we have:

$$\lim_{\mathbf{z}_{\setminus \mathbf{L}} \to \mathbf{0}} T_{X_j}(s_j) = \lim_{\mathbf{z}_{\setminus \mathbf{L}} \to \mathbf{0}} z_j \sum_{\mathbf{s} \in \{0,1\}^{|Ch(j)|}} \prod_{k \in Ch(j)} \alpha_{j,k}^{s_k} T_{X_k}(s_k) = 0. \quad (A.40)$$

Otherwise, when $j \in Ch_\mathbf{L}(i)$, we have:

$$\begin{aligned}
\lim_{\mathbf{z}_{\setminus \mathbf{L}} \to \mathbf{0}} T_{X_j}(s_j) &= \lim_{\mathbf{z}_{\setminus \mathbf{L}} \to \mathbf{0}} z_j \sum_{\mathbf{s} \in \{0,1\}^{|Ch(j)|}} \prod_{k \in Ch(j)} \alpha_{j,k}^{s_k} T_{X_k}(s_k) \\
&= z_j \sum_{\mathbf{s} \in \{0,1\}^{|Ch(j)|}} \prod_{k \in Ch(j)} \alpha_{j,k}^{s_k} \lim_{\mathbf{z}_{\setminus \mathbf{L}} \to \mathbf{0}} T_{X_k}(s_k) \\
&= z_j \sum_{\mathbf{s} \in \{0,1\}^{|Ch(j)|}} \prod_{k \in Ch(j)} \alpha_{j,k}^{s_k} T_{X_k}^\mathbf{L}(s_k),
\end{aligned} \quad (A.41)$$

and when $j \in Ch(i) \setminus Ch_\mathbf{L}(i)$ and $s_j = 0$, we have $\lim_{\mathbf{z}_{\setminus \mathbf{L}} \to \mathbf{0}} T_{X_j}(0) = 1$. Therefore, we have:

$$\begin{aligned}
&\lim_{\mathbf{z}_{\setminus \mathbf{L}} \to \mathbf{0}} G_{\epsilon_i} \Big( z_i \times \prod_{j \in Ch(i)} G_{i,j}(\mathbf{z}_{(j)}) \Big) \\
&= \exp \left\{ \mu_i \times \left( z_i \sum_{\mathbf{s} \in \{0,1\}^{|Ch_\mathbf{L}(i)|}} \prod_{j \in Ch_\mathbf{L}(i)} \alpha_{i,j}^{s_j} \lim_{\mathbf{z}_{\setminus \mathbf{L}} \to \mathbf{0}} T_{X_j}(s_j) \prod_{j \in Ch(i) \setminus Ch_\mathbf{L}(i)} (1 - \alpha_{i,j}) - 1 \right) \right\} \\
&= \exp \left\{ \mu_i \times \left( z_i \sum_{\mathbf{s} \in \{0,1\}^{|Ch_\mathbf{L}(i)|}} \prod_{j \in Ch_\mathbf{L}(i)} \alpha_{i,j}^{s_j} T_{X_j}^\mathbf{L}(s_j) \prod_{j \in Ch(i) \setminus Ch_\mathbf{L}(i)} (1 - \alpha_{i,j}) - 1 \right) \right\}.
\end{aligned} \quad (A.42)$$

According to the definition of $T_{X_i}^\mathbf{L}$, we obtain:

$$\lim_{\mathbf{z}_{\setminus \mathbf{L}} \to \mathbf{0}} G_{\epsilon_i} \Big( z_i \times \prod_{j \in Ch(i)} G_{i,j}(\mathbf{z}_{(j)}) \Big) = \exp \left\{ \mu_i \times \left( T_{X_i}^\mathbf{L}(1) - 1 \right) \right\}. \quad (A.43)$$

By substituting Eq. A.43 into Eq. A.36, we finally have:

$$\log G_\mathbf{X}^\mathbf{L}(\mathbf{z}) = \prod_{i \in \mathbf{L}} \exp \left\{ \mu_i \times \left( T_{X_i}^\mathbf{L}(1) - 1 \right) \right\} \prod_{i \in [d] \setminus \mathbf{L}} \exp\{-\mu_i\}, \quad (A.44)$$

which completes the proof. □

## A.4 Proof of Proposition 1

**Proposition 1** (Graphical implication of closed-form PGF). *Given a random vector* $\mathbf{X} = [X_1, ..., X_d]^T$ *following PB-SCM, for any subgraph* $G(\mathbf{L}, \mathbf{E})$ *with the subset of vertices* $\mathbf{L} \subseteq \mathbf{V}$ *such that the $i$-th vertex is the root vertex in* $G(\mathbf{L}, \mathbf{E})$, *a component* $C z_i \prod_{j \in L \setminus \{i\}} z_j^{p_j}$ *with constant* $C \neq 0$, *exponent* $p_j \in \mathbb{Z}^+$ *exits in* $T_{X_i}(1)$, *if and only if there exists a subgraph* $G(\mathbf{L}, \mathbf{E}')$ *with subset of the edges* $\mathbf{E}' \subseteq \mathbf{E}$ *such that for each* $j \in \mathbf{L} \setminus \{i\}$, *there are at least* $p_j$ *directed paths from* $X_i$ *to* $X_j$ *in the subgraph* $G(\mathbf{L}, \mathbf{E}')$.

*Proof.* We consider the subgraph $G(\mathbf{L}, \mathbf{E}')$ as a local structure associated with the node set $L$. According to Theorem 3, we have:

$$T_{X_i}^\mathbf{L}(1) = z_i \sum_{\mathbf{s} \in \{0,1\}^{|Ch_\mathbf{L}(i)|}} \prod_{j \in Ch_\mathbf{L}(i)} \alpha_{i,j}^{s_j} T_{X_j}^\mathbf{L}(s_j) \prod_{j \in Ch(i) \setminus Ch_\mathbf{L}(i)} (1 - \alpha_{i,j}) \quad (A.45)$$

We select a subset of edges $\mathbf{E}' \subseteq \mathbf{E}$, indicating that edges in $\mathbf{E}'$ are 'open' and edges in $\mathbf{E} \setminus \mathbf{E}'$ are 'closed' within the local structure containing the vertex set $L$. This means that edges not included in $\mathbf{E}'$ are considered non-existent in the subgraph. Consequently, in Eq. A.45, we have: (1) $\alpha_{i,j}^{s_j=1} T_{X_j}^\mathbf{L}(s_j = 1)$ if the edge $i \to j$ is in $\mathbf{E}'$, and (2) $\alpha_{i,j}^{s_j=0} T_{X_j}^\mathbf{L}(s_j = 0)$ if the edge $i \to j$ is not

in $\mathbf{E}'$. That is, each subset $\mathbf{E}'$ corresponds to an element in $\{0,1\}^{|Ch_\mathbf{L}(i)|}$, representing a possible combination of whether each edge is 'open' or is 'close'.

We describe the term $Cz_i \prod_{j\in L\setminus\{i\}} z_j^{p_j}$ through a recursive process. Initially considering the children of $X_i$, there exists a term in $T_{X_i}^{\mathbf{L}}(1)$ as follows:

$$
z_i \prod_{\substack{j\in Ch_\mathbf{L}(i)\\ i\to j\in\mathbf{E}'}} \alpha_{i,j}^1 T_{X_j}^{\mathbf{L}}(1) \prod_{\substack{j\in Ch_\mathbf{L}(i)\\ i\to j\notin\mathbf{E}'}} \alpha_{i,j}^0 T_{X_j}^{\mathbf{L}}(0) \prod_{j\in Ch(i)\setminus Ch_\mathbf{L}(i)} (1-\alpha_{i,j})
$$

$$
= z_i \prod_{\substack{j\in Ch_\mathbf{L}(i)\\ i\to j\in\mathbf{E}'}} \alpha_{i,j} z_j \left( \sum_{\mathbf{s}\in\{0,1\}^{|Ch_\mathbf{L}(j)|}} \prod_{k\in Ch_\mathbf{L}(j)} \alpha_{j,k}^{s_k} T_{X_k}^{\mathbf{L}}(s_k) \right) \underbrace{\prod_{\substack{j\in Ch_\mathbf{L}(i)\\ i\to j\notin\mathbf{E}'}} (1-\alpha_{i,j}) \prod_{j\in Ch(i)\setminus Ch_\mathbf{L}(i)} (1-\alpha_{i,j})}_{\text{Constant coefficient}}
$$

$$
= C \times z_i \prod_{\substack{j\in Ch_\mathbf{L}(i)\\ i\to j\in\mathbf{E}'}} z_j \left( \sum_{\mathbf{s}\in\{0,1\}^{|Ch_\mathbf{L}(j)|}} \prod_{k\in Ch_\mathbf{L}(j)} \alpha_{j,k}^{s_k} T_{X_k}^{\mathbf{L}}(s_k) \right)
$$

(A.46)

Continuing from the Eq. A.46, for each term in the product expansion, we can similarly expand based on the children of $X_j$. This expansion includes the following term:

$$
C \times z_i \prod_{\substack{j\in Ch_\mathbf{L}(i)\\ i\to j\in\mathbf{E}'}} z_j \left( \prod_{\substack{k\in Ch_\mathbf{L}(j)\\ j\to k\in\mathbf{E}'}} \alpha_{j,k}^1 T_{X_k}^{\mathbf{L}}(1) \underbrace{\prod_{\substack{k\in Ch_\mathbf{L}(j)\\ j\to k\notin\mathbf{E}'}} \alpha_{j,k}^0 T_{X_j}^{\mathbf{L}}(0) \prod_{k\in Ch(j)\setminus Ch_\mathbf{L}(j)} (1-\alpha_{j,k})}_{\text{Constant coefficient}} \right)
$$

$$
= C \times z_i \prod_{\substack{j\in Ch_\mathbf{L}(i)\\ i\to j\in\mathbf{E}'}} z_j \left( \prod_{\substack{k\in Ch_\mathbf{L}(j)\\ j\to k\in\mathbf{E}'}} z_k \sum_{\mathbf{s}\in\{0,1\}^{|Ch_\mathbf{L}(k)|}} \prod_{l\in Ch_\mathbf{L}(k)} \alpha_{k,l}^{s_l} T_{X_l}^{\mathbf{L}}(s_l) \prod_{l\in Ch(k)\setminus Ch_\mathbf{L}(k)} (1-\alpha_{k,l}) \right)
$$

$$
= C \times z_i \prod_{\substack{j\in Ch_\mathbf{L}(i)\\ i\to j\in\mathbf{E}'}} z_j \prod_{\substack{k\in Ch_\mathbf{L}(j)\\ j\to k\in\mathbf{E}'}} z_k \left( \sum_{\mathbf{s}\in\{0,1\}^{|Ch_\mathbf{L}(k)|}} \prod_{l\in Ch_\mathbf{L}(k)} \alpha_{k,l}^{s_l} T_{X_l}^{\mathbf{L}}(s_l) \prod_{l\in Ch(k)\setminus Ch_\mathbf{L}(k)} (1-\alpha_{k,l}) \right)
$$

(A.47)

The term "$z_i \prod_{\substack{j\in Ch_\mathbf{L}(i)\\ i\to j\in\mathbf{E}'}} z_j \prod_{\substack{k\in Ch_\mathbf{L}(j)\\ j\to k\in\mathbf{E}'}} z_k$" describes how $X_i$ can reach its descendants $X_k$, through the children $X_j$. This shows that all reachable vertices are expressed in the form of the product of $z$ variables, indicating that in the subgraph $G(\mathbf{L},\mathbf{E}')$, the number of paths $X_i$ takes to reach its descendants $k\in\text{Des}(i)$ is given in the power of $z_k$. Thus, the corresponding term for the subgraph $G(\mathbf{L},\mathbf{E}')$, $Cz_i \prod_{j\in L\setminus\{i\}} z_j^{p_j}$, is included in $T_{X_i}(1)$, where $p_j$ represents the number of paths from $X_i$ to $X_j$ in the subgraph $G(\mathbf{L},\mathbf{E}')$. Conversely, if such a subgraph $G(\mathbf{L},\mathbf{E}')$ does not exist, then $T_{X_i}(1)$ will not contain such a term $Cz_i \prod_{j\in L\setminus\{i\}} z_j^{p_j}$, since the number of directed paths from $i$ to $j$ is less then $p_j$. This completes the proof. $\qquad\square$

## B  Proof of Identifiability

We first introduce the following necessary Lemmas for our proof.

**Lemma 3.** *Given two vertices $i,j\in\mathbf{V}$ and the corresponding local PGF $G_\mathbf{X}^{\{i,j\}}(\mathbf{z})$, $\frac{\partial^2 \log G_\mathbf{X}^{\{i,j\}}(\mathbf{z})}{\partial z_i \partial z_j} = 0$ if and only if vertex $i$ is non-adjacent to vertex $j$.*

**Lemma 4.** *Given three vertices $i,j,k\in\mathbf{V}$ and the corresponding local PGF $G_\mathbf{X}^{\{i,j,k\}}(\mathbf{z})$, $\frac{\partial^2 \log G_\mathbf{X}^{\{i,j,k\}}(\mathbf{z})}{\partial z_i \partial z_k} = 0$ if and only if vertices $i,j,k$ form the structure $i\to j\leftarrow k$.*

**Lemma 5.** *Given three vertices $i,j,k\in\mathbf{V}$ and the corresponding local PGF $G_\mathbf{X}^{\{i,j,k\}}(\mathbf{z})$, $\frac{\partial^2 \log G_\mathbf{X}^{\{i,j,k\}}(\mathbf{z})}{\partial z_k^2} \neq 0$ if and only if vertices $i,j,k$ form the structure $i\to k\leftarrow j$ and vertex $i$ is adjacent to vertex $j$.*

The proofs of Lemma 3, Lemma 4, and Lemma 5 are deferred to section B.5.

## B.1 Proof of Theorem 4

**Theorem 4** (Identifiability of adjacent vertices). *Let $X_i, X_j \in \mathbf{X}$ be two arbitrary vertices with the corresponding local PGF $G_{\mathbf{X}}^{\{i,j\}}(\mathbf{z})$. Define the matrix $\mathbf{A}^{\{i,j\}} = \begin{pmatrix} G_{\mathbf{X}}^{\{i,j\}}(\mathbf{z}) & \frac{\partial G_{\mathbf{X}}^{\{i,j\}}(\mathbf{z})}{\partial z_i} \\ \frac{\partial G_{\mathbf{X}}^{\{i,j\}}(\mathbf{z})}{\partial z_j} & \frac{\partial^2 G_{\mathbf{X}}^{\{i,j\}}(\mathbf{z})}{\partial z_i \partial z_j} \end{pmatrix}$ with $z_i, z_j$ approach 1, the condition $\mathrm{Rank}(\mathbf{A}^{\{i,j\}}) = 1$ if and only if $X_i$ is non-adjacent to $X_j$.*

*Proof.* Considering the partial derivative of $\log G_{\mathbf{X}}^{\{i,j\}}(\mathbf{z})$, we have

$$\frac{\partial \log G_{\mathbf{X}}^{\{i,j\}}(\mathbf{z})}{\partial z_i} = \frac{1}{G_{\mathbf{X}}^{\{i,j\}}(\mathbf{z})} \frac{\partial G_{\mathbf{X}}^{\{i,j\}}(\mathbf{z})}{\partial z_i} \tag{B.1}$$

$$\frac{\partial^2 \log G_{\mathbf{X}}^{\{i,j\}}(\mathbf{z})}{\partial z_i \partial z_j} = \frac{1}{G_{\mathbf{X}}^{\{i,j\}}(\mathbf{z})} \frac{\partial^2 G_{\mathbf{X}}^{\{i,j\}}(\mathbf{z})}{\partial z_i \partial z_j} - \frac{1}{\left(G_{\mathbf{X}}^{\{i,j\}}(\mathbf{z})\right)^2} \frac{\partial G_{\mathbf{X}}^{\{i,j\}}(\mathbf{z})}{\partial z_i} \frac{\partial G_{\mathbf{X}}^{\{i,j\}}(\mathbf{z})}{\partial z_j} \tag{B.2}$$

**If part**    If vertex $i$ in non-adjacent to vertex $j$, according to Lemma 3, we have $\frac{\partial^2 \log G_{\mathbf{X}}^{\{i,j\}}(\mathbf{z})}{\partial z_i \partial z_j} = 0$, that is

$$\frac{1}{G_{\mathbf{X}}^{\{i,j\}}(\mathbf{z})} \frac{\partial^2 G_{\mathbf{X}}^{\{i,j\}}(\mathbf{z})}{\partial z_i \partial z_j} - \frac{1}{\left(G_{\mathbf{X}}^{\{i,j\}}(\mathbf{z})\right)^2} \frac{\partial G_{\mathbf{X}}^{\{i,j\}}(\mathbf{z})}{\partial z_i} \frac{\partial G_{\mathbf{X}}^{\{i,j\}}(\mathbf{z})}{\partial z_j} = 0. \tag{B.3}$$

By rearranging the equation, we obtain:

$$G_{\mathbf{X}}^{(i,j)}(\mathbf{z}) \frac{\partial^2 G_{\mathbf{X}}^{(i,j)}(\mathbf{z})}{\partial z_i \partial z_j} - \frac{\partial G_{\mathbf{X}}^{(i,j)}(\mathbf{z})}{\partial z_i} \frac{\partial G_{\mathbf{X}}^{(i,j)}(\mathbf{z})}{\partial z_j} = 0. \tag{B.4}$$

Notably, Eq. B.4 is the determinant of matrix $\mathbf{A}^{\{i,j\}} = \begin{pmatrix} G_{\mathbf{X}}^{\{i,j\}}(\mathbf{z}) & \frac{\partial G_{\mathbf{X}}^{\{i,j\}}(\mathbf{z})}{\partial z_i} \\ \frac{\partial G_{\mathbf{X}}^{\{i,j\}}(\mathbf{z})}{\partial z_j} & \frac{\partial^2 G_{\mathbf{X}}^{\{i,j\}}(\mathbf{z})}{\partial z_i \partial z_j} \end{pmatrix}$, and the $\det(\mathbf{A}^{\{i,j\}}) = 0$ means that the rank of matrix $\mathbf{A}^{\{i,j\}}$ is 1, i.e., $\mathrm{Rank}\left(\mathbf{A}^{\{i,j\}}\right) = 1$. This completes the proof of the if part.

**Only If part**    If $\mathrm{Rank}(\mathbf{A}^{\{i,j\}}) = 1$, we have $\det(\mathbf{A}^{\{i,j\}}) = 0$, which means:

$$G_{\mathbf{X}}^{\{i,j\}}(\mathbf{z}) \frac{\partial^2 G_{\mathbf{X}}^{\{i,j\}}(\mathbf{z})}{\partial z_i \partial z_j} - \frac{\partial G_{\mathbf{X}}^{\{i,j\}}(\mathbf{z})}{\partial z_i} \frac{\partial G_{\mathbf{X}}^{\{i,j\}}(\mathbf{z})}{\partial z_j} = 0 \tag{B.5}$$

Divide both sides by $\left(G_{\mathbf{X}}^{\{i,j\}}(\mathbf{z})\right)^2$. we have:

$$\frac{1}{G_{\mathbf{X}}^{\{i,j\}}(\mathbf{z})} \frac{\partial^2 G_{\mathbf{X}}^{\{i,j\}}(\mathbf{z})}{\partial z_i \partial z_j} - \frac{1}{\left(G_{\mathbf{X}}^{\{i,j\}}(\mathbf{z})\right)^2} \frac{\partial G_{\mathbf{X}}^{\{i,j\}}(\mathbf{z})}{\partial z_i} \frac{\partial G_{\mathbf{X}}^{\{i,j\}}(\mathbf{z})}{\partial z_j} = 0 \tag{B.6}$$

Comparing to Eq. B.2, we have $\frac{\partial^2 \log G_{\mathbf{X}}^{\{i,j\}}(\mathbf{z})}{\partial z_i \partial z_j} = 0$. Therefore, according to Lemma 3, $X_i$ is non-adjacent to $X_j$. This completes the proof of Theorem 4. $\square$

## B.2 Proof of Theorem 5

**Theorem 5** (Identifiability of triangle structure). *Let $X_i, X_j, X_k \in \mathbf{X}$ form a triangular structure with the corresponding local PGF $G_{\mathbf{X}}^{\{i,j,k\}}(\mathbf{z})$. Define the matrix $\mathbf{B}^{\{i,j,k\}} =$*

$$
\begin{pmatrix} G_{\mathbf{X}}^{\{i,j,k\}}(\mathbf{z}) & \frac{\partial G_{\mathbf{X}}^{\{i,j,k\}}(\mathbf{z})}{\partial z_k} \\ \frac{\partial G_{\mathbf{X}}^{\{i,j,k\}}(\mathbf{z})}{\partial z_k} & \frac{\partial^2 G_{\mathbf{X}}^{\{i,j,k\}}(\mathbf{z})}{\partial z_k^2} \end{pmatrix}
$$
*with* $z_i, z_j, z_k$ *approach* 1, *the condition* $\mathrm{Rank}(\mathbf{B}^{\{i,j,k\}}) = 2$ *if and only if* $X_k$ *is the vertex with an in-degree of* 2 *in this triangular structure, i.e.,* $X_i \overset{\frown}{\rightarrow} X_k \leftarrow X_j$.

*Proof.* Considering the second order partial derivative of $\log G_{\mathbf{X}}^{\{i,j,k\}}(\mathbf{z})$ with respect to $z_k$, we have

$$
\frac{\partial \log G_{\mathbf{X}}^{\{i,j,k\}}(\mathbf{z})}{\partial z_k} = \frac{1}{G_{\mathbf{X}}^{\{i,j,k\}}(\mathbf{z})} \frac{\partial G_{\mathbf{X}}^{\{i,j,k\}}(\mathbf{z})}{\partial z_k} \tag{B.7}
$$

$$
\frac{\partial^2 \log G_{\mathbf{X}}^{\{i,j,k\}}(\mathbf{z})}{\partial z_k^2} = \frac{1}{G_{\mathbf{X}}^{\{i,j,k\}}(\mathbf{z})} \frac{\partial^2 G_{\mathbf{X}}^{\{i,j,k\}}(\mathbf{z})}{\partial z_k^2} - \frac{1}{\left(G_{\mathbf{X}}^{\{i,j,k\}}(\mathbf{z})\right)^2} \left( \frac{\partial G_{\mathbf{X}}^{\{i,j,k\}}(\mathbf{z})}{\partial z_k} \right)^2 \tag{B.8}
$$

**If part**   If $X_i, X_j, X_k$ form the triangular where $X_k$ is the vertex with indegree of 2, according to Lemma 5, we have $\frac{\partial^2 \log G_{\mathbf{X}}^{\{i,j,k\}}(\mathbf{z})}{\partial z_k^2} \neq 0$, that is

$$
\frac{1}{G_{\mathbf{X}}^{\{i,j,k\}}(\mathbf{z})} \frac{\partial^2 G_{\mathbf{X}}^{\{i,j,k\}}(\mathbf{z})}{\partial z_k^2} - \frac{1}{\left(G_{\mathbf{X}}^{\{i,j,k\}}(\mathbf{z})\right)^2} \left( \frac{\partial G_{\mathbf{X}}^{\{i,j,k\}}(\mathbf{z})}{\partial z_k} \right)^2 \neq 0. \tag{B.9}
$$

By rearranging the equation, we obtain:

$$
G_{\mathbf{X}}^{\{i,j,k\}}(\mathbf{z}) \frac{\partial^2 G_{\mathbf{X}}^{\{i,j,k\}}(\mathbf{z})}{\partial z_k^2} - \left( \frac{\partial G_{\mathbf{X}}^{\{i,j,k\}}(\mathbf{z})}{\partial z_k} \right)^2 \neq 0, \tag{B.10}
$$

Notably, Eq. B.10 is the determinant of matrix $\mathbf{B}^{\{i,j,k\}} = \begin{pmatrix} G_{\mathbf{X}}^{\{i,j,k\}}(\mathbf{z}) & \frac{\partial G_{\mathbf{X}}^{\{i,j,k\}}(\mathbf{z})}{\partial z_k} \\ \frac{\partial G_{\mathbf{X}}^{\{i,j,k\}}(\mathbf{z})}{\partial z_k} & \frac{\partial^2 G_{\mathbf{X}}^{\{i,j,k\}}(\mathbf{z})}{\partial z_k^2} \end{pmatrix}$, and the $\det(\mathbf{B}^{\{i,j,k\}}) \neq 0$ means the rank of matrix $\mathbf{B}^{\{i,j,k\}}$ is full rank, i.e., $\mathrm{Rank}\left(\mathbf{B}^{\{i,j,k\}}\right) = 2$. This completes the proof of the if part.

**Only if part**   If $\mathrm{Rank}\left(\mathbf{B}^{\{i,j,k\}}\right) = 2$, we have

$$
\det\left(\mathbf{B}^{\{i,j,k\}}\right) = G_{\mathbf{X}}^{\{i,j,k\}}(\mathbf{z}) \frac{\partial^2 G_{\mathbf{X}}^{\{i,j,k\}}(\mathbf{z})}{\partial z_k^2} - \left( \frac{\partial G_{\mathbf{X}}^{\{i,j,k\}}(\mathbf{z})}{\partial z_k} \right)^2 \neq 0. \tag{B.11}
$$

Divide both sides by $\left(G_{\mathbf{X}}^{\{i,j,k\}}(\mathbf{z})\right)^2$, we have

$$
\frac{1}{G_{\mathbf{X}}^{\{i,j,k\}}(\mathbf{z})} \frac{\partial^2 G_{\mathbf{X}}^{\{i,j,k\}}(\mathbf{z})}{\partial z_k^2} - \frac{1}{\left(G_{\mathbf{X}}^{\{i,j,k\}}(\mathbf{z})\right)^2} \left( \frac{\partial G_{\mathbf{X}}^{\{i,j,k\}}(\mathbf{z})}{\partial z_k} \right)^2 \neq 0, \tag{B.12}
$$

and hence we have $\frac{\partial^2 \log G_{\mathbf{X}}^{\{i,j,k\}}(\mathbf{z})}{\partial z_k^2} \neq 0$. According to Lemma 4, the vertex $X_i, X_j, X_k$ form the triangular where $X_k$ is the vertex with an in-degree of 2, i.e., $X_i \overset{\frown}{\rightarrow} X_k \leftarrow X_j$. This completes the proof of the only if part. $\qquad \square$

### B.3   Proof of Theorem 6

**Theorem 6** (Identifiability of collider structure). *Let* $X_i, X_j, X_k \in \mathbf{X}$ *be three adjacent vertices with the corresponding local PGF* $G_{\mathbf{X}}^{\{i,j,k\}}(\mathbf{z})$. *Define the matrix* $\mathbf{C}^{\{i,j,k\}} = \begin{pmatrix} G_{\mathbf{X}}^{\{i,j,k\}}(\mathbf{z}) & \frac{\partial G_{\mathbf{X}}^{\{i,j,k\}}(\mathbf{z})}{\partial z_i} \\ \frac{\partial G_{\mathbf{X}}^{\{i,j,k\}}(\mathbf{z})}{\partial z_k} & \frac{\partial^2 G_{\mathbf{X}}^{\{i,j,k\}}(\mathbf{z})}{\partial z_i \partial z_k} \end{pmatrix}$ *with* $z_i, z_j, z_k$ *approach* 1, *the condition* $\mathrm{Rank}(\mathbf{C}^{\{i,j,k\}}) = 1$ *if and only if the vertices* $X_i, X_j, X_k$ *form the structure* $X_i \rightarrow X_j \leftarrow X_k$.

*Proof.* The proof of Theorem 6 is similar to that of Theorem 4. Consider the partial derivative of $\log G_{\mathbf{X}}^{\{i,j,k\}}(\mathbf{z})$, we have

$$\frac{\partial^2 \log G_{\mathbf{X}}^{\{i,j,k\}}(\mathbf{z})}{\partial z_i \partial z_k} = \frac{1}{G_{\mathbf{X}}^{\{i,j,k\}}(\mathbf{z})} \frac{\partial^2 G_{\mathbf{X}}^{\{i,j,k\}}(\mathbf{z})}{\partial z_i \partial z_k} - \frac{1}{\left(G_{\mathbf{X}}^{\{i,j,k\}}(\mathbf{z})\right)^2} \frac{\partial G_{\mathbf{X}}^{\{i,j,k\}}(\mathbf{z})}{\partial z_i} \frac{\partial G_{\mathbf{X}}^{\{i,j,k\}}(\mathbf{z})}{\partial z_k}$$

(B.13)

**If part**  If vertex $i, j, k$ form the structure $i \to j \leftarrow k$, according to Lemma 4, we have $\frac{\partial^2 \log G_{\mathbf{X}}^{\{i,j,k\}}(\mathbf{z})}{\partial z_i \partial z_k} = 0$, that is

$$\frac{1}{G_{\mathbf{X}}^{\{i,j,k\}}(\mathbf{z})} \frac{\partial^2 G_{\mathbf{X}}^{\{i,j,k\}}(\mathbf{z})}{\partial z_i \partial z_k} - \frac{1}{\left(G_{\mathbf{X}}^{\{i,j,k\}}(\mathbf{z})\right)^2} \frac{\partial G_{\mathbf{X}}^{\{i,j,k\}}(\mathbf{z})}{\partial z_i} \frac{\partial G_{\mathbf{X}}^{\{i,j,k\}}(\mathbf{z})}{\partial z_k} = 0.$$

(B.14)

By rearranging the equation, we obtain:

$$G_{\mathbf{X}}^{(i,j,k)}(\mathbf{z}) \frac{\partial^2 G_{\mathbf{X}}^{(i,j,k)}(\mathbf{z})}{\partial z_i \partial z_k} - \frac{\partial G_{\mathbf{X}}^{(i,j,k)}(\mathbf{z})}{\partial z_i} \frac{\partial G_{\mathbf{X}}^{(i,j,k)}(\mathbf{z})}{\partial z_k} = 0.$$

(B.15)

Notably, Eq. B.15 is the determinant of matrix $\mathbf{C}^{\{i,j,k\}} = \begin{pmatrix} G_{\mathbf{X}}^{\{i,j,k\}}(\mathbf{z}) & \frac{\partial G_{\mathbf{X}}^{\{i,j,k\}}(\mathbf{z})}{\partial z_i} \\ \frac{\partial G_{\mathbf{X}}^{\{i,j,k\}}(\mathbf{z})}{\partial z_k} & \frac{\partial^2 G_{\mathbf{X}}^{\{i,j,k\}}(\mathbf{z})}{\partial z_i \partial z_k} \end{pmatrix}$, and the $\det(\mathbf{C}^{\{i,j,k\}}) = 0$ means the rank of matrix $\mathbf{C}^{\{i,j,k\}}$ is 1, i.e., $\text{Rank}\left(\mathbf{C}^{\{i,j,k\}}\right) = 1$. This completes the proof of the if part.

**Only if part**  If $\text{Rank}(\mathbf{C}^{\{i,j,k\}}) = 1$, we have $\det(\mathbf{C}^{\{i,j,k\}}) = 0$, which means:

$$G_{\mathbf{X}}^{\{i,j,k\}}(\mathbf{z}) \frac{\partial^2 G_{\mathbf{X}}^{\{i,j,k\}}(\mathbf{z})}{\partial z_i \partial z_k} - \frac{\partial G_{\mathbf{X}}^{\{i,j,k\}}(\mathbf{z})}{\partial z_i} \frac{\partial G_{\mathbf{X}}^{\{i,j,k\}}(\mathbf{z})}{\partial z_k} = 0$$

(B.16)

Divide both sides by $\left(G_{\mathbf{X}}^{\{i,j,k\}}(\mathbf{z})\right)^2$. we have:

$$\frac{1}{G_{\mathbf{X}}^{\{i,j,k\}}(\mathbf{z})} \frac{\partial^2 G_{\mathbf{X}}^{\{i,j,k\}}(\mathbf{z})}{\partial z_i \partial z_k} - \frac{1}{\left(G_{\mathbf{X}}^{\{i,j,k\}}(\mathbf{z})\right)^2} \frac{\partial G_{\mathbf{X}}^{\{i,j,k\}}(\mathbf{z})}{\partial z_i} \frac{\partial G_{\mathbf{X}}^{\{i,j,k\}}(\mathbf{z})}{\partial z_k} = 0$$

(B.17)

Regarding to Eq. B.13, we have $\frac{\partial^2 \log G_{\mathbf{X}}^{\{i,j,k\}}(\mathbf{z})}{\partial z_i \partial z_k} = 0$, according to Lemma 4, vertices $i, j, k$ form the structure $i \to j \leftarrow k$. $\qquad \square$

## B.4  Proof of Theorem 7

**Theorem 7** (Graphical Implication of Identifiability). *Given a pair of adjacent vertices $X_i, X_j \in \mathbf{X}$ following PB-SCM, the causal direction of $X_i \to X_j$ is identifiable if there exists a vertex $X_k \in \mathbf{X} \setminus \{X_i, X_j\}$ such that $X_k \to X_j$.*

*Proof.* Given a pair of adjacent vertices such that $X_i \to X_j$, suppose there exists $X_k \in \mathbf{X} \setminus \{X_i, X_j\}$ such that $X_k \to X_j$. We consider two cases: (i) $X_i$ is adjacent to $X_k$ and (ii) $X_i$ is non-adjacent to $X_k$: (i) If $X_i$ is adjacent to $X_k$, then $X_i, X_j, X_k$ form a triangular structure where $X_i \to X_j \leftarrow X_k$. According to Theorem 5, we can identify that $X_j$ is the vertex with an indegree of 2, confirming that $X_i \to X_j$; (ii) If $X_i$ is non-adjacent to $X_k$, then $X_i, X_j, X_k$ form a collider structure $X_i \to X_j \leftarrow X_k$. This structure can be identified according to Theorem 6, which completes the proof. $\qquad \square$

## B.5  Proof of Lemma 3, 4 and 5

We first introduce some notations and necessary lemmas concerning the limits of the local PGF.

**Notation.** For simplify, we define $G_{\epsilon_i}^*(\mathbf{z}_{(i)}) := G_{\epsilon_i}\left(z_i \times \prod_{j \in Ch(i)} G_{i,j}(\mathbf{z}_{(j)})\right)$ as the component involving $\epsilon_i$ of PGF. According to Theorem A.1, we have $G_{\mathbf{X}}(\mathbf{z}) = \prod_{i \in [n]} G_{\epsilon_i}^*(\mathbf{z}_{(i)})$.

**Lemma 6.** *Considering the limit of function $G_{\epsilon_i}^*(\mathbf{z}_{(i)})$ as $z_i \to 0$, we have $\lim_{z_i \to 0} G_{\epsilon_i}^*(\mathbf{z}_{(i)}) = e^{-\mu_i}$.*

*Proof.* Since $G_{\epsilon_i}^*(\mathbf{z}_{(i)}) = G_{\epsilon_i}\left(z_i \times \prod_{j \in Ch(i)} G_{i,j}(\mathbf{z}_{(j)})\right)$, where $G_{\epsilon_i}(\cdot)$ is the PGF of Poisson noise $\epsilon_i$, we have:

$$G_{\epsilon_i}^*(\mathbf{z}_{(i)}) = \lim_{z_i \to 0} G_{\epsilon_i}\left(z_i \times \prod_{j \in Ch(i)} G_{i,j}(\mathbf{z}_{(j)})\right) = G_{\epsilon_i}(0) = e^{-\mu_i}, \tag{B.18}$$

which completes the proof. $\qquad\square$

**Lemma 7.** *Considering the limit of function $G_{i,j}(\mathbf{z}_{(j)})$ as $z_j \to 0$, we have $\lim_{z_j \to 0} G_{i,j}(\mathbf{z}_{(j)}) = 1 - \alpha_{i,j}$.*

*Proof.* Since $G_{i,j}(\mathbf{z}_{(j)}) = G_{B(\alpha_{i,j})}\left(z_j \times \prod_{k \in Ch(j)} G_{j,k}(\mathbf{z}_{(k)})\right)$, where $G_{B(\alpha_{i,j})}(\cdot)$ is the PGF of Bernoulli distribution with parameter $\alpha_{i,j}$, we have:

$$\lim_{z_j \to 0} G_{B(\alpha_{i,j})}\left(z_j \times \prod_{k \in Ch(j)} G_{j,k}(\mathbf{z}_{(k)})\right) = G_{B(\alpha_{i,j})}(0) = 1 - \alpha_{i,j}, \tag{B.19}$$

which completes the proof. $\qquad\square$

### B.5.1 Proof of Lemma 3

*Proof.* According to the definition of local PGF, we have

$$G_{\mathbf{X}}^{\{i,j\}}(\mathbf{z}) = \lim_{\mathbf{z}_{\setminus\{i,j\}} \to \mathbf{0}} G_{\mathbf{X}}(\mathbf{z}) = \lim_{\mathbf{z}_{\setminus\{i,j\}} \to \mathbf{0}} \prod_{l \in [d]} G_{\epsilon_l}^*(\mathbf{z}_{(l)}) \tag{B.20}$$

According to Lemma 6, we have $\lim_{z_l \to 0} G_{\epsilon_l}^*(\mathbf{z}_{(l)}) = e^{-\mu_l}$ for $l \neq i, j$, consequently, we have:

$$\begin{aligned}
G_{\mathbf{X}}^{\{i,j\}}(\mathbf{z}) &= \lim_{\mathbf{z}_{\setminus\{i,j\}} \to \mathbf{0}} G_{\epsilon_i}^*(\mathbf{z}_{(i)}) G_{\epsilon_j}^*(\mathbf{z}_{(j)}) \prod_{l \in [d] \setminus \{i,j\}} G_{\epsilon_l}^*(\mathbf{z}_{(l)}) \\
&= \lim_{\mathbf{z}_{\setminus\{i,j\}} \to \mathbf{0}} G_{\epsilon_i}^*(\mathbf{z}_{(i)}) G_{\epsilon_j}^*(\mathbf{z}_{(j)}) \prod_{l \in [d] \setminus \{i,j\}} \lim_{\mathbf{z}_{\setminus\{i,j\}} \to \mathbf{0}} G_{\epsilon_l}^*(\mathbf{z}_{(l)}) \\
&= \lim_{\mathbf{z}_{\setminus\{i,j\}} \to \mathbf{0}} G_{\epsilon_i}^*(\mathbf{z}_{(i)}) G_{\epsilon_j}^*(\mathbf{z}_{(j)}) \prod_{l \in [d] \setminus \{i,j\}} e^{-\mu_l}
\end{aligned} \tag{B.21}$$

**If part.** Suppose vertex $i$ is not adjacent to vertex $j$, we have:

$$\lim_{\mathbf{z}_{\setminus\{i,j\}} \to \mathbf{0}} G_{\epsilon_i}^*(\mathbf{z}_{(i)}) = \lim_{\mathbf{z}_{\setminus\{i,j\}} \to \mathbf{0}} G_{\epsilon_i}\left(z_i \times \prod_{k \in Ch(i)} G_{i,k}(\mathbf{z}_{(k)})\right). \tag{B.22}$$

Since $j \notin Ch(i)$, therefore $z_k \in \mathbf{z}_{\setminus\{i,j\}}$ for all $k \in Ch(i)$, we have:

$$\lim_{\mathbf{z}_{\setminus\{i,j\}} \to \mathbf{0}} G_{\epsilon_i}^*(\mathbf{z}_{(i)}) = G_{\epsilon_i}\left(z_i \times \prod_{k \in Ch(i)} \lim_{\mathbf{z}_{\setminus\{i,j\}} \to \mathbf{0}} G_{i,k}(\mathbf{z}_{(k)})\right), \tag{B.23}$$

where

$$\lim_{\mathbf{z}_{\setminus\{i,j\}} \to \mathbf{0}} G_{i,k}(\mathbf{z}_{(k)}) = \lim_{\mathbf{z}_{\setminus\{i,j\}} \to \mathbf{0}} G_{B(\alpha_{i,k})}\left(z_k \times \prod_{l \in Ch(k)} G_{k,l}(\mathbf{z}_{(l)})\right) = G_{B(\alpha_{i,k})}(0) \tag{B.24}$$

Substituting Eq.B.24 into Eq. B.23, we have

$$\begin{aligned}
\lim_{\mathbf{z}_{\setminus\{i,j\}} \to \mathbf{0}} G_{\epsilon_i}^*(\mathbf{z}_{(i)}) &= G_{\epsilon_i}\left(z_i \times \prod_{k \in Ch(i)} G_{B(\alpha_{i,k})}(0)\right) \\
&= \exp\left\{\mu_i z_i \times \prod_{k \in Ch(i)} (1 - \alpha_{i,k}) - \mu_i\right\}.
\end{aligned} \tag{B.25}$$

The derivation of $\lim_{\mathbf{z}_{\backslash\{i,j\}}\to\mathbf{0}} G^*_{\epsilon_j}(\mathbf{z}_{(j)})$ is similar to $\lim_{\mathbf{z}_{\backslash\{i,j\}}\to\mathbf{0}} G^*_{\epsilon_i}(\mathbf{z}_{(i)})$, it follows that,

$$\lim_{\mathbf{z}_{\backslash\{i,j\}}\to\mathbf{0}} G^*_{\epsilon_j}(\mathbf{z}_{(j)}) = \exp\left\{\mu_j z_j \times \prod_{k\in Ch(j)}(1-\alpha_{j,k}) - \mu_j\right\}. \tag{B.26}$$

Substituting Eq. B.25 and Eq. B.26 into Eq.B.21, we have

$$G_{\mathbf{X}}^{\{i,j\}}(\mathbf{z}) = \lim_{\mathbf{z}_{\backslash\{i,j\}}\to\mathbf{0}} G^*_{\epsilon_i}(\mathbf{z}_{(i)}) G^*_{\epsilon_j}(\mathbf{z}_{(j)}) \prod_{l\in[d]\backslash\{i,j\}} e^{-\mu_l}$$

$$= \exp\left\{\mu_i z_i \times \prod_{k\in Ch(i)}(1-\alpha_{i,k}) + \mu_j z_j \times \prod_{k\in Ch(j)}(1-\alpha_{j,k}) - \sum_{l\in[d]}\mu_l\right\}. \tag{B.27}$$

Taking the logarithm of Eq. B.27, we obtain:

$$\log G_{\mathbf{X}}^{\{i,j\}}(\mathbf{z}) = \mu_i z_i \times \prod_{k\in Ch(i)}(1-\alpha_{i,k}) + \mu_j z_j \times \prod_{k\in Ch(j)}(1-\alpha_{j,k}) - \sum_{l\in[d]}\mu_l. \tag{B.28}$$

Consequently, Taking the partial derivative of Eq. B.28, we finally have:

$$\frac{\partial \log G_{\mathbf{X}}^{\{i,j\}}(\mathbf{z})}{\partial z_i} = \mu_i \times \prod_{k\in Ch(i)}(1-\alpha_{i,k}), \tag{B.29}$$

$$\frac{\partial^2 \log G_{\mathbf{X}}^{\{i,j\}}(\mathbf{z})}{\partial z_i \partial z_j} = 0, \tag{B.30}$$

which completes the proof of the if part.

**Only if part.** We prove by contradiction, i.e., suppose that $j$ is a child vertex of vertex $i$, we aim to prove that $\frac{\partial^2 \log G_{\mathbf{X}}^{\{i,j\}}(\mathbf{z})}{\partial z_i \partial z_j} \neq 0$.

We first consider $G^*_{\epsilon_i}(\mathbf{z}_{(i)})$ in Eq. B.21, since $j\in Ch(i)$, we have

$$\lim_{\mathbf{z}_{\backslash\{i,j\}}\to\mathbf{0}} G^*_{\epsilon_i}(\mathbf{z}_{(i)}) = \lim_{\mathbf{z}_{\backslash\{i,j\}}\to\mathbf{0}} G_{\epsilon_i}\left(z_i \times G_{i,j}(\mathbf{z}_{(j)}) \times \prod_{k\in Ch(i)\backslash j} G_{i,k}(\mathbf{z}_{(k)})\right)$$

$$= G_{\epsilon_i}\left(z_i \times \lim_{\mathbf{z}_{\backslash\{i,j\}}\to\mathbf{0}} G_{i,j}(\mathbf{z}_{(j)}) \times \prod_{k\in Ch(i)\backslash j} \lim_{\mathbf{z}_{\backslash\{i,j\}}\to\mathbf{0}} G_{i,k}(\mathbf{z}_{(k)})\right) \tag{B.31}$$

$$= G_{\epsilon_i}\left(z_i \times \lim_{\mathbf{z}_{\backslash\{i,j\}}\to\mathbf{0}} G_{i,j}(\mathbf{z}_{(j)}) \times \prod_{k\in Ch(i)\backslash j} G_{B(\alpha_{i,j})}(0)\right).$$

We next focus on $\lim_{\mathbf{z}_{\backslash\{i,j\}}\to\mathbf{0}} G_{i,j}(\mathbf{z}_{(j)})$:

$$\lim_{\mathbf{z}_{\backslash\{i,j\}}\to\mathbf{0}} G_{i,j}(\mathbf{z}_{(j)}) = \lim_{\mathbf{z}_{\backslash\{i,j\}}\to\mathbf{0}} G_{B(\alpha_{i,j})}\left(z_j \times \prod_{k\in Ch(j)} G_{j,k}(\mathbf{z}_{(k)})\right)$$

$$= G_{B(\alpha_{i,j})}\left(z_j \times \prod_{k\in Ch(j)} \lim_{\mathbf{z}_{\backslash\{i,j\}}\to\mathbf{0}} G_{j,k}(\mathbf{z}_{(k)})\right) \tag{B.32}$$

$$= G_{B(\alpha_{i,j})}\left(z_j \times \prod_{k\in Ch(j)} G_{B(\alpha_{j,k})}(0)\right)$$

$$= 1 - \alpha_{i,j} + \alpha_{i,j} z_j \prod_{k\in Ch(j)} G_{B(\alpha_{j,k})}(0).$$

Substituting Eq. B.32 into Eq. B.31, we have:

$$\lim_{\mathbf{z}_{\backslash\{i,j\}}\to\mathbf{0}} G^*_{\epsilon_i}(\mathbf{z}_{(i)}) = G_{\epsilon_i}\left(z_i \times \left(1 - \alpha_{i,j} + \alpha_{i,j} z_j \prod_{k\in Ch(j)} G_{B(\alpha_{j,k})}(0)\right) \times \prod_{k\in Ch(i)\backslash j} G_{B(\alpha_{i,k})}(0)\right)$$

$$= \exp\left\{\prod_{k\in Ch(i)}(1-\alpha_{i,j})\mu_i z_i + \prod_{k\in Ch(i)\backslash j}(1-\alpha_{i,k}) \prod_{k\in Ch(j)}(1-\alpha_{j,k})\mu_i \alpha_{i,j} z_i z_j - \mu_i\right\}. \tag{B.33}$$

Next, we consider $G^*_{\epsilon_j}(\mathbf{z}_{(j)})$ in Eq. B.34, as $i \notin \mathrm{Ch}(j)$, therefore $z_k \in \mathbf{z}_{\backslash\{i,j\}}$ for all $k \in \mathrm{Ch}(j)$. Similar to the if part, we have

$$\lim_{\mathbf{z}_{\backslash\{i,j\}} \to \mathbf{0}} G^*_{\epsilon_j}(\mathbf{z}_{(j)}) = \exp\left\{ \prod_{k \in Ch(j)} (1 - \alpha_{j,k})\mu_j z_j - \mu_j \right\}. \tag{B.34}$$

Substituting Eq. B.33 and Eq. B.34 into Eq. B.21 and taking the logarithm of it, we have

$$\log G^{\{i,j\}}_{\mathbf{X}}(\mathbf{z}) = \prod_{k \in Ch(i)} (1 - \alpha_{i,j})\mu_i z_i + \prod_{k \in Ch(j)} (1 - \alpha_{j,k})\mu_j z_j$$
$$+ \prod_{k \in Ch(i)\backslash j} (1 - \alpha_{i,k}) \prod_{k \in Ch(j)} (1 - \alpha_{j,k})\mu_i \alpha_{i,j} z_i z_j - \sum_{l \in [d]} \mu_l. \tag{B.35}$$

Since the presence of $z_i z_j$, we have

$$\frac{\partial^2 \log G^{\{i,j\}}_{\mathbf{X}}(\mathbf{z})}{\partial z_i \partial z_j} = \prod_{k \in Ch(i)\backslash j} (1 - \alpha_{i,k}) \prod_{k \in Ch(j)} (1 - \alpha_{j,k})\mu_i \neq 0, \tag{B.36}$$

which completes the proof of the only if part. $\qquad\square$

### B.5.2 Proof of Lemma 4

*Proof.* The proof of Lemma 4 is similar to that of Lemma 3. We aim to show that the component $z_i z_j z_k$ is absent in $\log G^{\{i,j,k\}}_{\mathbf{X}}(\mathbf{z})$ if and only if the vertices $i, j, k$ form the structure $i \to j \leftarrow k$.

According to the definition of local PGF, we have

$$G^{\{i,j,k\}}_{\mathbf{X}}(\mathbf{z}) = \lim_{\mathbf{z}_{\backslash\{i,j,k\}} \to \mathbf{0}} G_{\mathbf{X}}(\mathbf{z}) = \lim_{\mathbf{z}_{\backslash\{i,j,k\}} \to \mathbf{0}} \prod_{l \in [d]} G^*_{\epsilon_l}(\mathbf{z}_{(l)}) \tag{B.37}$$

According to Lemma 6, we have $\lim_{z_l \to 0} G^*_{\epsilon_l}(\mathbf{z}_{(l)}) = e^{-\mu_l}$ for $l \neq i, j, k$, consequently, we have:

$$G^{\{i,j,k\}}_{\mathbf{X}}(\mathbf{z}) = \lim_{\mathbf{z}_{\backslash\{i,j,k\}} \to \mathbf{0}} G^*_{\epsilon_i}(\mathbf{z}_{(i)})G^*_{\epsilon_j}(\mathbf{z}_{(j)})G^*_{\epsilon_k}(\mathbf{z}_{(k)}) \prod_{l \in [d]\backslash\{i,j,k\}} G^*_{\epsilon_l}(\mathbf{z}_{(l)}) \tag{B.38}$$

$$= \lim_{\mathbf{z}_{\backslash\{i,j,k\}} \to \mathbf{0}} G^*_{\epsilon_i}(\mathbf{z}_{(i)})G^*_{\epsilon_j}(\mathbf{z}_{(j)})G^*_{\epsilon_k}(\mathbf{z}_{(k)}) \prod_{l \in [d]\backslash\{i,j,k\}} e^{-\mu_l} \tag{B.39}$$

Taking the logarithm of Eq. B.39 we have:

$$\log G^{\{i,j,k\}}_{\mathbf{X}}(\mathbf{z}) = \lim_{\mathbf{z}_{\backslash\{i,j,k\}} \to \mathbf{0}} \left( \log G^*_{\epsilon_i}(\mathbf{z}_{(i)}) + \log G^*_{\epsilon_j}(\mathbf{z}_{(j)}) + \log G^*_{\epsilon_k}(\mathbf{z}_{(k)}) \right) - \sum_{l \in [d]\backslash\{i,j,k\}} \mu_l \tag{B.40}$$

**If part**   Suppose $i, j, k$ form the structure $i \to j \leftarrow k$. we aim to prove that $\frac{\partial^2 \log G^{\{i,j,k\}}_{\mathbf{X}}(\mathbf{z})}{\partial z_i \partial z_k} = 0$ by showing that none of the functions $\lim_{\mathbf{z}_{\backslash\{i,j,k\}}} G^*_{\epsilon_i}(\mathbf{z}_{(i)})$, $\lim_{\mathbf{z}_{\backslash\{i,j,k\}}} G^*_{\epsilon_j}(\mathbf{z}_{(j)})$, $\lim_{\mathbf{z}_{\backslash\{i,j,k\}}} G^*_{\epsilon_k}(\mathbf{z}_{(k)})$ in Eq. B.40 involve the component $z_i z_k$, which implies that there is no a directed path between $i$ and $k$.

Since the $j$ is the child of $i$ and $k$, and $j$ has no child, similar to proof of Lemma 3, we have:

$$\lim_{\mathbf{z}_{\backslash\{i,j,k\}} \to \mathbf{0}} G^*_{\epsilon_j}(\mathbf{z}_{(j)}) = \exp\left\{ \mu_j z_j \times \prod_{k \in Ch(j)} (1 - \alpha_{j,k}) - \mu_j \right\}, \tag{B.41}$$

$$\lim_{\mathbf{z}_{\backslash\{i,j,k\}} \to \mathbf{0}} G^*_{\epsilon_i}(\mathbf{z}_{(i)}) = \exp\left\{ \prod_{l_1 \in Ch(i)} (1 - \alpha_{i,l_1})\mu_i z_i + \prod_{l_1 \in Ch(i)\backslash j} (1 - \alpha_{i,l_1}) \prod_{l_1 \in Ch(j)} (1 - \alpha_{j,l_1})\mu_i \alpha_{i,j} z_i z_j - \mu_i \right\}, \tag{B.42}$$

$$\lim_{\mathbf{z}_{\backslash\{i,j,k\}} \to \mathbf{0}} G^*_{\epsilon_k}(\mathbf{z}_{(k)}) = \exp\left\{ \prod_{l_2 \in Ch(k)} (1 - \alpha_{k,l_2})\mu_k z_k + \prod_{l_2 \in Ch(k)\backslash j} (1 - \alpha_{k,l_2}) \prod_{l_2 \in Ch(j)} (1 - \alpha_{j,l_2})\mu_k \alpha_{k,j} z_k z_j - \mu_k \right\}. \tag{B.43}$$

Since none of Eq. B.41, Eq. B.42, Eq. B.43 contain the term involving $z_i z_k$, the partial derivatives of the logarithms of these equations with respect to $z_i$ and $z_k$ are all zero. Therefore we have :

$$\frac{\partial^2 \log G_{\mathbf{X}}^{\{i,j,k\}}(\mathbf{z})}{\partial z_i \partial z_k} = 0, \tag{B.44}$$

which completes the proof of the if part.

**Only if part**   We prove by contradiction. Suppose that $i, j, k$ do not form the structure $i \to j \leftarrow k$. This implies $i, j, k$ form either (i) the chain structure $i \to j \to k$ or (ii) the fork structure $i \leftarrow j \to k$. We aim to prove that $\frac{\partial^2 \log G_{\mathbf{X}}^{\{i,j,k\}}(\mathbf{z})}{\partial z_i \partial z_k} \neq 0$.

We first discuss the case (i). If $i, j, k$ form the chain structure $i \to j \to k$, we prove that $\frac{\partial^2 \log G_{\mathbf{X}}^{\{i,j,k\}}(\mathbf{z})}{\partial z_i \partial z_k} \neq 0$ by showing that the $\lim_{\mathbf{z}\setminus\{i,j,k\}\to\mathbf{0}} \log G_{\epsilon_i}^*(\mathbf{z}_{(i)})$ in Eq. B.40 contains the component $z_i z_j z_k$.

According the Theorem 1 and Lemma 7, we have:

$$\lim_{\mathbf{z}\setminus\{i,j,k\}\to\mathbf{0}} G_{\epsilon_i}^*(\mathbf{z}_{(i)}) = \lim_{\mathbf{z}\setminus\{i,j,k\}\to\mathbf{0}} G_{\epsilon_i}\left( z_i \times \prod_{j\in Ch(i)} G_{i,j}(\mathbf{z}_{(j)}) \right)$$

$$= G_{\epsilon_i}\left( z_i \times \lim_{\mathbf{z}\setminus\{i,j,k\}\to\mathbf{0}} G_{i,j}(\mathbf{z}_{(j)}) \times \prod_{l\in Ch(i)\setminus\{j\}} \lim_{\mathbf{z}\setminus\{i,j,k\}\to\mathbf{0}} G_{i,l}(\mathbf{z}_{(l)}) \right)$$

$$= G_{\epsilon_i}\left( z_i \times \lim_{\mathbf{z}\setminus\{i,j,k\}\to\mathbf{0}} G_{i,j}(\mathbf{z}_{(j)}) \times \prod_{l\in Ch(i)\setminus\{j\}} (1 - \alpha_{i,l}) \right). \tag{B.45}$$

Consequently, we expand the $G_{i,j}(\mathbf{z}_j)$ and apply the Lemma 7, we have

$$\lim_{\mathbf{z}\setminus\{i,j,k\}\to\mathbf{0}} G_{i,j}(\mathbf{z}_{(j)})$$

$$= G_{B(\alpha_{i,j})}\left( z_j \times G_{B(\alpha_{j,k})}\left( z_k \times \prod_{n\in Ch(k)} \lim_{\mathbf{z}\setminus\{i,j,k\}\to\mathbf{0}} G_{k,n}(\mathbf{z}_{(n)}) \right) \times \prod_{m\in Ch(j)\setminus\{k\}} \lim_{\mathbf{z}\setminus\{i,j,k\}\to\mathbf{0}} G_{j,m}(\mathbf{z}_{(m)}) \right)$$

$$= G_{B(\alpha_{i,j})}\left( z_j \times G_{B(\alpha_{j,k})}\left( z_k \times \prod_{n\in Ch(k)} (1 - \alpha_{k,n}) \right) \times \prod_{m\in Ch(j)\setminus\{k\}} (1 - \alpha_{j,m}) \right)$$

$$= G_{B(\alpha_{i,j})}\left( \left( (1 - \alpha_{j,k})z_j + \alpha_{j,k}z_j z_k \underbrace{\prod_{n\in Ch(k)}(1 - \alpha_{k,n})}_{:=\beta_1} \right) \underbrace{\prod_{m\in Ch(j)\setminus\{k\}} (1 - \alpha_{j,m})}_{:=\beta_2} \right)$$

$$= G_{B(\alpha_{i,j})}((1 - \alpha_{j,k})\beta_2 z_j + \alpha_{j,k}\beta_1\beta_2 z_j z_k)$$

$$= 1 - \alpha_{i,j} + (1 - \alpha_{j,k})\alpha_{i,j}\beta_2 z_j + \alpha_{i,j}\alpha_{j,k}\beta_1\beta_2 z_j z_k. \tag{B.46}$$

For simplicity, the constant coefficients in the equation are replaced with $\beta_1, \beta_2$. Substituting Eq. B.46 into Eq. B.45, we have:

$$\lim_{\mathbf{z}\setminus\{i,j,k\}\to\mathbf{0}} G_{\epsilon_i}^*(\mathbf{z}_{(i)}) = G_{\epsilon_i}\left( z_i \times [1 - \alpha_{i,j} + (1 - \alpha_{j,k})\alpha_{i,j}\beta_2 z_j + \alpha_{i,j}\alpha_{j,k}\beta_1\beta_2 z_j z_k] \times \prod_{l\in Ch(i)\setminus\{j\}}(1 - \alpha_{i,l}) \right)$$

$$= G_{\epsilon_i}\left( [(1 - \alpha_{i,j})z_i + (1 - \alpha_{j,k})\alpha_{i,j}\beta_2 z_i z_j + \alpha_{i,j}\alpha_{j,k}\beta_1\beta_2 z_i z_j z_k] \times \prod_{l\in Ch(i)\setminus\{j\}}(1 - \alpha_{i,l}) \right). \tag{B.47}$$

Clearly, the equation contains the term involving $z_i z_j z_k$. We have:

$$\frac{\partial^2 \log G_{\mathbf{X}}^{\{i,j,k\}}(\mathbf{z})}{\partial z_i \partial z_k} = \alpha_{i,j}\alpha_{j,k}\beta_1\beta_2 \prod_{l\in Ch(i)\setminus\{j\}}(1 - \alpha_{i,l}) \times z_j \neq 0. \tag{B.48}$$

Then we consider the case (ii). If $i, j, k$ form the fork structure $i \leftarrow j \to k$, , we prove that $\frac{\partial^2 \log G_{\mathbf{X}}^{\{i,j,k\}}(\mathbf{z})}{\partial z_i \partial z_k} \neq 0$ by showing that the $\lim_{\mathbf{z}\setminus\{i,j,k\}\to\mathbf{0}} \log G_{\epsilon_j}^*(\mathbf{z}_{(j)})$ contain the component $z_i z_j z_k$.

According the Theorem 1 and Lemma 7, we have:

$$\lim_{\mathbf{z}_{\backslash\{i,j,k\}}\to\mathbf{0}} G^*_{\epsilon_i}(\mathbf{z}_{(i)}) = \lim_{\mathbf{z}_{\backslash\{i,j,k\}}\to\mathbf{0}} G_{\epsilon_i}\left(z_i \times \prod_{j\in Ch(i)} G_{i,j}(\mathbf{z}_{(j)})\right)$$

$$= G_{\epsilon_i}\left(z_i \times \lim_{\mathbf{z}_{\backslash\{i,j,k\}}\to\mathbf{0}} G_{i,j}(\mathbf{z}_{(j)}) \times \lim_{\mathbf{z}_{\backslash\{i,j,k\}}\to\mathbf{0}} G_{i,k}(\mathbf{z}_{(k)}) \times \prod_{l\in Ch(i)\backslash\{i,j\}} \lim_{\mathbf{z}_{\backslash\{i,j,k\}}\to\mathbf{0}} G_{i,l}(\mathbf{z}_{(l)})\right)$$

$$= G_{\epsilon_i}\left(z_i \times \lim_{\mathbf{z}_{\backslash\{i,j,k\}}\to\mathbf{0}} G_{i,j}(\mathbf{z}_{(j)}) \times \lim_{\mathbf{z}_{\backslash\{i,j,k\}}\to\mathbf{0}} G_{i,k}(\mathbf{z}_{(k)}) \times \prod_{l\in Ch(i)\backslash\{i,j\}}(1-\alpha_{i,l})\right).$$

$$(B.49)$$

Consequently, we expand the $G_{i,j}(\mathbf{z}_j)$, $G_{i,k}(\mathbf{z}_k)$ and apply the Lemma 7:

$$\lim_{\mathbf{z}_{\backslash\{i,j,k\}}\to\mathbf{0}} G_{i,j}(\mathbf{z}_{(j)}) = G_{B(\alpha_{i,j})}\left(z_j \times \lim_{\mathbf{z}_{\backslash\{i,j,k\}}\to\mathbf{0}} \prod_{m\in Ch(j)} G_{j,m}(\mathbf{z}_{(m)})\right)$$

$$= G_{B(\alpha_{i,j})}\left(z_j \times \prod_{m\in Ch(j)}(1-\alpha_{i,m})\right) \qquad (B.50)$$

$$= 1 - \alpha_{i,j} + \underbrace{\alpha_{i,j}\prod_{m\in Ch(j)}(1-\alpha_{i,m})}_{:=\beta_1} z_j,$$

and

$$\lim_{\mathbf{z}_{\backslash\{i,j,k\}}\to\mathbf{0}} G_{i,k}(\mathbf{z}_{(k)}) = G_{B(\alpha_{i,k})}\left(z_k \times \lim_{\mathbf{z}_{\backslash\{i,j,k\}}\to\mathbf{0}} \prod_{n\in Ch(k)} G_{k,n}(\mathbf{z}_{(n)})\right)$$

$$= G_{B(\alpha_{i,k})}\left(z_k \times \prod_{n\in Ch(k)}(1-\alpha_{k,n})\right) \qquad (B.51)$$

$$= 1 - \alpha_{i,k} + \underbrace{\alpha_{i,k}\prod_{n\in Ch(k)}(1-\alpha_{k,n})}_{:=\beta_2} z_k.$$

For simplicity, the constant coefficients in the equation are replaced with $\beta_1, \beta_2$. Substituting Eq. B.50 and Eq. B.51 into Eq. B.49, we have

$$\lim_{\mathbf{z}_{\backslash\{i,j,k\}}\to\mathbf{0}} G^*_{\epsilon_i}(\mathbf{z}_{(i)}) = G_{\epsilon_i}\left(z_i \times (1-\alpha_{i,j}+\beta_1 z_j) \times (1-\alpha_{i,k}+\beta_2 z_k) \times \prod_{l\in Ch(i)\backslash\{i,j,k\}}(1-\alpha_{i,l})\right).$$

$$(B.52)$$

Clearly, the equation contains the term $z_i z_j z_k$, Then we have

$$\frac{\partial^2 \log G_{\mathbf{X}}^{\{i,j,k\}}(\mathbf{z})}{\partial z_i \partial z_k} = \beta_1 \beta_2 \prod_{l\in Ch(i)\backslash\{i,j,k\}}(1-\alpha_{i,l}) \times z_j \neq 0, \qquad (B.53)$$

which completes the proof of the only if part. $\qquad\square$

### B.5.3   Proof of Lemma 5

*Proof.* According to the Theorem 3, we have

$$\log G_{\mathbf{X}}^{\{i,j,k\}}(\mathbf{z}) = \log \prod_{l\in\{i,j,k\}} \exp\left\{\mu_l \times \left(T_{X_l}^{\{i,j,k\}}(1)-1\right)\right\} \prod_{l\in[d]\backslash\{i,j,k\}} \exp\{-\mu_l\}$$

$$= \mu_i\left(T_{X_i}^{\{i,j,k\}}(1)-1\right) + \mu_j\left(T_{X_j}^{\{i,j,k\}}(1)-1\right) + \mu_k\left(T_{X_k}^{\{i,j,k\}}(1)-1\right) - \sum_{l\in[d]\backslash\{i,j,k\}}\mu_l$$

$$= \mu_i T_{X_i}^{\{i,j,k\}}(1) + \mu_j T_{X_j}^{\{i,j,k\}}(1) + \mu_k T_{X_k}^{\{i,j,k\}}(1) - \sum_{l\in[d]}\mu_l.$$

$$(B.54)$$

**If part**   Suppose that $i,j,k$ form the structure $i \to k \leftarrow j$ and $i \to j$, where vertex $i$ has two directed path lead to vertex $j$. We aim to demonstrate that $T_{X_i}^{\{i,j,k\}}(1)$ contain the term involving $z_i z_j z_k^2$ such that $\frac{\partial^2 z_i z_j z_k^2}{\partial z_k^2} = 2z_i z_j \neq 0$. Consequently, it follows that $\frac{\partial^2 \log G_{\mathbf{X}}^{\{i,j,k\}}(\mathbf{z})}{\partial z_k^2} \neq 0$.

According to the Theorem 3, $T_{X_i}^{\{i,j,k\}}(1)$ is expressed as follows:

$$T_{X_i}^{\{i,j,k\}}(1) = z_i \sum_{\mathbf{s} \in \{0,1\}^{|Ch_{\{i,j,k\}}(i)|}} \prod_{j \in Ch_{\{i,j,k\}}(i)} \alpha_{i,j}^{s_j} T_{X_j}^{\{i,j,k\}}(s_j) \prod_{j \in Ch(i) \setminus Ch_{\{i,j,k\}}(i)} (1 - \alpha_{i,j}),$$

(B.55)

where $Ch_{\{i,j,k\}}(i) = \{j, k\}$ in this structure. For simplicity, let $\beta_1 = \prod_{j \in Ch(i) \setminus Ch_{\{i,j,k\}}(i)} (1 - \alpha_{i,j})$,
we have:

$$T_{X_i}^{\{i,j,k\}}(1) = \beta_1 z_i \sum_{\mathbf{s} \in \{0,1\}^2} \alpha_{i,j}^{s_j} T_{X_j}^{\{i,j,k\}}(s_j) \alpha_{i,k}^{s_k} T_{X_k}^{\{i,j,k\}}(s_k).$$

(B.56)

We consider the term in B.56 where both $s_j = 1$ and $s_k = 1$, which is given by:

$$\alpha_{i,j}^{s_j=1} T_{X_j}^{\{i,j,k\}}(1) \alpha_{i,k}^{s_k=1} T_{X_k}^{\{i,j,k\}}(1) = \alpha_{i,j} \alpha_{i,k} T_{X_j}^{\{i,j,k\}}(1) T_{X_k}^{\{i,j,k\}}(1)$$

(B.57)

Consequently, since vertex $j$ has child $k$ and vertex $k$ has no child in this sturcture, we have:

$$T_{X_k}^{\{i,j,k\}}(1) = z_k \prod_{l \in Ch(k)} (1 - \alpha_{k,l})$$

$$T_{X_j}^{\{i,j,k\}}(1) = z_j \sum_{\mathbf{s} \in \{0,1\}} \alpha_{j,k}^{s_k} T_{X_k}^{\{i,j,k\}}(s_k) \prod_{l \in Ch(j) \setminus Ch_{\mathbf{L}}(j)} (1 - \alpha_{j,l})$$

$$= \left( z_j \alpha_{j,k} T_{X_k}^{\{i,j,k\}}(1) + (1 - \alpha_{j,k}) T_{X_k}^{\{i,j,k\}}(0) \right) \prod_{l \in Ch(j) \setminus Ch_{\mathbf{L}}(j)} (1 - \alpha_{j,l})$$

(B.58)

$$= \left( 1 - \alpha_{j,k} + \alpha_{j,k} z_j z_k \prod_{l \in Ch(k)} (1 - \alpha_{k,l}) \right) \prod_{l \in Ch(j) \setminus Ch_{\mathbf{L}}(j)} (1 - \alpha_{j,l}).$$

By combining Eq. B.56, B.57, and B.58, we can observe that $z_i z_j z_k^2$ is present. Therefore $\frac{\partial^2 \log G_{\mathbf{X}}^{\{i,j,k\}}(\mathbf{z})}{\partial z_k^2} \neq 0$, this complete the if part.

**Only if part**  We prove by contradiction. Suppose that vertices $i, j, k$ do not form the triangular structure that $i \to k \leftarrow j$ and $i$ is adjacent to $j$, we aim to prove that $\frac{\partial^2 \log G_{\mathbf{X}}^{\{i,j,k\}}(\mathbf{z})}{\partial z_k^2} = 0$. This setup implies two cases: (i) $i, j, k$ do not form a triangular structure, (ii) vertices $i, j, k$ form the triangular structure where the vertex has the indegree of 2 is not $k$.

We first consider the case (i). When $i, j, k$ do not form a triangular structure, they could instead form a collider, chain, or fork structure. According to Lemma 4 and the intermediate results derived in its proof, in each of these structures, the local PGFs do not contain a term involving $z_i z_j z_k^2$ such that $\frac{\partial^2 \log G_{\mathbf{X}}^{\{i,j,k\}}(\mathbf{z})}{\partial z_k^2} = 0$, which creates a contradiction.

Then we consider the case (ii). Suppose that vertices $i, j, k$ form the triangular $k \to i \leftarrow j$ and $j \to k$. Recall the Eq. B.54, the $\log G_{\mathbf{X}}^{\{i,j,k\}}(\mathbf{z})$ is given by:

$$\log G_{\mathbf{X}}^{\{i,j,k\}}(\mathbf{z}) = \mu_i T_{X_i}^{\{i,j,k\}}(1) + \mu_j T_{X_j}^{\{i,j,k\}}(1) + \mu_k T_{X_k}^{\{i,j,k\}}(1) - \sum_{l \in [d]} \mu_l.$$

(B.59)

Our goal is to show that none of the terms $T_{X_i}^{\{i,j,k\}}(1)$, $T_{X_j}^{\{i,j,k\}}(1)$ and $T_{X_k}^{\{i,j,k\}}(1)$ include a component involving $z_i z_j z_k^2$.

We first consider $T_{X_i}^{\{i,j,k\}}(1)$. Since vertex $i$ has no child in this structure, we have:

$$T_{X_i}^{\{i,j,k\}}(1) = z_i \prod_{l \in Ch(i)} (1 - \alpha_{k,l}),$$

(B.60)

which only involves $z_i$.

Next, we consider the $T_{X_k}^{\{i,j,k\}}(1)$, since vertex $k$ has only one child $i$ in this structure, we have:

$$
\begin{aligned}
T_{X_k}^{\{i,j,k\}}(1) &= z_k \sum_{\mathbf{s}\in\{0,1\}} \alpha_{k,i}^{s_i} T_{X_i}^{\{i,j,k\}}(s_i) \prod_{l\in Ch(k)\backslash\{i\}} (1-\alpha_{k,l}) \\
&= \left( z_k \alpha_{k,i} T_{X_i}^{\{i,j,k\}}(1) + z_k(1-\alpha_{k,i}) T_{X_i}^{\{i,j,k\}}(0) \right) \prod_{l\in Ch(k)\backslash\{i\}} (1-\alpha_{k,l}) \\
&= \left( z_k(1-\alpha_{k,i}) + \alpha_{k,i} \prod_{l\in Ch(i)} (1-\alpha_{k,l}) z_i z_k \right) \prod_{l\in Ch(k)\backslash\{i\}} (1-\alpha_{k,l}),
\end{aligned}
\tag{B.61}
$$

which leads to terms involving $z_k$ and $z_i z_k$.

Finally, we consider $T_{X_j}^{\{i,j,k\}}(1)$. Since vertex $j$ has two children $i$ and $k$ in this structure, similar to the if part, we have:

$$
T_{X_j}^{\{i,j,k\}}(1) = z_j \sum_{\mathbf{s}\in\{0,1\}^2} \alpha_{j,i}^{s_j} T_{X_i}^{\{i,j,k\}}(s_i) \alpha_{j,k}^{s_k} T_{X_k}^{\{i,j,k\}}(s_k) \prod_{j\in Ch(i)\backslash Ch_{\{i,j,k\}}(i)} (1-\alpha_{i,j})
\tag{B.62}
$$

In analyzing the equation, we will focus on the terms $z_j \sum_{\mathbf{s}\in\{0,1\}^2} T_{X_i}^{\{i,j,k\}}(s_i) T_{X_k}^{\{i,j,k\}}(s_k)$, treating the other components as constants for simplicity. We have:

$$
\begin{aligned}
z_j \sum_{\mathbf{s}\in\{0,1\}^2} T_{X_i}^{\{i,j,k\}}(s_i) T_{X_k}^{\{i,j,k\}}(s_k) &= z_j T_{X_i}^{\{i,j,k\}}(0) T_{X_k}^{\{i,j,k\}}(0) + z_j T_{X_i}^{\{i,j,k\}}(1) T_{X_k}^{\{i,j,k\}}(0) \\
&\quad + z_j T_{X_i}^{\{i,j,k\}}(0) T_{X_k}^{\{i,j,k\}}(1) + z_j T_{X_i}^{\{i,j,k\}}(1) T_{X_k}^{\{i,j,k\}}(1)
\end{aligned}
\tag{B.63}
$$

Since $T_{X_i}^{\{i,j,k\}}(0) = T_{X_k}^{\{i,j,k\}}(0) = 1$, we have:

$$
z_j \sum_{\mathbf{s}\in\{0,1\}^2} T_{X_i}^{\{i,j,k\}}(s_i) T_{X_k}^{\{i,j,k\}}(s_k) = z_j + z_j T_{X_i}^{\{i,j,k\}}(1) + z_j T_{X_k}^{\{i,j,k\}}(1) + z_j T_{X_i}^{\{i,j,k\}}(1) T_{X_k}^{\{i,j,k\}}(1)
\tag{B.64}
$$

According to previous discussion, we have $T_{X_i}^{\{i,j,k\}}(1)$ involves $z_i$, $T_{X_k}^{\{i,j,k\}}(1)$ involves $z_k$ and $z_i z_k$. Consequently, we derive the following result:

- $z_j T_{X_i}^{\{i,j,k\}}(1)$ involves $z_i z_j$,

- $z_j T_{X_k}^{\{i,j,k\}}(1)$ involves $z_j z_k$ and $z_i z_j z_k$,

- $z_j T_{X_i}^{\{i,j,k\}}(1) T_{X_k}^{\{i,j,k\}}(1)$ involves $z_i z_j z_k$ and $z_i^2 z_j z_k$.

We can observe that $z_k$ appears only to the first power within Eq. B.61, and B.62. In other words, $\log G_{\mathbf{X}}^{\{i,j,k\}}(\mathbf{z})$ does not contain any term involving $z_k^2$. Therefore, the second partial derivative of $\log G_{\mathbf{X}}^{\{i,j,k\}}(\mathbf{z})$ with respect to $z_k$ equal to zero, which creates a contradiction. This completes the only if part. $\qquad\square$

## C  Additional Experiment Details

Each experiment reported in the main paper was conducted on a 12th Gen Intel(R) Core(TM) i3-12100 CPU with 16GB RAM, without the use of a GPU. The runtime for each experiment is provided to facilitate reproduction under similar conditions. The significance level (alpha) for the rank hypothesis test used in these experiments is set at 0.01.

**Additional Metrics**  The main paper presents the F1 scores and Structural Hamming Distance (SHD) from synthetic data experiments. This section extends these results by providing Precision, Recall, and runtime metrics for each experiment, as detailed in the following Table.4, Table. 5 and Table. 6.

Table 4: Sensitivity to Avg. Indegree Rate.

| | Recall↑ | | | | Precision↑ | | | |
|---|---|---|---|---|---|---|---|---|
| Avg. Indegree | 2.0 | 2.5 | 3.0 | 3.5 | 2.0 | 2.5 | 3.0 | 3.5 |
| Ours | $0.88 \pm 0.07$ | $0.92 \pm 0.08$ | $0.91 \pm 0.04$ | $0.92 \pm 0.05$ | $\mathbf{0.63 \pm 0.06}$ | $\mathbf{0.73 \pm 0.06}$ | $\mathbf{0.81 \pm 0.04}$ | $\mathbf{0.87 \pm 0.04}$ |
| Cumulant | $\mathbf{0.96 \pm 0.04}$ | $\mathbf{0.98 \pm 0.02}$ | $\mathbf{0.96 \pm 0.03}$ | $\mathbf{0.98 \pm 0.01}$ | $0.58 \pm 0.03$ | $0.63 \pm 0.03$ | $0.69 \pm 0.04$ | $0.72 \pm 0.04$ |
| PC | $0.64 \pm 0.20$ | $0.66 \pm 0.13$ | $0.57 \pm 0.13$ | $0.63 \pm 0.13$ | $0.56 \pm 0.16$ | $0.58 \pm 0.10$ | $0.52 \pm 0.12$ | $0.58 \pm 0.10$ |
| GES | $0.55 \pm 0.15$ | $0.56 \pm 0.11$ | $0.47 \pm 0.12$ | $0.41 \pm 0.11$ | $0.43 \pm 0.14$ | $0.43 \pm 0.11$ | $0.37 \pm 0.10$ | $0.34 \pm 0.09$ |
| OCD | $0.24 \pm 0.22$ | $0.27 \pm 0.23$ | $0.28 \pm 0.16$ | $0.36 \pm 0.14$ | $0.23 \pm 0.21$ | $0.27 \pm 0.23$ | $0.27 \pm 0.17$ | $0.37 \pm 0.14$ |

Table 5: Sensitivity to Sample Size.

| | Recall↑ | | | | Precision↑ | | | |
|---|---|---|---|---|---|---|---|---|
| Sample Size | 5000 | 15000 | 30000 | 50000 | 5000 | 15000 | 30000 | 50000 |
| Ours | $0.78 \pm 0.11$ | $0.87 \pm 0.05$ | $0.91 \pm 0.04$ | $0.92 \pm 0.05$ | $\mathbf{0.72 \pm 0.08}$ | $\mathbf{0.77 \pm 0.04}$ | $\mathbf{0.81 \pm 0.04}$ | $\mathbf{0.82 \pm 0.04}$ |
| Cumulant | $\mathbf{0.92 \pm 0.07}$ | $\mathbf{0.96 \pm 0.03}$ | $\mathbf{0.96 \pm 0.03}$ | $\mathbf{0.97 \pm 0.03}$ | $0.60 \pm 0.05$ | $0.65 \pm 0.03$ | $0.69 \pm 0.04$ | $0.68 \pm 0.04$ |
| PC | $0.43 \pm 0.10$ | $0.56 \pm 0.11$ | $0.57 \pm 0.14$ | $0.71 \pm 0.11$ | $0.46 \pm 0.13$ | $0.53 \pm 0.11$ | $0.52 \pm 0.12$ | $0.62 \pm 0.08$ |
| GES | $0.41 \pm 0.11$ | $0.48 \pm 0.20$ | $0.47 \pm 0.13$ | $0.48 \pm 0.22$ | $0.38 \pm 0.10$ | $0.41 \pm 0.20$ | $0.37 \pm 0.10$ | $0.39 \pm 0.22$ |
| OCD | $0.28 \pm 0.10$ | $0.35 \pm 0.18$ | $0.28 \pm 0.16$ | $0.39 \pm 0.21$ | $0.34 \pm 0.14$ | $0.35 \pm 0.19$ | $0.27 \pm 0.17$ | $0.36 \pm 0.20$ |

Table 6: Runtime of each method under the default setting.

| | Ours | Cumulant | PC | GES | OCD |
|---|---|---|---|---|---|
| Runtime (second) | $7.94 \pm 0.75$ | $77.07 \pm 4.94$ | $5.00 \pm 1.35$ | $6.90 \pm 1.86$ | $9216 \pm 1368$ |

## D  Additional Discussion

### D.1  Discussion on the Benefit of Local PGF

In this section, we illustrate the benefit of introducing local PGF through a toy example. First, it is important to note that, based on the closed form of the PGF, it is theoretically possible to identify causal structures using the original PGF, as the closed form encapsulates the entire graph structure. This could be achieved by exhaustively analyzing all terms present in the PGF. However, such an approach is intractable because the search space for these terms grows exponentially with the number of vertices. Additionally, the method would involve high-order differentiations, which are difficult to estimate accurately.

To illustrate this point, we provide a toy example with a causal structure of 5 vertices, as shown in Fig. 6. In this example, we focus on identifying the edge $X_4 \rightarrow X_5$. For the correct structure where $X_4 \rightarrow X_5$, the terms $C_1 \times z_1 z_2 z_3 z_4^2 z_5^2$ and $C_2 \times z_1 z_2 z_3 z_4^2 z_5^4$ exist in the PGF. Conversely, for the reverse direction $X_4 \leftarrow X_5$, the terms $C_1' \times z_1 z_2 z_3 z_4^2 z_5^2$ and $C_2' \times z_1 z_2 z_3 z_4^4 z_5^2$ appear in the PGF.

Notably, as shown in Fig. 6 (c) and (f), the term $z_1 z_2 z_3 z_4^2 z_5^2$ exists in the PGF for both structures. This is because there are always at least two directed paths from $X_1$ to each of $X_4$ and $X_5$, regardless of the direction between $X_4$ and $X_5$. This implies that taking the second derivative with respect to $z_4$ does not reveal any asymmetry between the structures. Therefore, if we want to identify the direction using the global PGF, we have to apply a test involving the third derivative that distinguishes the correct structure by showing the absence of terms with $z_4^3$ in the PGF for the correct direction.

In conclusion, when the graph structure becomes more complex and the number of paths between vertices increases, higher-order derivatives are required, making implementation challenging. Local PGF offers an alternative approach, enabling us to avoid these difficulties.

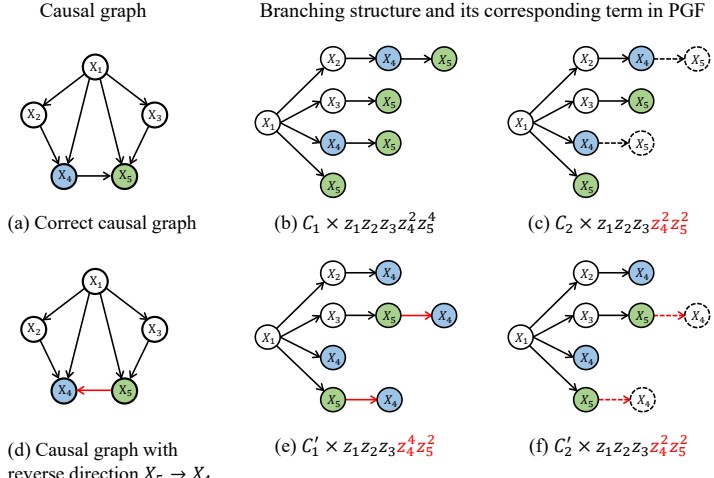

Figure 6: Illustration of causal graphs and their possible branching structures with the corresponding terms in PGF. (a) Correct causal graph where $X_4 \to X_5$. (b)-(c) Branching structures of (a) with corresponding PGF terms: (b) includes all paths; (c) excludes path from $X_4$ to $X_5$. (d) Causal graph with reversed direction $X_5 \to X_4$. (e)-(f) Branching structures of (d) with corresponding PGF terms: (e) includes all paths; (f) excludes path from $X_5$ to $X_4$.

### D.2 Discussion on the connection with the cumulant-based method

In this section, we discuss the connection between the cumulant-based method and our method, as well as the advantages of our method.

The cumulant-based method (Qiao et al. [2024b]) introduces the concept of $k$-path cumulants summation, denoted as $\tilde{\Lambda}_k(X_i \rightsquigarrow X_j)$. Here, $\tilde{\Lambda}_k(X_i \rightsquigarrow X_j) \neq 0$ indicates the existence of a common ancestor $m$ for vertices $i$ and $j$, with $k$ directed paths leading from $m$ to $j$. Such path information, specifically the number of directed paths from one vertex to another, is encoded within the PGF. If there are $k$ directed paths from vertex $m$ to $j$, then terms involving $z_m z_j^k$ will appear in the PGF. Therefore, detecting the number of directed paths to vertex $j$ is equivalent to identifying the order of $z_j$ in the PGF.

However, only the highest non-zero order of $\tilde{\Lambda}_k(X_i \rightsquigarrow X_j)$ is useful for identifying the causal direction, as it does not yield asymmetry at lower orders. For example, consider vertices $X_i$, $X_j$, and $X_m$, where $X_i \to X_j$, and there are $k$ directed paths from $X_m$ to $X_i$, and $k + p$ directed paths from $X_m$ to $X_j$ where $p \geq 1$. In this scenario, it has $\tilde{\Lambda}_k(X_i \rightsquigarrow X_j) \neq 0$ and $\tilde{\Lambda}_k(X_j \rightsquigarrow X_i) \neq 0$, and $\tilde{\Lambda}_{k+1}(X_i \rightsquigarrow X_j) \neq 0$ while $\tilde{\Lambda}_{k+1}(X_j \rightsquigarrow X_i) = 0$. The latter shows an asymmetry that the former does not. This means that the cumulant-based method must detect the highest non-zero order of $\tilde{\Lambda}_k(X_j \rightsquigarrow X_i)$, i.e., the highest order of $z_j$ in the corresponding terms in the PGF, to reveal the asymmetry. Moreover, certain unshielded collider structures, such as $X_1 \to X_2$ and $X_3 \to X_2$ in Fig. 1(a) of the paper, are non-identifiable using this method. This is because both the correct and reverse directions result in $\tilde{\Lambda}_{k=1} \neq 0$ and $\tilde{\Lambda}_{k=2} = 0$, which leads to non-identifiability.

In contrast, our method can fully leverage lower-order information to identify causal directions due to the local property of the PGF. By removing redundant directed paths involving a vertex through setting the corresponding $z$ to approach zero in the PGF, we can focus on identifying within a small local structure, thereby avoiding the need for high-order information.

## E   Broader Impacts

We identify several important societal impacts of our proposed method, including both positive and potential negative impacts:

1) This paper introduces advancements in the modeling and identification of causal relationships from count data, thereby revealing the causal mechanism of Poisson count data.

2) Inadequate data and training can result in inaccurate causal graphs, potentially leading to a misunderstanding of the underlying causal relationships. Such misunderstandings may prompt inappropriate or risky decision-making by stakeholders relying on these insights.

To mitigate the potential negative societal impacts mentioned above, we encourage research and practice to follow these instructions:

1) Integration of human oversight is recommended, where domain experts should verify and complement model outputs with their expertise to guide decision-making effectively.

2) Continuous monitoring and updating of model parameters should be implemented to align with real-world data and expert feedback, ensuring the accuracy and applicability of causal predictions.

