# OpenReview forum: "On the Identifiability of Poisson Branching Structural Causal Model Using Probability Generating Function"
_NeurIPS.cc/2024/Conference — NeurIPS 2024 spotlight_

### Official Review · Reviewer_CtKZ · 2024-06-22

**Soundness:** 4
**Presentation:** 3
**Contribution:** 3
**Rating:** 6
**Confidence:** 5

**Summary:**

In order to achieve causal structure learning on PB-SCM, this paper explores the identifiability of PB-SCM using the Probability Generating Function (PGF).  Furthermore, this paper enables the identification of the local structures by testing their corresponding component appearances in the PGF. Building on this, this paper proposes a practical algorithm for learning causal skeletons and identifying causal directions of PB-SCM using PGF.

**Strengths:**

The use of Probability Generating Function (PGF) to address the identifiability of the Poisson Branching Structural Causal Model (PB-SCM) is a novel approach. The proposed practical algorithm for learning causal skeletons and identifying causal directions of PB-SCM using PGF is well-detailed. The authors provide a clear and systematic approach, making it easier for practitioners to implement the algorithm. This paper makes several theoretical contributions and has a strong theoretical basis.

**Weaknesses:**

1. While these experiments demonstrate the effectiveness of the method, additional experiments on a wider variety of real-world datasets from different domains would strengthen the argument for the method's generalizability.
2. The symbols T and the corresponding formulas that appear in Figure 3 are explained later in the text, which may cause significant confusion for readers.
3. The term ch(i) in Formula 3 lacks a definition. It is possibly intended to be Des(i) as defined earlier in the text.

**Questions:**

1. What are the accuracies of learning the Probability Generating Function and the subsequent causal structure learning? What are the bottlenecks of the proposed method?
2. The experiments on real-world data provide insufficient information. How do other methods perform on the real datasets?
3. How does the proposed method perform on networks that are not purely Poisson Bayesian networks or on mixed networks of general Bayesian and Poisson Bayesian networks? Additionally, does this method require prior knowledge that the data originates from a non-Poisson Bayesian network?

**Limitations:**

This paper briefly touches on the limitations of the proposed method. This paper has no negative social impact.

---

> ### Author Rebuttal · Authors · 2024-08-06
>
> > **W1:** While these experiments ... argument for the method's generalizability.
>
> **A1:** We appreciate the suggestion that makes this paper more complete and closer to realistic applications. We further examined our method using a shopping mall paid search campaign dataset [1]. This dataset contains data from a five-month paid search campaign for a U.S. shopping mall, spanning from July to November 2021. In this dataset, we focus on the variables *Impressions*, *Clicks*, and *Conversions*, which are fundamental count data in e-commerce scenarios. These variables follow causal relationships: *Impressions* $\rightarrow$ *Clicks*, *Clicks* $\rightarrow$ *Conversions*, and *Impressions* $\rightarrow$ *Conversions*, forming a triangular structure.
>
> Our method successfully identifies the adjacent vertices between *Impressions* and *Clicks*, as well as the other two causal directions. This outcome is consistent with our theoretical result, which means that our method can be applied to more real-world scenarios.  Following your suggestion, we have added these experimental results to our paper.
>
> [1] https://www.kaggle.com/datasets/marceaxl82/shopping-mall-paid-search-campaign-dataset
>
> > **W2:** The symbols T and the corresponding ... which may cause significant confusion for readers.
>
> **A2:** Thank you for your suggestion, which will help enhance the clarity and readability of our paper. We have added an explanation of $T_{X_j}$ in the caption of Figure 3: ``Illustration of the graphical implications in the closed-form solution for the PGF of PB-SCM. Here, $\alpha_{i,j} T_{X_j}(1)$ indicates that $X_j$ is reached from $X_i$ in this branching structure, while $(1-\alpha_{i,j})T_{X_j}(0)$ indicates the opposite."
>
> > **W3:** The term ch(i) in Formula 3 lacks a definition. It is possibly intended to be Des(i) as defined earlier in the text.
>
> **A3:** Thank you for your careful review and for pointing this out. In the context of a given DAG $G(\mathbf{V}, \mathbf{E})$, $Ch(i)$ is defined as $Ch(i) = \\{j \mid i \to j \in \mathbf{E}\\}$, representing the set of children of vertex $i$. We have added this definition alongside the definitions of $Pa(i)$ and $Des(i)$ to ensure clarity.
>
> > **Q1:** What are the accuracies of learning the Probability Generating Function and the subsequent causal structure learning? What are the bottlenecks of the proposed method?
>
> **A4:** Thank you very much for this suggestion to improve the completeness of our paper.
> The accuracies of our method are supported by the fact that the empirical probability generating function (ePGF) $\bar G_{\mathbf{X},n}(\mathbf{z}) = \frac{1}{n}\sum_{i=1}^{n} z_{1}^{X_{1}^{(i)}}\cdots z_{d}^{X_{d}^{(i)}}$ is an unbiased estimator of the PGF $G_{\mathbf{X}}(\mathbf{z})$ and almost surely converges to $G_{\mathbf{X}}(\mathbf{z})$ as the sample size $n\to \infty$, according to the strong law of large numbers [2][3].
>
> However, as mentioned in the conclusion section, the performance of our method can be limited by the scale of the dimension. While the ePGF exhibits effectiveness in estimating PGF, it probably encounters challenges when dealing with high-dimensional data, where the main reason is that the product of multiple $ z_{i}^{X_{i}}$ can result in an extremely small number. Thus, a promising future direction is to develop a sample-efficient method for estimating PGF.
>
> [2] Esquível, Manuel. "Some applications of probability generating function based methods to statistical estimation." Discussiones Mathematicae Probability and Statistics 29.2 (2009): 131-153.
>
> [3] Nakamura, Miguel, and Víctor Pérez-Abreu. "Empirical probability generating function: An overview." Insurance: Mathematics and Economics 12.3 (1993): 287-295.
>
> > **Q2:** The experiments on real-world data provide insufficient information. How do other methods perform on the real datasets?
>
> **A5:** Thanks for your valuable suggestion to improve the completeness of our experiments. We have further examined the performance of other methods on the real datasets. The cumulant-based method successfully identified the correct direction of edges, except for $ F\rightarrow R$. This can be attributed to the fact that only a few fouls directly cause a red card, thus leading to estimation issues with the cumulant method. As for PC, GES, and OCD, they struggle to recover the structure, with their F1 scores being 0.53, 0.18, and 0.44 respectively. Following your suggestion, we will add these experimental results to our paper.
>
> > **Q3:** How does the proposed method perform on networks that are not purely Poisson Bayesian networks or on mixed networks of general Bayesian and Poisson Bayesian networks? Additionally, does this method require prior knowledge that the data originates from a non-Poisson Bayesian network?
>
> **A6:** Thank you for this insightful question, which allows us to clarify the scope and applicability of our method. The main contribution of our paper is to propose a more comprehensive identifiability result for PB-SCM using PGF. Despite that our PGF-based method may generate an incorrect graph structure when applied to data with inappropriate distributions - since different distributions have different PGFs - We want to emphasize that the assumptions of PB-SCM are widespread in many real-world scenarios.
>
> By leveraging the thinning operation, PB-SCM explicitly models the inherent branching relationships between discrete variables, particularly for count data. Intuitively, the influence from a parent vertex to a child vertex is directly represented by the change in event counts, as the thinning operation outputs an integer representing the contribution of the count from the parent event. This directly reflects the dynamics of count data, effectively modeling the natural generating process in many real-world systems. Therefore, although our method relies on model assumptions, these assumptions are both common and typical.

---

> > ### Comment · Reviewer_CtKZ · 2024-08-12
> >
> > Thanks for your response, it addresses my concerns. I will keep my rating.

---

> > > ### Author Response · Authors · 2024-08-12
> > > **Thank you!**
> > >
> > > Thank you for your time and thoughtful review. We're pleased with your positive feedback and look forward to your continued support in the discussion phases. Thanks again!

---

### Official Review · Reviewer_nGrv · 2024-07-04

**Soundness:** 4
**Presentation:** 3
**Contribution:** 4
**Rating:** 7
**Confidence:** 4

**Summary:**

The paper focuses on causal discovery from observational count data, proposing a method to address the identifiability gap in Poisson Branching Structural Causal Models (PB-SCM) using Probability Generating Function (PGF). The authors develop a closed-form solution for the PGF of PB-SCM, enabling the identification of local causal structures by testing component appearances in the PGF. The effectiveness of the method is demonstrated through experiments on both synthetic and real datasets.

**Strengths:**

The paper has good presentation and motivation.

The authors derive a compact and exact closed-form solution for the PGF of PB-SCM, establishing a connection between the PGF and the causal structure.

The paper proposes a practical algorithm to learn causal skeletons and identify causal directions in PB-SCM using the derived PGF.

The proposed method is validated through experiments on synthetic and real-world datasets, demonstrating its effectiveness and superiority over existing methods in identifying causal structures in count data.

**Weaknesses:**

The proposed method cannot handle high-dimensional covariates.

The orientation of edges is not detailed.

Figure 4, the caption should be correct.
``F: Foul, Y1: Yellow card, Y2: Second yellow card, R: Red card, S: Substitution, H: Hand ball'' -> ``$F$: Foul, $Y_1$: Yellow card, $Y_2$: Second yellow card, $R$: Red card, $S$: Substitution, $H$: Hand ball''

The soundness of the proposed method requires the causal sufficiency assumption, which limits the applications of the method in real-world scenarios.

Minor:

Building on this, We -> Building on this, we

**Questions:**

n/a

**Limitations:**

Yes

---

> ### Author Rebuttal · Authors · 2024-08-06
>
> > **W1:** The proposed method cannot handle high-dimensional covariates.
>
> **A1:** Thank you for pointing out this issue; this will be an important problem for us to address in future work. As mentioned in the conclusion section, the performance of our method can be limited by the scale of the dimension.
> While the empirical PGF is effective in estimating the PGF, it encounters challenges when dealing with high-dimensional data. The main reason is that the product of multiple $\displaystyle z_{i}^{X_{i}}$ can result in extremely small numbers, which affects numerical stability and accuracy. Therefore, a promising future direction is to develop a sample-efficient method for estimating the PGF to better handle high-dimensional covariates.
>
> > **W2:** The orientation of edges is not detailed.
>
> **A2:** Thank you for pointing out the need for more detail on edge orientation. According to the proposed closed form of PGF, the terms in the PGF correspond directly to the graphical structure, thus we can perform orientation by checking whether specific terms exist in PGF.
>
> For instance, Eq. (5) in the paper presents the closed form of the PGF for the triangular structure in Fig. 2. The presence of the term $z_1z_2z^2_3$ in Eq. (5) indicates that there are two directed paths leading to $X_3$ from either $X_1$ or $X_2$. Here, the order of $z_3$ is 2, meaning that $X_3$ is the vertex with two in-degrees in the triangular structure. Consequently, we can orient the edges as $X_1 \to X_3$ and $X_2 \to X_3$. Building on this, we orient each edge by considering a local triangular structure with this edge and examining whether this edge points to a vertex with two in-degrees, as described in Algorithm 1 (Lines 5-7) in the paper.
>
> Otherwise, if an edge $X_i - X_j$ cannot be part of a triangular structure, we attempt to orient it in a pattern such as $X_i - X_j \leftarrow X_k$ or $X_i - X_j - X_k$ with another vertex $X_k$. Specifically, by examining whether the term $z_iz_jz_k$ does not appear in the local PGF involving $X_i, X_j, X_k$, we can conclude that there is no directed path between $X_i$ and $X_j$ through $X_k$. In such cases, we orient the edges as $X_i \to X_j$, and $X_j \leftarrow X_k$ as described in Algorithm 1 (Lines 8-10).
>
> Following your suggestion, we will provide a more detailed description of the orientation process in the method section.
>
> > **W3:** Figure 4, the caption should be correct.
>
> **A3:** Thank you for pointing out the typos and providing the correction. Following your suggestion, we have corrected the caption of Figure 4: Football Dateset Result ($F$: Foul, $Y_1$: Yellow card, $Y_2$: Second yellow card, $R$: Red card, $S$: Substitution, $H$: Handball)
>
> > **W4:** The soundness of the proposed method requires the causal sufficiency assumption, which limits the applications of the method in real-world scenarios.
>
> **A4:** Thank you for your valuable suggestion. Relaxing the causal sufficiency assumption will indeed be a focus of our future work. Nevertheless, this assumption is a common assumption in many causal inference methods. Several prominent works in the field also utilize this assumption. For instance, Spirtes et al. in their seminal work [1] on causal discovery algorithms assume causal sufficiency to ensure accurate inference of causal structures. Similarly, Pearl discusses the importance of causal sufficiency in structural causal models [2]. These references highlight that while the causal sufficiency assumption can limit applicability in some real-world scenarios, it is a foundational assumption that underpins many existing causal inference methods. In future work, we will further explore the identification of latent confounders to address the limitations posed by this assumption.
>
> [1] Spirtes, Peter, Clark Glymour, and Richard Scheines. Causation, prediction, and search. MIT press, 2001.
>
> [2] Pearl, Judea. Causality. Cambridge university press, 2009.

---

> > ### Comment · Reviewer_nGrv · 2024-08-12
> >
> > Thank you for your rebuttal. My concerns are fully addressed. I will keep my rating.

---

> > > ### Author Response · Authors · 2024-08-12
> > > **Thank you!**
> > >
> > > Thank you for taking the time to review our manuscript. We're glad to see your positive feedback and appreciate your continued support during the discussion phases. Thanks again!

---

### Official Review · Reviewer_Gqam · 2024-07-12

**Soundness:** 4
**Presentation:** 4
**Contribution:** 4
**Rating:** 8
**Confidence:** 4

**Summary:**

This paper addresses the identifiability of Poisson Branching Structure Causal Model (PB-SCM) using the probability generating function, during which a compact, exact closed-form PGF solution is developed and the identifiability of PB-SCM is complete based on such relation. A practical algorithm for learning causal skeletons and causal directions using PGF is proposed, with effectiveness demonstrated on synthetic and real datasets.

**Strengths:**

- This paper addresses the identification of the PB-SCM model using a probability-generating function. Compared with the previous cumulant-based, the identifiability of PB-SCM is further complete.
- The theory releases an interesting connection between the probability-generating function and the graph of PB-SCM which allows us to further explore the identifiability of the PB-SCM.
- Based on the theoretical results an efficient algorithm is proposed, and the effectiveness of the proposed method is also verified in the experiments.
- The writing is clear and well-organized and the intuition of the proposed method is effectively conveyed.

**Weaknesses:**

- This paper states that the cumulant-based method is exactly the method that identifies the causal direction by detecting the highest order of $z_i$. Can you explain why? It would be interesting to see the connection between these two methods.
- The local collider structure used in section 4 seems to be referred to the unshielded collider structure which should be stated clearly to avoid ambiguity.
- Theorem 7 states that the identifiability of the causal direction is given the known causal skeleton. Can it be extended to address the general identifiability of the whole causal structure?
- Since the paper is quite dense, more illustrations can be provided in the toy example to illustrate the connection between PGF and the graph and how it leads to the identifiability.
- The citing format should be improved as it is mixed with the main text.

Typos:
- Line 204 similarly

**Questions:**

- What is the connection between the cumulant-based method and the proposed PGB-based method?
- Can you further improve Theorem 7?

**Limitations:**

The authors had adequately addressed the limitations in their conclusion and the broader societal impacts in their appendix.

---

> ### Author Rebuttal · Authors · 2024-08-06
>
> > **W1:** This paper states that the cumulant-based ... connection between these two methods. \& **Q1:** What is the connection between the cumulant-based method and the proposed PGF-based method?
>
> **A1:** Thank you for your question allowing us to further clarify the connection between the cumulant-based method and our method, as well as the advantages of our method.
>
> The cumulant-based method [1]  introduces the concept of $k$-path cumulants summation, denoted as $\tilde\Lambda_{k}(X_{i} \leadsto X_{j})$. Here, $\tilde\Lambda_{k}(X_{i}\leadsto X_{j})\neq 0$ indicates the existence of a common ancestor $m$ for vertices $i$ and $j$, with $k$ directed paths from $m$ to $j$. Such path information is encoded within the PGF. If there are $k$ paths from vertex $m$ to $j$, terms involving $z_m z_j^k$ appear in the PGF. Thus, detecting paths to vertex $j$ is equivalent to identifying the order of $z_j$ in the PGF.
>
> However, only the highest non-zero order of $\tilde\Lambda_{k}(X_{i} \leadsto X_{j})$, involving $k+1$ order cumulants, helps identify the causal direction, as it does not show asymmetry at lower orders. For example, consider vertices $X_i$, $X_j$, and $X_m$, where $X_i \to X_j$, and there are $k$ paths from $X_m$ to $X_i$, and $k+p$ paths from $X_m$ to $X_j$ where $p \geq 1$. Here, $\tilde\Lambda_{k}(X_{i} \leadsto X_{j}) \neq 0$ and $\tilde\Lambda_{k}(X_{j}\leadsto X_{i}) \neq 0$, and $\tilde\Lambda_{k+1}(X_{i} \leadsto X_{j}) \neq 0$ while $\tilde\Lambda_{k+1}(X_{j} \leadsto X_{i}) = 0$. The latter shows an asymmetry that the former does not. This means that the cumulant-based method must detect the highest non-zero order of $\tilde\Lambda_{k}(X_{j} \leadsto X_{i})$, i.e., the highest order of $z_i$ in PGF terms, to reveal the asymmetry. Moreover, certain local (unshielded) collider structures, such as $X_1 \to X_2$ and $X_3 \to X_2$ in Fig. 1(a), are non-identifiable using this method because both directions result in $\tilde\Lambda_{k=1} \neq 0$ and $\tilde\Lambda_{k=2} = 0$, leading to non-identifiability.
>
> In contrast, our method fully leverages lower-order information due to the local property of the PGF. By removing redundant paths by setting the corresponding $z$ to zero in the PGF, we focus on a small local structure, avoiding the need for high-order information.
>
> [1] Qiao, Jie, et al. "Causal Discovery from Poisson Branching Structural Causal Model Using High-Order Cumulant with Path Analysis." Proceedings of the AAAI Conference on Artificial Intelligence. Vol. 38. No. 18. 2024.
>
> > **W2:** The local collider structure ... stated clearly to avoid ambiguity.
>
> **A2:** Thank you for your insightful question, which helps us clarify the terminology used in our paper. Exactly, the local collider structure in this paper can be referred to as the unshielded collider structure. Following your suggestion, we have added the definition of a local structure: Given a DAG $G(\mathbf{V},\mathbf{E})$, a local structure $G'(\mathbf{L},\mathbf{E_L})$ is a subgraph of $G(\mathbf{V},\mathbf{E})$ with vertex set $\mathbf{L} \subset \mathbf{V}$ and edge set $\mathbf{E_L} = \\{i \to j \mid i,j \in \mathbf{L}, i \to j \in \mathbf{E}\\}$.
>
> Given vertices $i, j, k$, if $i \to j \leftarrow k$ and $i$ and $k$ are not adjacent, they form an unshielded collider structure, which we refer to as a local collider structure. If there is a direct edge between $i$ and $k$, this edge will be included in the local structure formed by $i, j, k$, known as a local triangular structure. We will clearly define these terms to avoid ambiguity.
>
> > **W3:** Theorem 7 states ... identifiability of the whole causal structure? \& **Q2:** Can you further improve Theorem 7?
>
> **A3:** Thank you for your valuable suggestion.
> Indeed, it is theoretically possible to identify the whole causal structure without relying on the causal skeleton because the PGF uniquely encodes distribution, corresponding to a unique causal structure. However, the complexity of such an approach can be exceedingly high.
>
> Let us clarify what this identifiability issue is and why learning the causal skeleton is essential for addressing the identifiability of the whole causal structure.
> As a foundation of identifiability, the closed-form solution of the PGF is directly associated with the causal structure,
> which allows us to recover the causal structure by exploring whether different structures correspond to the form of PGF.
> However, if we try to directly analyze the whole causal structure, the number of possible forms of PGF we have to examine can be large since the number of structures significantly increases with the number of vertices. The large search space makes structural identifiability difficult.
>
> Therefore, to address this issue, we introduce the local PGF, which enables us to examine its form in a small local structure, i.e., local triangular structure and local (unshielded) collider structure. This approach reduces the complexity of the search space by focusing on identifiable local patterns. Therefore, we require the causal skeleton and prioritize the identifiability of adjacent vertices in the Identifiability section.
>
> > **W4:** Since the paper is dense, more illustrations ... identifiability.
>
> **A4:** Thank you for your suggestion, we will add more illustrations to the toy example to enhance understanding. Specifically, we have enhanced the description of $T_{X_i}$ in the caption of Fig. 3: ``Here, $\alpha_{i,j}T_{X_j}(1)$ indicates that $X_j$ is reached from $X_i$ in this branching structure, while $(1-\alpha_{i,j})T_{X_j}(0)$ indicates the opposite.'' This aims to provide an intuitive understanding of the connection between the PGF and the graph structure.
>
> > **W5:** The citing format ... main text.
>
> **A5:** Thank you for your suggestion, which will help enhance the clarity and readability of our paper. We will change the citation format to use square brackets to clearly distinguish them from the main text.

---

> > ### Comment · Reviewer_Gqam · 2024-08-12
> >
> > Thanks for the detailed responses. The responses have well addressed my concerns and questions, which further confirms my rating.

---

> > > ### Author Response · Authors · 2024-08-12
> > > **Thank you!**
> > >
> > > We sincerely appreciate the time and effort you have dedicated to reviewing our manuscript. We are grateful for your positive evaluation of our work and are hopeful for your continued support in the subsequent discussion phases. Thank you!

---

### Official Review · Reviewer_w346 · 2024-07-13

**Soundness:** 3
**Presentation:** 3
**Contribution:** 3
**Rating:** 7
**Confidence:** 5

**Summary:**

This paper investigates the causal structure learning on the Poisson Branching Structural Causal Model (PB-SCM)and its identifiability using probability-generating function (PGF). The identifiability is established by developing the closed-form solution of the PGF which can be utilized to identify the causal structure of PB-SCM. The proposed methods are validated through synthetic experiments and real-world experiments.

**Strengths:**

- This work proposes a PGF-based method for identifying the causal structure which is interesting.
- The authors propose the closed-form solution of PGF and establish the connection to the graph. By this, this work shows that there still remains a gap in the identifiability of the PB-SCM and addresses this gap using the local property of the PGF. This is a novel and significant contribution.
- The proposed method is overall sound with a theoretical guarantee.
- This paper is well-present and well-motivated.

**Weaknesses:**

- The authors propose to use the local PGF for identifying the causal structure. It seems that the global PGF can also be used for identifying the causal structure and it would be more beneficial if more discussions could be involved.
- What is the benefit of using the local property of PGF? Is it possible to set the z_i not approach to zero?
- The authors propose to use the Gaussian distribution for the rank test. It would be better to illustrate why the trace of the matrix converges to a normal distribution.
- The experiments only contain some random experiments. It would be more beneficial if there are some case studies.
- The experiments should explain the arrow in the table after the metric.

**Questions:**

See weakness

**Limitations:**

See weakness

---

> ### Author Rebuttal · Authors · 2024-08-06
>
> > **Q1**: The authors propose to use the local PGF for identifying the causal structure. It seems that the global PGF can also be used for identifying the causal structure and it would be more beneficial if more discussions could be involved.
>
> **A1**:Thank you very much for this suggestion to improve the completeness of our paper. It is indeed theoretically feasible to identify causal structures through the global PGF. However, using the PGF directly to determine causal directions involves computing higher-order derivatives, which can complicate the implementation and thus affect its feasibility.
>
> To illustrate this point, we provide a toy example with a causal structure of 5 vertices, as shown in Fig. R1 of our attached PDF. In this example, we focus on identifying the edge $X_4 \to X_5$. For the correct structure where $X_4 \to X_5$, the terms $C_1 \times z_1z_2z_3z_4^2z_5^4$ and $C_2 \times z_1z_2z_3z_4^2z_5^2$ exist in the PGF. Conversely, for the reverse direction $X_4 \leftarrow X_5$, the terms $C'_1 \times z_1z_2z_3z_4^4z_5^2$ and $C'_2 \times z_1z_2z_3z_4^2z_5^2$ appear in the PGF.
>
> Notably, as shown in Fig. R1 (c) and (f), the term $z_1z_2z_3z_4^2z_5^2$ exists in the PGF for both structures. This is because there are always at least two directed paths from $X_1$ to each of $X_4$ and $X_5$, regardless of the direction between $X_4$ and $X_5$. This implies that taking the second derivative with respect to $z_4$ does not reveal any asymmetry between the structures.
> Therefore, if we want to identify the direction using the global PGF, we have to apply a test involving the third derivative that distinguishes the correct structure by showing the absence of terms with $z_4^3$ in the PGF for the correct direction. Following your suggestion, we have added this toy example and discussion to the paper to enhance the understanding of the connection between the PGF and causal structures.
>
> > **Q2**: What is the benefit of using the local property of PGF? Is it possible to set the $\displaystyle z_{i}$ not approach to zero?
>
> **A2**: Thank you for this valuable question to emphasize our contributions. As demonstrated in the answer to Q1, it is possible to use the PGF directly without setting $z_{i}$ to approach zero when identifying directions. However, when the graph structure becomes more complex and the number of paths between vertices increases, higher-order derivatives are required, making implementation challenging.
>
> Therefore, we propose using the local PGF, which simplifies the process by isolating specific terms in the PGF. This is done by setting other $z_i$ to approach zero, effectively reducing the complexity of the derivative calculations. In the example above, by letting $z_2$ and $z_3$ approach zero, we can remove terms in the PGF that involve $z_2$ and $z_3$, such as $C_1 \times z_1z_2z_3z_4^2z_5^2$ and $C_2 \times z_1z_2z_3z_4^2z_5^4$. This reduction allows us to focus on examining the local triangular structure formed by $X_1$, $X_4$, and $X_5$, which only requires estimating the second derivative.
>
> > **Q3**:  The authors propose to use the Gaussian distribution for the rank test. It would be better to illustrate why the trace of the matrix converges to a normal distribution.
>
> **A3**: Thank you for this question. As mentioned in Section 4.5, we use the empirical PGF (ePGF) $\bar G_{\mathbf{X},n}(\mathbf{z}) = \frac{1}{n}\sum_{i=1}^{n} z_{1}^{X_{1}^{(i)}} \cdots z_{d}^{X_{d}^{(i)}}$ to estimate the PGF $ G_{\mathbf{X}}(\mathbf{z})$, which is an unbiased estimator of the PGF [1][2]. According to the central limit theorem, the quantity $ n^{1/2}\\{\bar G_{\mathbf{X},n}(\mathbf{z}) - G_{\mathbf{X}}(\mathbf{z}) \\}$ converges in distribution to a normal distribution $ N(0, \sigma^2)$ with zero mean and variance $\sigma^2$, which can be estimated using the bootstrap method. Therefore, in matrix $\mathbf{A}^{\\{i,j\\}}$, the estimations of the local PGF $ G_{\mathbf{X}}^{\\{i,j\\}}(\mathbf{z})$ and its partial derivative $\frac{\partial^2 G_{\mathbf{X}}^{\\{i,j\\}}(\mathbf{z})}{\partial z_{i} \partial z_{j}}$ also converge in distribution to normal distributions.
> Since the sum of two normal distributions is also a normal distribution, the traces of matrices $\mathbf{A}^{\\{i,j\\}}$ converge to a normal distribution. The same reasoning applies to matrices $\mathbf{B}^{\\{i,j,k\\}}$ and $\mathbf{C}^{\\{i,j,k\\}}$.
>
> [1] Esquível, Manuel. "Some applications of probability generating function based methods to statistical estimation." Discussiones Mathematicae Probability and Statistics 29.2 (2009): 131-153.
>
> [2] Nakamura, Miguel, and Víctor Pérez-Abreu. "Empirical probability generating function: An overview." Insurance: Mathematics and Economics 12.3 (1993): 287-295.
>
> > **Q4**: The experiments only contain some random experiments. It would be more beneficial if there are some case studies.
>
> **A4**: Thank you for your valuable suggestion, which allows our experiments more comprehensive. We have added three case studies involving causal graphs with 3, 4, and 5 vertices respectively. The results are presented in Table R1 in our attached PDF.
>
> > **Q5**: The experiments should explain the arrow in the table after the metric.
>
> **A5**: Thank you for your suggestion. We have added a description of the arrows in the table: ↑ indicates that a higher value is better, while ↓ indicates that a lower value is better.

---

### Author Rebuttal · Authors · 2024-08-06

Dear Reviewers w346, Gqam, nGrv, and CtKZ,

Thanks for the thoughtful and constructive reviews, which improve the completeness and readability of our paper. It is encouraging that reviewers think that the proposed PGF-based method for identifying PB-SCM is novel (w346, Gqam, CtKZ) and interesting (Gqam), our theoretical result is sound (w346, CtKZ), our paper is well-present and well-organized(w346, Gqam, nGrv), and our experimental results show the effectiveness (Gqam, nGrv) of our PGF-based method. We here provide a general response to summarize the modifications of the paper.

- To Reviewer w346, we have added an additional discussion on using the (global) PGF to identify the causal structure.
- To Reviewer w346, we have explained the benefits of using the local properties of the PGF.
- To Reviewer w346, we have clarified why the trace of the matrix converges to a normal distribution.
- To Reviewer w346, we have included three case studies in our attached PDF involving causal graphs with 3, 4, and 5 vertices, respectively.
- To Reviewer w346, we have added descriptions of the arrows in Table 1 and Table 2 of the paper.
- To Reviewer Gqam, we have illustrated the connection between the cumulant-based method and our PGF-based method.
- To Reviewer Gqam, we have clarified the terminology used for the local collider structure.
- To Reviewer Gqam, we have explained the necessity of the causal skeleton in addressing the identifiability of the entire causal structure.
- To Reviewer Gqam, we have improved the description of the toy example and the citation format.
- To Reviewer nGrv, we have discussed the limitations of our method and the reasonableness of the causal sufficiency assumption.
- To Reviewer nGrv, we have added a detailed description of how to orient the edges.
- To Reviewer nGrv, we have corrected the caption of Fig. 4.
- To Reviewer CtKZ, we have added a discussion on the accuracies and bottlenecks of our PGF-based method.
- To Reviewer CtKZ, we have included an additional real-world experiment, and provided the performance of other methods in the real-world experiment of the paper.
- To Reviewer CtKZ, we have added an explanation of $T_{X_i}$ in the caption of Fig. 3 and included the definition of $Ch(i)$.
- To Reviewer CtKZ, we have discussed potential issues our method may encounter when assumptions are violated.
- To all reviewers, we have polished our paper and corrected the pointed-out typos.

Thanks again for your time dedicated to carefully reviewing this paper. We hope that our response properly addresses your concerns.

With best regards,
Authors of submission 17086

---

### Decision · Program_Chairs · 2024-09-25

**Decision:**

Accept (spotlight)

**Comment:**

This paper  has received very positive feedback from reviewers, who praised its novel approach to identifying causal structures in Poisson Branching Structural Causal Models (PB-SCM) using Probability Generating Functions (PGF). The reviewers highlighted the paper's strong theoretical contributions, including the development of a closed-form PGF solution and an efficient algorithm for learning causal structures.  The authors provided a comprehensive rebuttal, addressing concerns by clarifying theoretical connections, improving explanations, adding case studies, and discussing the method's limitations and future directions. Overall, the paper is considered technically sound and impactful, with reviewers recommending its acceptance with minor revisions to enhance clarity and completeness.